# Sea ice floe size: its impact on pan-Arctic and local ice mass, and required model complexity

Adam W. Bateson[1], Daniel L. Feltham[1], David Schröder[1,2], Yanan Wang[3], Byongjun Hwang[3], Jeff K. Ridley[4], Yevgeny Aksenov[5]

[1]Centre for Polar Observation and Modelling, Department of Meteorology, University of Reading, Reading, RG2 7PS, United Kingdom
[2]British Antarctic Survey, Cambridge, CB3 0ET, United Kingdom
[3]School of Applied Sciences, University of Huddersfield, Huddersfield, United Kingdom,
[4]Hadley Centre for Climate Prediction and Research, Met Office, Exeter, EX1 3PB, United Kingdom
[5]National Oceanography Centre Southampton, Southampton, SO14 3ZH, United Kingdom

*Correspondence to*: Adam W. Bateson (a.w.bateson@pgr.reading.ac.uk)

**Abstract**

Sea ice is composed of discrete units called floes. Observations show that these floes can adopt a range of sizes spanning orders of magnitude, from metres to tens of kilometres. Floe size impacts the nature and magnitude of interactions between the sea ice, ocean, and atmosphere including lateral melt rate and momentum and heat exchange. However, large-scale geophysical sea ice models employ a continuum approach and traditionally either assume floes adopt a constant size or do not include an explicit treatment of floe size. In this study we apply novel observations to analyse two alternative approaches to modelling a floe size distribution (FSD) within the state-of-the-art CICE sea ice model. The first model considered is a prognostic floe size-thickness distribution where the shape of the distribution is an emergent feature of the model and is not assumed a priori. The second model considered, the WIPoFSD (Waves-in-Ice module and Power law Floe Size Distribution) model, assumes floe size follows a power law with a constant exponent. We introduce a parameterisation motivated by idealised models of in-plane brittle fracture to the prognostic model and demonstrate that the inclusion of this scheme enables the prognostic model to achieve a reasonable match against the novel observations for mid-sized floes (100 m – 2 km). While neither FSD model results in a significant improvement in the ability of CICE to simulate pan-Arctic metrics in a stand-alone sea ice configuration, larger impacts can be seen over regional scales in sea ice concentration and thickness. We find that the prognostic model particularly enhances sea ice melt in the early melt season, whereas for the WIPoFSD model this melt increase occurs primarily during the late melt season. We then show that these differences between the two FSD models can be explained by considering the effective floe size, a metric used to characterise a given FSD. Finally, we discuss the advantages and disadvantages to these different approaches to modelling the FSD. We note that although the WIPoFSD model is unable to represent potentially important features of annual FSD evolution seen with the prognostic model, it is less computationally expensive and produces a better fit to novel FSD observations derived from 2-m resolution MEDEA imagery, possibly making this a stronger candidate for inclusion in climate models.

## 1 Introduction

The Arctic sea ice cover consists of contiguous pieces of sea ice referred to as floes (WMO, 2014). Floe size has a direct impact on several processes that are important to the evolution of the sea ice, including lateral melt rate (Steele, 1992; Bateson et al., 2020); momentum exchange between the sea ice, ocean, and atmosphere (Lüpkes et al., 2012; Tsamados et al., 2014); surface moisture flux over sea ice (Wenta and Herman, 2019); sea ice rheology i.e. the mechanical response of sea ice to stress (e.g. Shen et al., 1986; Wilchinsky and Feltham, 2006; Rynders et al., 2022); and the clustering of sea ice into larger agglomerates (Herman, 2012). Historically, continuum sea ice models such as CICE (Hunke et al., 2015) have assumed that floes are of a uniform size or do not explicitly consider floe size at all when evaluating sea ice thermodynamics (Bateson et al., 2020; Keen et al., 2021) or dynamics (Tsamados et al., 2014). In contrast, observations show that floe sizes can span a large range, from metres to tens of kilometres (Stern et al., 2018a). Model studies suggest that floe size has a non-negligible impact on sea ice

extent and volume through changing lateral and total sea ice melt, particularly in areas where the sea ice cover largely consists of small floes (Bateson et al., 2020; Bateson, 2021a). Floe size has been found to be particularly important in the Marginal Ice Zone (MIZ); a region of sea ice cover influenced by waves and swell penetrating from the open ocean (Aksenov et al., 2017; Roach et al., 2019; Bateson et al., 2020). The MIZ is taken in this study as regions with sea ice concentration between 15 – 80

%, a definition commonly used due to an absence of observations of ocean surface waves in sea ice over the necessary spatial scales and timescales (Strong et al., 2017; Horvat et al., 2020).

Observations of the floe size distribution (FSD) show a large ratio of smaller floes to larger floes; this distribution of floe sizes is often summarized using a truncated power law (Rothrock and Thorndike, 1984; Toyota et al., 2006; Perovich and Jones, 2014; Stern et al., 2018b). Studies generally show that a power law produces a reasonable fit to the observations presented,

though the validity of using a power law to fit floe size data remains an open question (Stern et al., 2018b), with several studies disputing the extent to which a power law is a good description of the FSD (Herman, 2010; Horvat et al., 2019; Herman et al., 2021). The exponents of power laws fitted to observations of the non-cumulative floe number density show a large amount of variability, from -1.9 to -3.5 (see summary of observations in Stern et al., 2018a). Observations show spatial and temporal variability of the FSD. Stern et al. (2018b) analysed satellite imagery collected over the Beaufort and Chukchi seas and reported

an approximately sinusoidal seasonal cycle in the exponent with a minimum exponent of about -2.8 in August and a maximum exponent of about -1.9 in April in both 2013 and 2014 for floes larger than 2 km. Perovich and Jones (2014) also found evidence of seasonal variation in the exponent; aerial photographic imagery was analysed from the Beaufort Sea over the period June to September 1998 for floes between 10 m to 10 km in size. They noted a change in exponent from -3.0 over June and July to -3.2 in late August, coinciding with a high wind speed event driving fragmentation of floes under wind and ocean

stress. The exponent then increased to over -3.1 by September due to sea ice freeze-up and floe welding.

Modelling studies have been used to understand how the observed FSD shape and behaviours could emerge from relevant processes. These FSD models can be roughly divided into two different classes: (i) models where the shape of the FSD emerges from the constituent sea ice dynamical-thermodynamical processes (e.g. Zhang et al., 2016; Roach et al., 2018, 2019); (ii) models where the general shape of the FSD is fixed, generally to a power law (e.g. Bennetts et al., 2017; Bateson et al., 2020).

Hybrid approaches have also been proposed, e.g. Boutin et al. (2020) allows the shape of the FSD to evolve in response to processes such as lateral melting, but resets the distribution to a power law after a wave break-up event. These modelling studies have incorporated one or several processes that have been observed to influence floe size: lateral melting and growth at the edges of floes (e.g. Perovich and Jones, 2014; Roach et al., 2018); break-up of sea ice floes into smaller pieces from ocean waves (e.g. Kohout et al., 2014); floes welding together in ocean freeze-up conditions (Roach et al., 2018); the formation

mechanism of new floes (Roach et al., 2018); and rafting and ridging of floes during floe collisions (Horvat and Tziperman, 2015). The limited spatial and temporal coverage of floe size observations has prohibited effective evaluation of these models, though there have been recent efforts to develop satellite-derived FSD products to enable such evaluations (Horvat et al., 2019). It is nevertheless anticipated that important processes are not yet represented in these models. For example, thermodynamically-driven break-up of floes along existing cracks and refrozen leads in the sea ice cover (Perovich et al.,

35 2001).

In this study we will consider the prognostic FSTD (Floe-Size-Thickness distribution model) of Roach et al. (2018, 2019) and the WIPoFSD model (Waves-in-Ice module and Power law Floe Size Distribution model) of Bateson et al. (2020). The prognostic model is within the model class where the shape of the distribution emerges primarily from parameterisations at the process level. The WIPoFSD model is within the class of models where the shape of the FSD in the model is actively

constrained, in this case by approximating the FSD as a power law. These models present useful case studies to examine the advantages and disadvantages of different approaches to modelling the FSD and its impacts on sea ice. We introduce a new quasi-restoring brittle fracture scheme into the prognostic model, which crudely accounts for in-plane fracture processes in winter and thermodynamically-driven break-up of floes along existing cracks and other linear features over the subsequent

melt season (Bateson, 2021a). We complete simulations including the FSD models within a version of CICE where floe size impacts both lateral melt volume and momentum exchange between the sea ice, ocean, and atmosphere. We compare the performance of the prognostic model both with and without the brittle fracture scheme in simulating the shape of novel observations of the FSD and also assess the accuracy of a power-law fit to these observations. By examining the impact of the two FSD models on the sea ice mass balance, we consider whether either FSD model can improve the performance of CICE in our model configuration. The impact of both FSD models on key sea ice and MIZ metrics is investigated, including their interannual variability and spatial differences. Finally, we explore how differences in the impacts of the two models emerge and consider the implications of the results presented here for different strategies in modelling the FSD.

This paper is structured as follows. Section 2 describes the CICE model setup used in this study, the two FSD models, and new model components and modifications introduced in this study. Section 3 describes the methodology for this research, including a description of novel observations of the FSD and an overview of model experiments. Section 4 presents the results of the analysis and simulations, divided into 3 sub-sections: a comparison of the modelled FSD to observations; a comparison of model output to the observed sea ice extent and volume reanalysis; and a pan-Arctic comparison of the impacts of the two FSD models. Section 5 discusses the results and section 6 presents conclusions and summarises the study. This study uses several key terms related to the sea ice floe size distribution. To ensure clarity, Table 1 summarises and defines these key terms.

## 2 Model description

Here we will use the CPOM (Centre for Polar Observation and Modelling) version of the Los Alamos Sea Ice model v5.1.2, known as CICE (Hunke et al., 2015). In section 2.1.1, we will outline key details of CICE that are pertinent to this study. Within the CPOM-CICE setup, we use the prognostic mixed-layer ocean model of Petty et al. (2014), and the form drag scheme of Tsamados et al. (2014). An overview of each of these model components are provided in sections 2.1.2 and 2.1.3, respectively. In section 2.1.4, we outline how the treatment of lateral melt and form drag in the CPOM-CICE setup has been modified for use with FSD models. In section 2.2 and 2.3 we will provide an overview of the two FSD models considered here: a modified version of the prognostic FSD model of Roach et al. (2018), and a modified version of the WIPoFSD model of Bateson et al. (2020).

### 2.1 Description of Standard Model Physics

### 2.1.1 Standard CICE model

In this study, we model the Arctic sea ice cover using a local version of the CICE sea ice model in a standalone setup. CICE is a continuum numerical model of sea ice that has been designed for use within fully coupled climate models. Full details of CICE can be found within Hunke et al. (2015). The model consists of several different components designed to simulate the evolution of sea ice on the geophysical scale including sea ice and snow thermodynamics, sea ice dynamics, a sea ice thickness distribution, and advection. The standard sea ice thickness distribution in CICE distributes ice area between five thickness categories, with the spacing increasing for thicker categories. The ice area in a given category evolves in response to dynamic and thermodynamic processes according to a linear remapping scheme (Lipscomb, 2001). Sea ice melt is subdivided into three separate components within CICE: melt from the upper surface of the sea ice floe (top melt), melt from the bottom surface of the floe (basal melt), and melt from the sides of the floe (lateral melt). The adaptation of the standard CICE lateral melt treatment for use with FSD models is described in section 2.1.4.

The lateral melt volume is explicitly calculated within CICE:

$$\frac{1}{A}\frac{dA}{dt}$$

$$= \frac{\pi}{\alpha_{shape}L}w_{lat.} \tag{1}$$

$A$ represents the sea ice area fraction, such that the term on the left-hand side, $\frac{1}{A}\frac{dA}{dt}$, represents the fractional rate of sea ice area

loss due to lateral melt (units of $s^{-1}$). The rate of sea ice volume loss from lateral melt is the product of $\frac{1}{A}\frac{dA}{dt}$ with the sea ice area and mean thickness. $\alpha_{shape}$ and $L$ are the constant floe shape and diameter parameters, set to 0.66 and 300 m in standard CICE. $w_{lat}$ is the lateral melt rate (units of $ms^{-1}$); it is a function of the elevation of the sea surface temperature above freezing for standalone CICE. The lateral heat flux, $F_{lat}$, is calculated from the volume of lateral melt (all fluxes have units of

$Jm^{-2}s^{-1}$). The melting or freezing potential at the sea ice-surface ocean interface, $F_{frzmlt}$, is calculated as:

$$F_{frzmlt}$$
$$= \frac{\Delta T c_{p,oc} \rho_w h_{mix}}{\Delta t}, \tag{2}$$

where $\Delta T$ is the difference in sea surface temperature from freezing (units of $K$), $c_{p,oc}$ is the specific heat capacity of the surface ocean ($Jkg^{-1}K^{-1}$), $\rho_w$ is the density of seawater ($kgm^{-3}$), $h_{mix}$ is surface mixed-layer depth (units of $m$), and $\Delta t$ is

the model timestep. The magnitude of $F_{frzmlt}$ is capped at a fixed value. Following CICE sign conventions, a negative value for $F_{frzmlt}$ corresponds to melting of sea ice. The following condition applies to the lateral and basal flux during periods of melt:

$$|F_{bot} + F_{lat}| \le |F_{frzmlt}|. \tag{3}$$

Here $F_{bot}$ is the net downward heat flux from the sea ice to the ocean. Where the melting potential is exceeded, $F_{bot}$ and $F_{lat}$

are both reduced by a common factor such that the condition set by Eq. (3) is satisfied.

**2.1.2 The mixed-layer model**

Ocean mixed-layer properties are important in determining lateral and basal melt rates, which are both relevant for evaluating the impact of floe size on the sea ice cover (e.g. Bateson et al., 2020). Here, a modified version of the prognostic bulk mixed-layer model of Petty et al. (2014) is used rather than a constant prescribed mixed-layer depth, to better represent sea ice-mixed

layer interactions and feedbacks (e.g. the ice-ocean albedo feedback) without the complexity and computational expense of a full ocean model (e.g. Frew et al., 2019). In this model, the mixed-layer temperature, salinity, and depth are all evaluated prognostically. The deep ocean below the mixed layer is restored to observations, and the model is zero-dimensional i.e. defined for each model grid cell without lateral interactions between grid cells. Full details of the original scheme are available in Petty et al. (2014). The original Petty et al. (2014) mixed-layer model was set-up and tested for the Southern Ocean where

the stronger winds and waves, weaker upper ocean stratification, and larger extent of the MIZ enables a high wind power input leading to a much deeper mixed-layer when compared to the Arctic. Here we adopt several adjustments to the mixed-layer model made by Tsamados et al. (2015) to ensure reasonable performance of the mixed-layer model in the Arctic. The three-component model of surface layer, mixed layer, and deep ocean is replaced with a two-component model, with just a mixed layer and deep ocean. In addition, the mixed-layer temperature and salinity are restored to the 10 m depth temperature and sea

surface salinity from a monthly climatology reanalysis dataset.

**2.1.3 Form drag scheme**

Recent versions of CICE include an implementation of the form drag scheme (Hunke et al., 2015) following Tsamados et al. (2014), which aims to better describe the turbulent momentum and heat exchange between the sea ice, ocean, and atmosphere by accounting for the topography of sea ice. The scheme of Tsamados et al. (2014) replaces the constant drag coefficients in

CICE with explicit representations of both form drag and skin drag terms. $C_a$, the updated expression for the atmospheric neutral drag coefficient, can be calculated in terms of contributions from specific spatial features of the sea ice cover:

$$C_a = C_a^{skin} + C_a^{f,rdg} + C_a^{f,floe} + C_a^{f,pond}. \tag{4}$$

$C_w$, the updated expression for the ocean neutral drag coefficient, can similarly be calculated as:

$$C_w = C_w^{skin} + C_w^{f,rdg} + C_w^{f,floe}. \tag{5}$$

Here $C^{skin}$ refers to the skin drag term, and $C^{f,rdg}$, $C^{f,floe}$ and $C^{f,pond}$ refer to form drag terms for ridges and keels, floe edges, and melt pond edges respectively. Tsamados et al. (2014) outline the following expression for $C_a^{f,floe}$ in the case of

surface momentum exchange over the sea ice-atmosphere interface with a reference height of 10 m:

$$C_a^{f,floe} = \frac{1}{2}\frac{c_{fa}}{\alpha_{shape}}S_c^2\frac{H_f}{L}A\left[\frac{\ln\left(\frac{H_f}{z_{0w}}\right)}{\ln\left(\frac{10}{z_{0w}}\right)}\right]^2.$$ (6)

Here $c_{fa}$ is a local form drag coefficient, taken to be constant. $\alpha_{shape}$ is a geometrical parameter to account for the shape of the floes. The ratio $\frac{c_{fa}}{\alpha_{shape}}$ takes the value 0.2. $L$ is the average floe diameter, which in Tsamados et al. (2014) is calculated as a function of sea ice concentration as per the parameterisation outlined in Lüpkes et al. (2012). $z_{0w}$ is the roughness length of water upstream of the floe, given by 3.27 x 10$^{-4}$ m (Hunke et al., 2015). $H_f$ is the freeboard of the floe i.e. the distance between the upper surface of the floe and the sea surface. To calculate $C_w^{f,floe}$, the form drag of sea ice floes at the sea ice-ocean interface, $H_f$ in Eq. (6) is replaced with $D$, the draft. $D$ is defined as the distance between the lower surface of the floe and the sea surface. $S_c$ is the sheltering function and is calculated as a function of sea ice area fraction, $A$, using an approximation from Lüpkes et al. (2012):

$$S_c = 1 - e^{-s_{lf}(1-A)}.$$ (7)

$s_{lf}$ is the floe sheltering attenuation coefficient, with $s_{lf} = 11$ as per Lüpkes et al. (2012).

**2.1.4 Modifications to standard CICE to incorporate FSD effects**

The CPOM-CICE setup has been adapted to represent the impact of the FSD on the sea ice cover via both the lateral melt rate and floe edge contribution to form drag. Eq. (1), used to calculate the fraction of sea ice area lost due to lateral melting, is modified to:

$$\frac{1}{A}\frac{dA}{dt} = \frac{\pi}{\alpha_{shape}l_{eff}}w_{lat.}$$ (8)

$L$, the constant floe diameter, has been replaced by $l_{eff}$, the effective floe size. $l_{eff}$ is the diameter of the set of identical floes that has the same total perimeter as a set of floes of variable size with the same total ice area (Bateson et al., 2020). $l_{eff}$ is applicable here because the lateral melt volume is proportional to the total floe perimeter. Eq. (6), the expression for the floe edge contribution to form drag at the sea ice-atmosphere interface, has also been modified:

$$C_a^{f,floe} = \frac{1}{2}\frac{c_{fa}}{\alpha_{shape}}S_c^2\frac{H_f}{l_{eff}}A\left[\frac{\ln\left(\frac{H_f}{z_{0w}}\right)}{\ln\left(\frac{10}{z_{0w}}\right)}\right]^2.$$ (9)

The equivalent expression at the sea ice-ocean interface is modified similarly. $l_{eff}$ is used here since it characterises the average floe length scale.

**2.2 The prognostic FSD model**

**2.2.1 Prognostic model overview**

The prognostic FSD model used here has been adapted from the version presented by Roach et al. (2018). At the core of the prognostic FSD model is the joint floe size-thickness probability distribution (FSTD), $f(r,h)drdh$. This describes the fraction of a grid cell covered by floes with a radius between $r$ and $r + dr$ and thickness between $h$ and $h + dh$. Processes that change floe size represented in this model include lateral melting and freezing, wave-induced break-up of floes, and welding together of floes. The model also allows the formation of new floes, complete melt out of existing floes, and advects the FSTD between grid cells. The parameterisation introduced in Roach et al. (2019) to determine the size of newly formed floes from the local wave conditions has also been included in the setup used here. A full description of the original prognostic floe size-thickness distribution model is presented in Roach et al. (2018) with details of the wave-dependent floe formation parameterisation available in Roach et al. (2019). Unlike Roach et al. (2019), we do not use a separate wave model coupled to CICE to calculate the necessary wave properties within the sea ice-covered grid cells for use with the wave-dependent floe formation parameterisation. Instead, we adapt the scheme used in Roach et al. (2018), which calculates in-ice wave properties using an

extrapolation approach from forcing external to the sea ice cover to calculate the necessary in-ice wave properties. We also introduce a novel treatment of brittle fracture to the prognostic model. This brittle fracture scheme is described in section 2.2.2. For the prognostic FSD model $l_{eff,n}$, the effective floe size for the $n^{th}$ sea ice thickness category, is calculated in terms of $L(r, h)$, the modified areal FSTD, where the integral of $L(r, n)$ over the range $r_{min}$ to $r_{max}$ is 1 i.e. the distribution is normalised per thickness category:

$$l_{eff,n} = \frac{2}{\int_{r_{min}}^{r_{max}} r^{-1} L(r,n) \, dr}. \tag{10}$$

A representative $l_{eff}$ for the full FSTD is then calculated as the area-weighted average of $l_{eff,n}$ across the thickness categories.

### 2.2.2 The brittle fracture scheme

It will be shown in section 4.1 that the prognostic model struggles to capture the shape of the observed FSD for mid-sized floes. Sensitivity studies show that it not possible to modify existing parameterisations in the prognostic FSD model to substantially improve model performance against observations (Bateson, 2021a). This suggests there are important processes currently not represented within the prognostic model. A leading candidate for the latter is brittle fracture and associated processes. Satellite imagery of the Arctic sea ice cover, especially over the winter pack ice, shows linear features such as leads and fractures referred to as slip lines or linear kinematic features existing at scales of kilometres (Kwok, 2001; Schulson, 2001). These linear features have been found to intersect at acute angles, from scales of millimetres to kilometres, creating individual diamond shaped regions and floes over the sea ice cover (Weiss, 2001; Schulson, 2004). The similarity of these linear features to fracture patterns formed in laboratory studies of the shear rupture mechanism, where a crack forms once a large enough shear stress is imposed, has been used to argue that the shear rupture mechanism is responsible for the linear features seen in the pack ice (Weiss and Schulson, 2009). A discrete element model of the sea ice incorporating compressive, tensile, and shear rupture failure mechanisms acting under wind stress has been shown to produce distributions of fractures that are comparable to the distribution of linear features seen in the Arctic pack ice (Wilchinsky et al., 2010). The existence of this brittle fracture behaviour in sea ice from microscopic to macroscopic scales makes it an interesting candidate to consider in terms of FSD evolution, though the scaling of brittle fracture remains an area of ongoing research (e.g. Weiss and Schulson, 2009; Hutchings et al., 2011; Weiss and Dansereau, 2017).

Clearly, brittle fracture events can have a direct impact on the size of larger floes and potentially also smaller floes. However, plausible indirect mechanisms also exist. Perovich et al. (2001) observed that summer floe break-up of sea ice in the central Arctic in 1998 was driven by thermodynamic weakening of cracks and refrozen leads in the sea ice cover during a period when the dynamic forcing and internal sea ice stress was expected to be small. More recent observational studies have also suggested a link between sea ice melt and floe break-up (Arntsen et al., 2015; Hwang et al., 2017). This suggests that linear features in the sea ice that form from the brittle fracture and subsequent refreezing of sea ice in winter can then influence sea ice break-up in summer as the sea ice thins and weakens. A full physically-derived parameterisation of the impacts of brittle fracture on the sea ice cover, both directly and indirectly via thermodynamic weakening, requires additional direct observations of these processes within the sea ice cover and is therefore beyond the scope of this study. It is nevertheless possible to use theoretical models of brittle fracture processes to explore the potential impact of brittle fracture related mechanisms on FSD shape. In a brittle fracture event cracks can propagate and, where they exceed a critical speed, become unstable and branch. Individual branches and fractures can also merge, with the lifetime of the fracture determining the size of the subsequent fragment that forms. The branching results in a hierarchical process, with several levels of branches forming from the same central fissure (Åstrom et al., 2004; Kekäläinen et al., 2007). Idealised models of brittle fracture show that the fragment size distribution adopts a power law with an exponent of -2 and an upper cut-off determined by an exponential in the square of the fragment size (Gherardi and Lagomarsino, 2015).

In order to investigate the potential impact of in-plane brittle fracture processes on the FSD, the prognostic model has been modified to include a quasi-restoring brittle fracture scheme, which applies a conditional restoring to the FSD towards the

theoretical distribution produced by idealised models of brittle fracture i.e. a power law with an exponent of -2. In this scheme brittle fracture can transfer sea ice area fraction from a larger floe size category to the adjacent smaller category. In addition, the following condition must be fulfilled:

$$\frac{\ln n_i - \ln n_{i-1}}{\ln d_i - \ln d_{i-1}} > -2. \tag{11}$$

Here $n$ and $d$ refer to the floe number density and diameter at the midpoint of category $i$ respectively. This condition means that the restoring scheme only applies where the slope between adjacent categories in log-log space is greater (more positive) than -2 i.e. only when the ratio of larger floes to smaller floes exceeds a given value. The sea ice area fraction transferred in a single timestep between two adjacent categories is $C_{bf}a_i$ where $a_i$ is the area fraction of the larger category. $C_{bf}$ the restoring constant, is calculated as:

$$C_{bf} = \frac{\tau}{\Delta t}. \tag{12}$$

Here $\tau$ is the restoring timescale, and $\Delta t$ is the model timestep. Figure 1 provides a visual summary of the quasi-restoring scheme. The motivation for this scheme is to impose a restoring tendency on the FSD to the predicted shape of the distribution if it were acting only under brittle fracture. The transfer of sea ice area fraction is only allowed in one direction from larger to adjacent smaller categories since floes cannot unfracture. This process is conservative in sea ice area i.e. the reduction in floe area in the larger category will be matched by an increase in floe area in the smaller category.

A value for the restoring timescale, $\tau$, needs to be determined. Both direct and indirect mechanisms have been discussed above describing how brittle fracture can impact the sea ice cover. Fracture events occur regularly through autumn, winter and spring within the pack ice to break up floes and form features such as leads, though these generally freeze up again. The result of these fracture events is to create a network of linear features that define weaker regions of ice interspersing stronger / thicker ice. Idealised models of brittle fracture suggest that the size distribution of the thicker regions of ice follow a power law with an exponent of -2. The linear features are then vulnerable to increased thinning and melting, increasing the likelihood of break-up along these features during late spring and summer as the sea ice retreats. This effectively 'releases' the floe size distribution defined during brittle fracture events outside of the melt season. It is this second mechanism that is of more relevance when considering the impacts of the FSD on the seasonal retreat of the Arctic sea ice. In this context, $\tau$, the restoring timescale, refers to the timescale for the sea ice to thin sufficiently such that the sea ice is vulnerable to in-plane fracture events along existing weaknesses. Sea ice thickness away from the ice edge at the start of the melt season is generally in the range of $1-3$ m. Vertical melt rates are of the order of $5-15$cm day$^{-1}$. Therefore, significant thinning can generally occur over timescales as short as a week up to a couple of months. For simplicity, $\tau$ is here set to 30 days.

There are clear limitations associated with this approach. The use of a fixed timescale makes it difficult to capture both the direct mechanism of brittle fracture impact on floe size, which dominates outside of the melt season, and the indirect mechanism via thermodynamic weakening, which is more important within the melt season. The latter mechanism has been prioritised in this case in determining the timescale given it is the FSD state in the melt season that is of primary importance for understanding FSD impacts on the Arctic sea ice (Bateson et al., 2020). Just considering the thermodynamic weakening mechanism, the use of a fixed timescale is still a simplification given the significant spatial and temporal variability of relevant factors to this mechanism such as melt rates, ice strength, and dynamic forcing. In addition, the brittle fracture approach assumes transfer of sea ice area fraction only between adjacent categories, whereas physically a larger floe can break down into floes of any smaller size. Nevertheless, this new scheme remains a useful way to approximate the impact of brittle fracture on FSD shape. Results will be presented in section 4.1 to demonstrate that the inclusion of this new brittle fracture scheme significantly improves prognostic FSD model performance against observations in simulating FSD shape for mid-sized floes.

**2.3 The WIPoFSD model**

The WIPoFSD model used in this study has been adapted from the version presented Bateson et al. (2020), which in turn was based on the coupled ocean–waves–in–ice model NEMO–CICE–WIM developed at the UK National Oceanography Centre

(NOC). The NEMO–CICE–WIM model approximates the shape of the FSD as a multiple-exponent truncated power law with coefficients depending on ice fraction; in this study we use a constant exponent as per Bateson et al. (2020). The WIPoFSD model fits the number-weighted FSD, $N(x)$, where $x$ is the floe diameter, to a power-law distribution:

$$N(x|d_{min} \leq x \leq l_{var}) = Cx^{\alpha}. \tag{13}$$

$N$ has units of reciprocal metres, all floe size variables have units of metres, and $\alpha$, the WIPoFSD model exponent, is dimensionless. $l_{var}$, the variable FSD tracer, evolves in each grid cell as a function of physical processes between the upper and lower floe diameter cut-offs, $d_{max}$ and $d_{min}$ respectively. Bateson et al. (2020) suggested that $l_{var}$ can be taken as representing the history of a given area of sea ice in terms of physical processes that affect the FSD. The model is initiated with $l_{var}$ set to $d_{max}$ in all grid cells where sea ice is present. $C$ is calculated such that the total floe area is equal to the total

sea ice area. The model parameterises the role of four processes in the evolution of the FSD: lateral melting, wave-induced break-up, winter growth, and advection. A full description of how these processes are represented within the WIPoFSD model, including a description of the advection scheme for waves in sea ice, is available in Bateson et al. (2020); a summary has also been provided here in Appendix A.

The version of the WIPoFSD model used here includes a modified lateral melting scheme to that presented in Bateson et al.,

(2020). In the WIPoFSD model, processes are parameterised in terms of how they impact $l_{var}$ and useful properties such as $l_{eff}$ can easily be calculated from $l_{var}$. The appeal of this approach is that it is both simple and enables an exploration of the broader impacts of a power-law distribution on the sea ice cover whilst retaining spatial and temporal variability in $l_{eff}$. For mechanical processes such as wave break-up, the use of $l_{var}$ is particularly suitable. Wave break-up acts to reduce the number of larger floes and increase the number of smaller floes; $l_{var}$ effectively marks the boundary between these two contrasting

effects. However, it is not possible to define two clear regimes of how floe size changes in response to lateral melting; instead, floes across the distribution reduce in diameter by the same magnitude in response to a lateral melting event. Here, we have modified the lateral melting scheme to calculate the change in $l_{eff}$ rather than $l_{var}$, since it is possible to calculate exactly how much the total floe perimeter, and therefore $l_{eff}$, would increase or decrease in response to any change in the FSD. Whilst it is not possible to exactly capture cumulative changes to the FSD over several timesteps due to the constraints of having a fixed

exponent power-law distribution with a lower floe size limit, it is possible to calculate exactly how the effective floe size, $l_{eff}$, will change over one timestep by integrating across the updated FSD after a lateral melting event. The updated effective floe size, $l_{eff,new}$, can then be calculated as follows:

$$l_{eff,new} = \frac{[l_{var}^{3+\alpha} - d_{min}^{3+\alpha}]A_{new}}{(3+\alpha)A_{old}}\left(\frac{[l_{var}^{2+\alpha} - d_{min}^{2+\alpha}]}{(2+\alpha)} - \frac{2\Delta l[l_{var}^{1+\alpha} - d_{min}^{1+\alpha}]}{(1+\alpha)}\right)^{-1}. \tag{14}$$

Here $\Delta l$ is the length of lateral melting experienced by each floe edge and $A_{old}$ and $A_{new}$ refer to the sea ice area fraction

before and after the lateral melting event, respectively. $l_{var,new}$ can then be calculated from $l_{eff,new}$ using a Newton-Raphson iterative scheme. A full derivation of Eq. (14) and description of the iterative scheme can be found in Appendix B.

**3 Methodology**

**3.1 Sea ice simulations**

All simulations in this study are initiated with an ice-free Arctic on 1st January 1980 and evaluated over a 37-year period until

31st December 2016. The first ten years of these simulations are taken as spin-up, with timeseries presented over the period 1990-2016. Averages are calculated over the period 2000-2016, taken as representative of the current climatology. The CPOM version of CICE is run over a pan-Arctic domain with a 1° tripolar (129×104) grid. Sections of the Hudson Bay and Canadian Arctic Archipelago are not included within the model domain. Surface forcing is obtained from 6-hourly NCEP-2 reanalysis fields (Kanamitsu et al., 2002). The mixed-layer properties are restored over a timescale of 5 days to a monthly climatology

reanalysis at 10 m depth taken from the MyOcean global ocean physical reanalysis product (MYO reanalysis; Ferry et al., 2011). The deep ocean is restored after detrainment from the mixed layer over a timescale of 90 d to the winter climatology

(for this we take the mean conditions on 1 January from 1993 to 2010) from the MYO reanalysis. The minimum mixed-layer depth is set to 10 m. $H_s$, the significant wave height (m), and $T_p$ the peak wave period (s), of ocean surface wave fields are obtained from the ERA-Interim reanalysis dataset (Dee et al., 2011). The forcings are updated at 6 h intervals, but only for locations where the sea ice is at less than 1 % sea ice concentration. The ERA-Interim reanalysis has been selected for the ocean surface wave field forcing as this dataset has generally been found to perform well in comparison to other reanalyses against observations of wind speed and wind speed profile in the Arctic during summer months (e.g. Jacobson et al., 2012; de Boer et al., 2014).

The general CPOM-CICE setup used for all simulations in this study has several further differences to CICE version 5.1.2, in addition to those described in section 2, based on recent work by Schröder et al. (2019). The maximum meltwater added to melt ponds is reduced from 100 % to 50 %. This produces a more realistic distribution of melt ponds (Rösel et al., 2012). Snow erosion, to account for a redistribution of snow based on wind fields, snow density, and surface topography, is parameterised based on Lecomte et al. (2015) with the additional assumptions described by Schröder et al. (2019). The "bubbly" conductivity formulation of Pringle et al. (2007) is also included, which results in larger thermal conductivities for cooler ice. The longwave emissivity is increased from 0.95 to 0.976. The following parameters are modified from the default values used in Tsamados et al. (2014): atmospheric background drag coefficient, ocean background drag coefficient, ridge impact parameter, and keel impact parameter of the form drag parameterisation. They are set to 0.001, 0.0005, 0.1 and 0.5 respectively. Schröder et al. (2019) discuss how these changes increase ice drift over level ice and reduce it over ridged ice leading to more realistic ice drift patterns.

A total of five different simulations are used here; a summary of these simulations is presented in Table 2. The reference simulation, *ref*, sets $l_{eff}$ to a fixed value of 300 m. The prognostic FSD setup, *prog-best*, uses the standard 12 floe size categories outlined in Roach et al. (2018) and the 5 standard CICE thickness categories (Hunke et al., 2015) and includes the brittle fracture scheme described in section 2.2.3. Apart from the modifications outlined in section 2.2.2 – 2.2.4, prognostic model setup and parameter choices are identical to Roach et al. (2018). The simulation with the WIPoFSD model, *WIPo-best*, uses an identical setup to that in Bateson et al. (2020), but now incorporating the updated lateral melting scheme described in section 2.3. For the WIPoFSD model parameters, $d_{max}$ is set to the standard value used by Bateson et al. (2020) of 30 km. The minimum floe size that can be resolved in *prog-best* is 5.375 m and hence $d_{min}$ will be set to the same value. $\alpha$ is set to -2.56, the average value across the three locations represented in the novel FSD observations that will be discussed in section 3.2. This means that parameter or model choices for both *WIPo-best* and *prog-best* have been selected to produce a best fit to the same set of observations. Sensitivity studies to these parameter choices have previously been performed for the WIPoFSD model and the version of the prognostic FSD model considered here (i.e. including the brittle fracture scheme) in Bateson et al. (2020) and Bateson (2021a) respectively. Two additional simulations are performed using the prognostic model to compare against observations of the FSD. These two simulations will include a total of 16 floe size categories using Gaussian spacing rather than the standard 12. The prognostic model produces an unphysical increase or 'uptick' in the largest few categories, at least partially a result of having a fixed maximum floe size in the model (Roach et al., 2018). By using 16 floe size categories rather than 12, the largest 4 floe size categories that include this 'uptick' will fall outside the range of floe sizes included in the comparison to observations. The first of these additional prognostic simulations, *prog-16*, is otherwise identical to *prog-best*. The second, *prog-16-nobf*, excludes the brittle fracture scheme described in section 2.2.3. Whilst 16 floe size categories could also be retained for the *prog-best* simulation, the increase in model run time increases non-linearly with increasing number of categories (e.g. the *prog-16* simulation takes around 60% longer to run than the *prog-best* simulation). In addition, the improved resolution of the shape of the distribution for floes of a size of 1 km or larger is not significant when considering the impact of an FSD on sea ice via the floe edge contribution to form drag and lateral melt rate, which both scale to the inverse of floe size. Therefore, 12 floe size categories represents a more practical choice for the prognostic FSD model, particularly for use in climate models where a high computational cost would be prohibitive to the introduction of new physics. Figure 2

presents an example of the model output from *prog-16-nobf*, showing the perimeter density distribution within the MIZ for April, June, and August, averaged over 2000-2016. Figure 2 demonstrates how the 'uptick' is confined to the largest 3-4 floe size categories.

## 3.2 FSD observations

To assess the performance of the two alternative FSD models, we consider a new observational dataset that has not been used to motivate the development of either FSD model. The observations consist of 41 separate images over the period 2000-2014, covering May – July, and collected from three regions: the Chukchi Sea (70 N, 170 W); the East Siberian Sea (82 N, 150 E); and the Fram Strait (84.9 N, 0.5 E). The raw floe size data has been retrieved using the algorithm described in Hwang et al. (2017) from the GFL HRVI (Global Fiducials Library high-resolution visible-band image) imagery that has been declassified by the MEDEA group (Kwok and Untersteiner, 2011). This has been made available publicly as LIDPs (Literal Image Derived Products) at 1 m resolution (available at http://gfl.usgs.gov/). The total image size varies between observations, but generally has length dimensions of 5 – 20 km. The resolution of the imagery was reduced from 1 m to 2 m prior to processing by the algorithm.

The first step of processing the raw floe size data, consisting of a list of individual floe sizes, is to sort them into the Gaussian-distributed floe size categories used within the prognostic model for ease of comparison. Any floes that exceed the upper diameter cut-off of the largest category, 1892 m, will be discarded from the analysis. This step is necessary because the two models simulate the full FSD, and not individual floes. Floes large compared to the image size are inadequately sampled in observations to construct the full FSD. For example, the presence of a single large floe, comparable to the image size, can cause a large perturbation across the distribution reported for that location. Instead, only floe size categories that are small enough to consistently be populated by multiple floes across all sampled images are retained. A lower floe diameter cut-off of 104.8 m is also applied to this analysis, taken to be the smallest floe size that can be reliably resolved from the observations for the methodology and resolution used. The limiting factor on the smallest resolved floe size is the ability to resolve gaps between floes. Once floes outside the range of 104.8 m to 1892 m in diameter have been discarded, the total area of remaining floes is calculated and taken to be the total sea ice area for normalising the reported perimeter density (perimeter per unit sea ice area). The average normalised perimeter density for each floe size category is then reported at the mid-point of that category. The floe perimeter density distribution, $\rho_{FSD}(x)$, is considered in this study rather than the floe number or area distribution since it is the perimeter that has been identified as most relevant to the impact of the FSD on sea ice when considering lateral melt (Bateson et al., 2020). It is defined here as:

$$P_{FSD} = \int_{x_{min}}^{x_{max}} \rho_{FSD}(x) \, dx \tag{15}$$

Here $x_{min}$ and $x_{max}$ refer to the minimum and maximum floe diameters respectively within an FSD. $P_{FSD}$ is the perimeter density across the whole distribution, calculated as the total perimeter divided by the total area of all floes in the distribution. The concept of perimeter density has been used previously to this study e.g. Roach et al. (2019), Bateson et al. (2020).

To compare model output to the observations of the FSD, two sample years will be selected for each location: Chukchi Sea, May – June 2006 (4 LIDPs), May 2014 (4 LIDPs); East Siberian Sea, June 2001 (3 LIDPs), June – July 2013 (2 LIDPs); Fram Strait, June 2001 (6 LIDPs), June 2013 (2 LIDPs). These specific years have been selected as they all include at least two separate LIDP-derived floe size observations. Perimeter density distributions from the prognostic model are reported as an average over one or two months for the relevant region. The months selected for this average are chosen to minimise the difference between the mean day of collection for observations and median day of the model output. Figure 3 shows the specific areas over which the FSD is averaged. Each case study area consists of a set of 5 x 5 grid cells that includes the location where the observations were drawn from.

## 3.3 Observations of sea ice extent and volume

We use sea ice concentration products from the Bootstrap algorithm version 3 (Comiso, 1999) and NASA Team algorithm

version 1 (Cavalieri et al., 1996). Both datasets have a spatial resolution of 25 km x 25 km and a temporal resolution of 1 day. Comparisons will focus on the pan-Arctic extent rather than the spatial distribution of sea ice concentration due to the high uncertainty in summer and MIZ of satellite-derived concentration products (Meier and Notz, 2010). We use the sea ice volume product from PIOMAS, the Pan-Arctic Ice Ocean Modelling and Assimilation System (Zhang and Rothrock, 2003). Whilst the PIOMAS volume product is a reanalysis and does not incorporate direct observations of the sea ice thickness, it has been evaluated using available observations of sea ice thickness (e.g. Schweiger et al., 2011). This product is often used to test model performance in simulating the total Arctic sea ice volume (Schröder et al., 2019) due to the challenges in estimating sea ice thickness from radar altimetry and limited availability of in-situ thickness measurements (e.g. Massonnet et al., 2012; Stroeve et al., 2012; Ridley et al., 2018).

## 4. Results

### 4.1 A comparison to observations of the FSD

The observations described in section 3.2 can be used to test how well both the prognostic FSD model and a power-law fit using the same exponent across all case studies captures the shape of the observed FSD for mid-sized floes. Figure 4 compares FSD observations, a power-law fit, and prognostic model output from both the *prog-16* (with brittle fracture) and *prog-16-nobf* (without brittle fracture) simulations for each selected case study region and time period described in section 3.2. The power-law fit is used here to represent *WIPo-best*, since the WIPoFSD model is built on the assumption that the FSD can be approximated by a truncated power-law with a singular time-invariant exponent. In practice, the emergent FSD from *WIPo-best* will be identical to the power-law fit shown in Fig. 4 provided the floe size range included is consistently below $l_{var}$, which is the case for all the case studies considered. When comparing observations across the sites considered in Fig. 4 there is clear variability between the different case studies, but this variability is substantially smaller than the differences between the prognostic model without brittle fracture and the observations across all the case studies considered. It cannot be expected that an FSD model can precisely replicate an observed FSD given other differences will exist between the model and the observed sea ice state such as ice thickness and concentration, but it can be expected that a simulated FSD should be within the variability in FSD shown by observations if an FSD model is accurately capturing the relevant processes. Figure 4 therefore suggests that *prog-16-nobf* performs poorly in capturing the behaviour of the FSD for mid-sized floes. The perimeter density distribution predicted by the *prog-16-nobf* for each category is in general multiple orders of magnitude from the observed value. In particular, the slope of the distribution is much steeper (more negative) for the model output than observations i.e. the model predicts smaller floes within the range 104.8 m - 1892 m take up a much larger proportion of the total sea ice area than is suggested by observations. Figure 4 also shows that *prog-16* significantly improves the shape of the emergent perimeter density distribution for the floe size range considered compared to *prog-16-nobf*. The updated model performs particularly well in the East Siberian Sea and Fram Strait (panels c-f in Fig. 4) but less well in the Chukchi Sea (panels a-b in Fig. 4). Overall, the inclusion of the quasi-restoring brittle fracture scheme represents a significant improvement in the ability of the prognostic model to capture the shape of the FSD for mid-sized floes over the period May – July. The purpose of this comparison against floe size observations is to ensure that the WIPoFSD and prognostic model setups used in this study perform comparably well to the same dataset. There are limitations to this evaluation of model performance, however. In particular, floes smaller than 100 m or larger than 2 km are not considered for the reasons outlined in section 3.2, and the former are especially significant for determining the impact of a given FSD on sea ice concentration and thickness.

Whilst in Fig. 4 we consider prog-16 with 16 floe size categories, for comparisons against *WIPo-best* on sea ice behaviour within CICE we consider *prog-best* with 12 floe-size categories since this represents a more practical setup of the prognostic FSD model for use in sea ice and climate simulations, as discussed in section 3.1. In a comparison of model output from *prog-16* and *prog-best* (not presented here) larger differences can be seen in the shape of the distributions in the larger floe size categories due to the presence of the 'uptick' but these differences tend towards negligible in the smallest categories i.e. those most relevant in determining FSD impact on the sea ice cover (e.g. Tsamados et al., 2015; Bateson et al., 2020).

It is worth commenting briefly on how the brittle fracture scheme can improve model performance compared to observations, given it is a counterintuitive result that increasing floe break-up would produce a shallower slope in perimeter density. As discussed in section 3.2, the largest floe size categories in the prognostic model are excluded from the comparison to observations to exclude the non-physical 'uptick' that forms (Fig. 2). Whilst a reduction in ice area fraction in the largest

category and an increase in the smallest category can be expected, the change in ice area fraction in the remaining categories depends precisely on the balance between ice area fraction lost from that category (sink) and ice area fraction gained from the adjacent larger category (source). In this case, the presence of the 'uptick' shown in Fig. 2 for the prognostic model without brittle fracture results in the source of floe area being larger than the sink for most floe size categories and a net reduction in gradient overall from including brittle fracture.

**4.2 A comparison to observations of sea ice extent and volume**

In this section the CICE simulations *prog-best* and *WIPo-best*, summarised in Table 2, will be compared against observations or reanalysis of the sea ice cover. Both of these simulations have been optimised against the FSD observations presented in section 4.1. The *prog-best* simulation includes the brittle fracture scheme described in section 2.2.2 and the exponent selected for the *WIPo-best* simulation is the average fitted exponent across the FSD observations. These two simulations, alongside the

*ref* simulation that applies a constant floe size, will be tested against observations by considering the following metrics: the performance of the simulations in capturing the annual and interannual variability of the sea ice extent and volume; the performance of the simulations in capturing interannual trends in the sea ice extent and volume; and whether the inclusion of either FSD model reduces known model bias in the sea ice area fraction.

Figure 5 shows timeseries for the total Arctic sea ice extent and volume in March and September over 1990 – 2016 for *ref*,

*WIPo-best*, *prog-best* alongside observations (extent) or reanalysis (volume) over the same period. The differences between the simulations are significantly smaller than the difference between *ref* and the observations / reanalysis in both March and September. This plot shows that the inclusion of either FSD model does not produce an improvement in the ability of CICE to simulate pan-Arctic sea ice extent and volume or the variability in these metrics, with the size of any changes well within observational uncertainty, though this conclusion does not necessarily extend to climate simulations where FSD impacts on

sea ice feedbacks with the ocean or atmosphere could produce larger changes to the sea ice state.

Previous studies, e.g. Bateson et al. (2020) and Roach et al. (2018), show that the largest impacts of including an FSD model occur within the MIZ. Figure 6 compares timeseries from 1990 – 2016 for the MIZ and pack ice extent in both March and September for each of *ref*, *prog-best*, and *WIPo-best* in addition to the satellite-derived observations. Figure 6 shows that all three simulations simulate both the MIZ and pack ice extent within observational uncertainty, with any differences between

the simulations significantly smaller than the observational uncertainty, though this is partly due to the large differences in MIZ and pack ice extent between the two observational products. The simulations are unable to replicate a negative trend in March pack extent shown by the observations. *prog-best* produces both a higher MIZ extent and variability on average in March compared to *ref* but a lower extent in September. In comparison, *WIPo-best* shows a reduced MIZ extent throughout the year compared to *ref*. In September, all three simulations produce very similar pack extents, but in March there is a moderate

reduction for *prog-best* and a small reduction for *WIPo-best* relative to *ref*. Overall, inclusion of FSD processes within CICE results in changes to extent metrics of order $1 \times 10^5 \, km^2$. This corresponds to a percentage change in extent varying between around 1 % to 10 % across the different months and regions considered.

**4.3 A comparison of the two FSD models**

In this section, the *prog-best* and *WIPo-best* simulations will be compared directly, considering several metrics including total

sea ice extent and volume, and spatial difference plots for area fraction, thickness, and $l_{eff}$. The aim of this comparison is to understand the differences in the impacts of the two alternative FSD models and how these differences emerge.

**4.3.1 Pan-Arctic extent and volume**

Figure 7 shows the percentage difference in the sea ice extent and volume for both *prog-best* and *WIPo-best* relative to *ref*

averaged over 2000 to 2016, indicating the impact of each FSD scheme compared to assuming a constant floe size. The shaded region in Fig. 7 covers twice the standard deviation from the mean in each direction. The prognostic model produces a mean reduction in sea ice extent of just under 2% in June, compared to less than a 1% reduction with the WIPoFSD model; this reduction is just under 2% for both models in August. The average reduction in sea ice volume in September is 2.5% and 4% for *WIPo-best* and *prog*-best respectively relative to *ref*. The minimum reduction for the prognostic model is 1.5% in the spring months, compared to just 0.5% with the WIPoFSD model. The prognostic model also shows a larger interannual variability compared to the WIPoFSD model.

Figure 7 considers differences in the mean behaviour only, however the mean behaviour may obscure important trends. Figure 8 shows the percentage difference in total Arctic sea ice extent and volume for both *prog-best* and *WIPo-best* compared to *ref* from 1990 – 2016 in March and September. The differences are consistent with Fig. 7, with *prog-best* generally showing larger reductions than *WIPo-best* relative to *ref* other than for September extent. There is a possible positive trend for the difference in the March sea ice extent and a negative trend for the September sea ice extent for both *prog-best* and *WIPo-best* relative to *ref*, but this is inconclusive due to high interannual variability relative to the strength of the trend. More robust trends can be seen in the sea ice volume e.g. *prog-best* produces an average reduction in September volume of 2% compared to *ref* in the 1990s increasing to about a 5% reduction in the 2010s. A similar but weaker trend can be seen for *WIPo-best* relative to *ref*. The reduction in the March volume changes from about 1.1% in the 1990s to about 1.5% in the 2010s for *prog-best* relative to *ref*, whereas there is no evidence of any trend for *WIPo-best* relative to *ref*. Figure 8 shows that the interannual variability shown in Fig. 7 can be partly explained by long term trends, particularly for *prog-best* relative to *ref*.

### 4.3.2 Sea ice melt components

In *WIPo-best* and *prog-best*, floe size impacts the sea ice via two model components: form drag and lateral melt volume. Several previous studies, including Tsamados et al. (2015) and Roach et al. (2018), found that increases in the lateral melt volume resulting from higher floe perimeter were compensated by a reduction in the basal melt. Bateson et al. (2020) demonstrated that this compensation effect was shown to primarily be a result of the physical reduction of sea ice area in locations of high basal melt. Figure 9 explores whether the same basal melt compensation effect is produced by the prognostic model. Figure 9 shows annual timeseries of the difference in the cumulative top, basal, lateral, and total melt for both *prog-best* and *WIPo-best* relative to *ref* averaged over 2000 – 2016. For both models a significant increase in lateral melt is compensated by a reduction in basal melt of similar magnitude, leading to only a small net increase in total melt. Whilst the increase in the lateral melt for *prog-best* is higher than for *WIPo-best*, both show an increase in the total melt of a small and similar magnitude. This suggests that any feedbacks on the total melt resulting from the decrease in area from enhanced lateral melt, such as the albedo feedback, are weak even for the *prog-best* simulation. The similar magnitude of change in the total melt also means that the results shown in Figs 7 and 8, where the sea ice volume is lower in both September and March for *prog-best* compared to *WIPo-best*, are unlikely to be driven by an increase in the total melt. This point will be discussed further in section 5.2. Also shown in Fig. 9 is the difference in melt components for *prog-best* compared to *WIPo-best*. The difference in cumulative total melt peaks in July and then decreases and switches sign. This is consistent with Fig. 7, where *prog-best* shows a stronger reduction in sea ice extent in the early melt season compared to *WIPo-best*, but the reduction in extent in August is comparable.

### 4.3.3 Spatial distribution of ice area fraction, thickness, and effective floe size

Previous studies e.g. Bateson et al. (2020) and Roach et al. (2018), have shown large FSD model impacts locally even where pan-Arctic impacts are small. Figure 10 shows maps of differences in sea ice area fraction and thickness for both *prog-best* and *WIPo-best* relative to *ref*, and spatial distribution plots of $l_{eff}$ for both *prog-best* and *WIPo-best*. Results are presented for March, June, and September. The spatial pattern of the reduction in area fraction is similar for both *prog-best* and *WIPo-best* relative to *ref* but the magnitude is larger for the former in the early melt season. The region where $l_{eff}$ drops significantly below 280 m in *WIPo-best* is generally confined to the outer MIZ. For the prognostic model, $l_{eff}$ is generally well under 100

m across the MIZ. The distribution in $l_{eff}$ corresponds to the regions of largest reduction in sea ice area fraction for *prog-best* relative to *ref* in the early melt season. The reduction in sea ice area fraction in the September MIZ is comparable in magnitude for both *prog-best* and *WIPo-best*, which is consistent with the results presented in Figs 7 and 9. For *prog-best*, $l_{eff}$ is shown to increase within the MIZ over the course of the melt season from March to September, which can explain the different results in the early and late melt season. *prog-best* shows an increase in the sea ice area fraction across much of the pack ice in September, with a particularly strong response in the central Beaufort Sea. This response is not seen with *WIPo-best* because the maximum $l_{eff}$ is 300 m for the selected model parameters i.e. the same as the fixed floe size in *ref*, whereas for *prog-best* it can be as high as 1700 m. For *prog-best* relative to *ref*, reductions in sea ice thickness persist through March and June across the central Arctic, but for *WIPo-best* differences only persist in locations that become marginal for at least some of the year and along the Canadian Archipelago. In September, the reduction in thickness spans the full Arctic for *prog-best*, whereas differences are mostly confined to the outer MIZ for *WIPo-best*.

Figure 10 shows much higher spatial variability in $l_{eff}$ for *prog-best* compared to *WIPo-best*. A good case study is the relatively low $l_{eff}$ seen in the Chukchi Sea during March and June. The floe formation mechanism is important to FSD evolution in this region of the Arctic since it experiences ice-free conditions for at least part of the year. Higher wave activity is also expected in this region due to an increased fetch via the Bering Strait, and this will increase the proportion of floes that form in smaller floe size categories. Other regions that experience ice-free conditions are generally more sheltered from wave activity due to adjacency to continental land mass. The only comparable regions in terms of wave exposure are the Greenland Sea and Barents Sea, where lower values of $l_{eff}$ can also be seen. Further analysis (Bateson, 2021a) indicates more generally that the high spatial variability in $l_{eff}$ for the prognostic model cannot easily be attributed to a single process but is particularly sensitive to the floe formation mechanism, brittle fracture scheme, and welding, all processes not explicitly represented in the WIPoFSD model. The processes included in the WIPoFSD model, such as wave break-up of floes and lateral melt, are not found to have a large impact on the spatial distribution of $l_{eff}$ within the prognostic model. One point of interest here is the regions of reduced $l_{eff}$ shown for the WIPoFSD model appear to correspond well with the MIZ defined using wave activity presented in Horvat et al. (2020), which suggests that a possible application of FSD models would be an alternative way of defining the MIZ compared to the sea ice concentration-derived definition.

### 4.3.4 Standard deviation of sea ice area fraction, thickness, and effective floe size

Figure 10 is useful to understand the spatial distribution of the pan-Arctic changes in sea ice state shown in Fig. 7, but the inclusion of an FSD model may not only act to change the mean state of the sea ice but also the interannual variability. Furthermore, a small change in mean sea ice state may disguise a much larger change in sea ice variability. Figure 11 shows the standard deviation in sea ice area fraction and thickness for *ref*, the difference in standard deviation for area fraction, thickness, and $l_{eff}$ for *prog-best* and *WIPo-best* relative to *ref*. Changes to the standard deviation in area fraction have a low magnitude and are isolated rather than part of a more systematic change in behaviour. For the standard deviation in thickness, differences are again small and isolated in March, but larger changes can be seen within the September MIZ of up to $10 - 20\%$. These changes in thickness variability correspond to where the largest differences in sea ice thickness can be seen in Fig. 10 and are consistent with the high interannual variability of the reduction in sea ice volume suggested in Fig. 8. For *WIPo-best*, variability in $l_{eff}$ is generally only seen within the MIZ as $l_{eff}$ remains close to the maximum value within the pack ice. For *prog-best*, the standard deviation in $l_{eff}$ broadly correlates to the magnitude of $l_{eff}$ seen in Fig. 10. High interannual variability can be seen across the pack ice in both March and September for *prog-best*, suggesting that all locations experience some differences in the contributing processes to the emergent FSD year on year. These distinct patterns in interannual variability of $l_{eff}$ for *prog-best* and *WIPo-best* may therefore be a useful metric to measure in the Arctic to discriminate between the different approaches to modelling the FSD.

### 5. Discussion

## 5.1 Inclusion of brittle fracture in FSD models

Observations of the sea ice cover suggest brittle fracture processes have a role in the evolution of the FSD via both the direct impact of a fracture event on floe size and indirectly via thermodynamic weakening and subsequent break-up along weaknesses in the sea ice cover resulting from prior fracture events that subsequently froze up (e.g. Perovich et al., 2001, Arntsen et al., 2015; Hwang et al., 2017). This motivated the inclusion of a scheme to approximate the impact of brittle fracture processes on the FSD within the prognostic model based on idealised models of brittle fracture.

The inclusion of the quasi-restoring brittle fracture scheme into the prognostic FSD model significantly improved the simulated shape of the FSD for mid-sized floes of 100 m – 2000 m (Fig. 4). This was particularly true when comparing model output to observations in the Fram Strait and East Siberian Sea (panels c-f in Fig. 4), however, improvements were less impressive for the Chukchi Sea site (panels a-b in Fig. 4). To understand the difference in model performance at these sites, consider their locations shown in Fig. 3. Brittle fracture acts on the FSD at two of the three sites throughout the year, but only for part of the year for the Chukchi Sea since it fully transitions to an ice-free state over the melting season, unlike the other sites. The restoring timescale for the brittle fracture scheme, $\tau$, only determines the timescale for two neighbouring floe size categories to reach an equilibrium state rather than the entire distribution. The prognostic FSD model consists of 16 categories for the simulations considered in Fig. 3 and it would take several months for a starting state with all sea ice area in the largest floe size category to reach equilibrium across all floe size categories, a long enough lag to explain the different prognostic model performance for the Chukchi Sea. A physical interpretation for $\tau$ has previously been given as the timescale for sea ice to thin sufficiently that the sea ice is vulnerable to in-plane fracture events along existing weaknesses. It was discussed in section 2.2.2 that assuming a fixed value for $\tau$ is a significant approximation given the significant spatial and temporal variability of dependent factors such as sea ice thickness and melt rates. The difference in performance between sites considered in this study can therefore be considered a result of this approximation.

In order to understand the implications of the results presented in Fig. 4 for prognostic floe size modelling it is useful to consider why the brittle fracture scheme is able to improve model performance against observations. Considering the distributions presented in Fig. 4 for the standard prognostic model without brittle fracture, it does not obviously follow that a redistribution of sea ice area from larger categories to smaller categories would improve the shape of the distribution compared to the observations given the gradient is already too negative. However, the largest floe size categories in the prognostic model are excluded from the comparison to observations to exclude the non-physical 'uptick' that forms, as demonstrated in Fig. 2. The inclusion of brittle fracture acts to reduce the size of the uptick and redistributes sea ice area to mid-sized floe categories. There are two plausible factors that can produce this uptick: the truncation of the maximum possible floe size such that sea ice area accumulates in the largest category that would otherwise be distributed over several larger categories; and missing floe fragmentation processes in the prognostic model. Observations show that floes can exceed a size of 10 km (Stern et al., 2018 a), providing evidence that the former factor contributes to the uptick, but the results presented in Fig. 4 suggest that missing floe fragmentation processes also contribute, with brittle fracture related mechanisms being a leading candidate.

We do not suggest that our brittle fracture scheme is taken to be the final solution to representing brittle fracture processes in the prognostic model. The current scheme makes several significant simplifications including only allowing area transfer between adjacent categories and using a fixed restoring timescale. However, sensitivity studies show that the brittle fracture scheme does not dominate the shape of the emergent FSD and other processes continue to be important in the evolution of the FSD, particularly winter growth processes such as floe formation and welding (Bateson, 2021a). Despite the limitations with the quasi-restoring brittle fracture scheme, the results shown in Fig. 4 strongly suggest that brittle fracture or related fragmentation processes are required to capture the shape of the FSD for mid-sized floes, motivating the need to develop a physically derived brittle fracture parameterisation for FSD models.

An interesting comparison can be made between the treatment of brittle fracture within the prognostic model presented here and recent developments introducing the concept of 'damage' to the treatment of rheology within sea ice models (Dansereau

et al., 2016). One such sea ice rheology, named the Maxwell-elasto-brittle (Maxwell-EB) rheology, has been applied within the continuous and fully Lagrangian sea ice model neXtSIM (Rampal et al., 2019). This new rheology retains a 'memory' of any fracture events, effectively tracking how 'damaged' the sea ice cover is, and modifies the sea ice properties accordingly. This concept of 'damage' has clear parallels with the mechanism discussed above of how winter in-plane brittle fracture events can determine how the sea ice breaks-up in summer and may therefore present a useful basis for the development of a full parametrisation of brittle fracture processes for use in FSD models. Boutin et al. (2021) also demonstrated that the Maxwell-EB rheology can be combined with an FSD model in order to explore how wave break-up of floes can impact sea ice dynamics, highlighting an application of floe size modelling not considered in this study.

**5.2 Differences in the impacts of the FSD models and how they emerge**

Focusing first on a pan-Arctic scale, Figs 5-6 do not provide any evidence that the inclusion of an FSD model improves the performance of CICE in simulating the aggregated Arctic sea ice extent and volume behaviours and trends against observations or reanalyses. This is an important result for climate modellers since it assuages concerns that the FSD represents a source of structural uncertainty in climate models, though this conclusion needs to be confirmed using fully coupled climate simulations. This also does not preclude either FSD model from being an important improvement to sea ice models; these improvements may be at a regional scale rather than at a pan-Arctic scale. The accuracy of sea ice concentration measured using passive microwave data can be as low as ± 20% in summer or the MIZ (Meier and Notz, 2010). Measurements in sea ice thickness from radar altimetry can also have high uncertainty, with snow depth and density being the primary source of error (Tilling et al., 2018). It is therefore non-trivial to obtain high accuracy observations of the spatial distribution of sea ice concentration and thickness, and the use of the latter to validate models requires a careful use of case studies such as demonstrated by Schröder et al. (2019). Nevertheless, significant biases have been identified in coupled climate models in simulating the sea ice concentration (Ivanova et al., 2016) and CICE, in particular, has been shown to overpredict the sea ice concentration at the sea ice edge and underpredict the concentration within the pack ice (Schröder et al., 2019). In Bateson et al. (2020), the WIPoFSD model was found to provide a limited correction to this model bias. Similarly, Fig. 9 shows that the prognostic model produces a stronger correction to this model bias, driving reductions in sea ice area fraction in the MIZ and small increases in area fraction in the pack ice.

Figures 7-10, described in section 4.3, all highlight a key difference in the impact of the two FSD models. *prog-best* produces a stronger reduction in sea ice area fraction relative to *ref* in the early melt season but a more comparable reduction by August compared to *WIPo-best*. This difference can be attributed to the different treatment of floe formation and growth processes between the two models. The WIPoFSD model uses a simple restoring approach that operates over a short timescale of 10 days, which is applied during conditions of freezing. This means that over winter $l_{eff}$ will be at or close to its maximum value uniformly across the sea ice, except in locations at the outer sea ice edge that are exposed to wave break-up, as shown in Fig. 10. In comparison, the prognostic model aims to represent physically floe formation and growth processes such that the homogeneity produced after the freeze-up season with the WIPoFSD model is not seen with the prognostic model in Fig. 10. In the early melt season $l_{eff}$ is particularly low across the MIZ due to wave activity in this region causing existing floes to fragment and new floes to form in the smallest size category. As the melting season proceeds, floes in the smallest floe size categories preferentially lose surface area and melt out in response to lateral melting, a behaviour that is also visible in Fig. 10 e.g. $l_{eff}$ increases in the Fram Strait between March and June. This behaviour is not possible with the WIPoFSD model since it has a fixed exponent and minimum floe size. These results show the value of $l_{eff}$ in being able to characterise and understand how the inclusion of either FSD model impacts the sea ice cover and in understanding how differences in these impacts emerge. These results also show the potential limitations of using a simplified FSD model such as the WIPoFSD model; even though a power law might in general be a good fit to the FSD over the melt season, there could still be important mechanisms and features of FSD impacts that it fails to properly capture.

In section 4.3.2 it was noted that the total cumulative melt is slightly higher for *WIPo-best* compared to *prog-best* despite the

larger reduction in volume for *prog-best* compared to *WIPo-best* relative to *ref* shown in Figs 7 and 8. Figure 9 also shows the lateral to basal melt ratio is higher for *prog-best* compared to *WIPo-best*; the change in this ratio may present an alternative explanation to explain the larger volume reduction produced by *prog-best*. Two floes with the same diameter but different thickness will, under identical conditions, contribute the same volume to the total basal melt (provided the thinner floe does not melt out) whereas the thicker floe will contribute a greater volume to the total lateral melt, since lateral melt volume is proportional to thickness. This means that increasing the lateral melt contribution to the total melt increases the loss of thick ice in a given melt season. Vertical sea ice growth rates are inversely proportional to the sea ice thickness. Therefore, whilst the moderate reductions in thickness across large areas of sea ice from basal melt can be recovered within a single freeze-up season, the recovery of thick ice that has completely melted out from lateral melt will take several seasons of freeze-up to recover despite being over a smaller area. The reason for larger reductions in sea ice volume for *prog-best* compared to *WIPo-best* may therefore be a result of a changes to the ice thickness distribution that emerge due to the higher lateral to basal melt ratio for *prog-best* compared to *WIPo-best*. This result shows that the inclusion of an FSD model can have important mechanistic impacts on sea ice evolution, even if the immediate change in pan-Arctic properties are small. Smith et al. (2022) recently demonstrated that the partitioning between lateral and basal melt can also have a large impact on open water formation, with implications for albedo feedback.

### 5.3 Advantages and disadvantages of FSD models

We have examined examples of two broad categories of FSD models where either the FSD shape is imposed or where the FSD shape emerges from parametrisations at process level. A factor that must always be considered when introducing new components to sea ice and climate models is computational efficiency. A simple, low-cost FSD plugin would in general be preferential for use in climate models, but only if it is able to capture how the FSD will behave in future climate scenarios. A high level of uncertainty around model parameters and parameterisations would also preclude the use of a given FSD model in climate simulations. The WIPoFSD model is not computationally expensive; including the WIPoFSD model within CICE increases model run time by 30%. In comparison, the prognostic model is computationally expensive and is data intensive. The use of 12 floe size categories with the standard 5 thickness categories introduces a total of 60 floe size-thickness outputs to the model and simulation times increase by a factor of 2.1. Extending this to 16 floe size categories leads to a total of 80 categories and a further increase in simulation run time. The number of floe size-thickness categories can also make it difficult to diagnose and understand how changes to the sea ice state emerge in response to prognostic model processes. In comparison, identifying the mechanisms driving changes in sea ice state is more straightforward with the WIPoFSD model. Development of the prognostic model can also be more time intensive since each process requires either observations or lab-based studies to determine a suitable parameterisation to describe the physical process in the model. It is worth noting that future advancements in modelling techniques may reduce or mitigate the computational expense or complexity of either model e.g. Horvat and Roach (2022) presented a machine-learning-based parameterisation to simulate wave break-up of sea ice floes that can replace the existing treatment of wave break-up in the prognostic model. The study found that CICE simulations including the prognostic model with this new parameterisation have an approximately 40% longer run time than CICE simulations without the prognostic model i.e. a comparable cost to the WIPoFSD model.

Another important test for any FSD model is whether it simulates realistic FSD shape and variability. Figure 4 shows a power law produces a strong fit to observations of floes over a mid-sized (100 m – 2 km) range. The prognostic model with brittle fracture performed comparably to the power-law fit except in the Chukchi Sea. However, this comparison only included observations covering one quarter of the year and excluded floes smaller than about 100 m and larger than 2 km. Floes smaller than 100 m are particularly important for determining the impact of a given FSD on sea ice evolution (Bateson et al. 2020) and there is growing evidence that the power law may not hold across all floe sizes (Horvat et al., 2019) and that the power-law exponent changes significantly over an annual cycle (Stern et al., 2018b). In Bateson et al. (2020), it was found that imposing the annual cycle reported by Stern et al. (2018b) on the exponent only had a small impact on the sea ice state. The annual cycle

imposed was taken as the mean from the Chukchi and Beaufort Seas only, so it is not sufficient evidence to conclude that a fixed exponent is a reasonable assumption.

A key advantage of the prognostic model approach is the shape of the FSD is an emergent feature of the model rather than imposed, avoiding the need to make assumptions about FSD shape or variability, notwithstanding the newly introduced brittle fracture scheme, which, as discussed above, requires further development. This also means the prognostic model can be used to understand the role of individual processes in determining the emergent FSD and can respond to future changes in the behaviour or strength of these processes. In section 5.2 it is shown how behaviours seen within the prognostic model such as the preferential melt out of smaller floes from the distribution or explicit simulation of floe formation and growth processes have impacts on the evolution of the sea ice cover. These impacts are not seen within the WIPoFSD model due to the restrictions of assuming a fixed FSD shape. Furthermore, the WIPoFSD model effectively operates by tuning model parameters to best capture observations of the FSD, however this tuning may not be appropriate over the full timescale of a simulation.

**5.4 Limitations of these results**

A significant limitation originates from the limited availability of observations to constrain FSD model parameters and parameterisations. Whilst the parameters selected for the standard setup of the WIPoFSD model considered here were motivated as a best fit to observations, Bateson et al. (2020) demonstrated high sensitivity in the model response within the observational uncertainty of these parameters. For the prognostic model, the uncertainties are primarily associated with both the representation of existing processes and important processes not yet represented in the model. For example, Roach et al. (2019) demonstrated that using a full wave model coupled to CICE rather than the in-ice wave scheme used here approximately doubled the total lateral melt, though this was compensated by a reduction in basal melt of comparable magnitude.

A second limitation emerges from the use of a standalone sea ice model since this prevents the full representation of sea ice-ocean or atmosphere feedbacks. For example, Roach et al. (2019) found with a coupled sea ice-ocean model that their prognostic FSD setup produced an increase in lateral melt in the Arctic about 3-4 times higher than the reduction in basal melt, resulting in an approximately 20% increase in the total lateral and basal melt. Roach et al. (2019) do not identify the mechanism responsible for this increase, so it is not clear whether the FSD model setups used here would produce a similar response in a coupled CICE-NEMO setup.

In addition, this study has not explored the role of the FSD in sea ice evolution in the Southern Ocean. Several observational studies e.g. Alberello et al. (2019) show the presence of FSDs dominated by pancake ice floes smaller than 10 m in the Southern Ocean. Sensitivity studies presented in Bateson et al. (2020) suggest that distributions dominated by such small floes can result in a significant increase in total melt and a large corresponding reduction in sea ice volume. Therefore, conclusions regarding the role of the FSD in Arctic sea ice evolution do not necessarily extend to the Antarctic. For example, Roach et al. (2019) apply a version of the prognostic model to both the Arctic and Antarctic (including a coupled wave model but not the brittle fracture scheme) and demonstrate a pan-Antarctic reduction in sea ice volume whereas in the Arctic there are regions of volume increase and decrease, with the latter found primarily in the MIZ.

**6. Conclusion**

In this study we have evaluated and compared two alternative approaches to modelling the sea ice floe size distribution: a prognostic model where the shape of the FSD emerges from the model physics (Roach et al., 2018, 2019), and the WIPoFSD model where the shape of the FSD is constrained to a power law with a fixed exponent (Bateson et al., 2020). New, high-resolution observations of the FSD were used to assess model performance in simulating the FSD shape for mid-sized floes and to determine FSD model parameters for the comparison. The prognostic model was unable to simulate the observed FSD shape, however the introduction of a new scheme to approximate the effects of brittle fracture related processes significantly improved prognostic model performance.

Simulations were completed using the two FSD models within a standalone setup of the sea ice model CICE, though it should be noted that both FSD models can easily be implemented into any continuum sea ice model. Whilst impacts of both FSD

models were small over a pan-Arctic scale, larger impacts could be found regionally. Changes to the spatial distribution in sea ice area fraction were consistent with known model biases, particularly for the prognostic model. Clear differences were found between the two models in the strength of the early melt season and long-term trends in the sea ice volume. The faster retreat of sea ice in the early melt season for the prognostic model compared to the WIPoFSD model was attributed to the different model treatments of floe formation and growth in winter. The slower retreat in extent during the later melt season for the prognostic model was found to be a result of melt out of smaller floes, a feedback process not possible with the WIPoFSD model due to the restrictions on FSD shape. It was also highlighted how changes to the lateral to basal melt ratio can indirectly impact the volume of winter sea ice growth via the ice thickness distribution. These results are important for climate modellers as they suggest that the FSD may not be a significant source of structural uncertainty in climate models. FSD processes will, however, be of importance for several key applications and research questions such as regional sea ice modelling and the formation of open water during the melt season (e.g. Smith et al., 2022).

We discussed advantages and disadvantages of the two approaches to modelling the FSD. The WIPoFSD model is more computationally efficient and a simple model to interpret. In addition, a power law fit using a single exponent averaged across all FSD observations produced a good fit to the observed FSD across all locations in the dataset considered here. However, some behaviours seen with the prognostic model could not be replicated by the WIPoFSD model due to restrictions on FSD shape. Furthermore, the prognostic model is better able to respond to regime change of the processes that determine the FSD shape and new modelling techniques could significantly mitigate the computational cost (e.g. Horvat and Roach, 2022).

Future work should focus on the development of a full physical treatment of the impact of brittle fracture on the FSD. Whilst the quasi-restoring scheme presented here is a useful tool to improve prognostic model performance and based on idealised models of brittle fracture, its current formulation relies on significant approximations. The concept of defining the 'damage' of a given area of sea ice such as used within the Maxwell-EB rheology (Dansereau et al., 2016) presents a promising basis for future developments of the brittle fracture scheme. In addition, it would also be beneficial to evaluate whether the conclusions reached in this study extend to the Antarctic. Finally, the results presented here highlight the need to collect further observations of the FSD. Horvat et al. (2019) demonstrated that it is possible to estimate the area-weighted floe size from satellite imagery. It is plausible that $l_{eff}$ could also be estimated from satellite imagery especially since the methodology of Horvat et al. (2019) involved collecting linear statistics of floe size, and $l_{eff}$ is a linearly averaged representation of the FSD. This would establish a way to observationally establish the spatial and temporal variability of the FSD. These observations can provide further constraints for FSD models, which have previously been demonstrated to have high sensitivity to FSD parameters (Bateson et al., 2020) and how individual processes are represented or parameterised (Roach et al., 2019).

**Appendices**

**Appendix A – Description of processes represented in the WIPoFSD model**

The WIPoFSD model presented in Bateson et al. (2020) includes four mechanisms that can change $l_{var}$ and therefore the FSD. The first of these, lateral melting, is treated by assuming the reduction in $l_{var}^2$ from lateral melting is proportional to the reduction in the sea ice area fraction from lateral melting, $\Delta A_{lm}$:

$$l_{var,final} = l_{var,initial} \sqrt{1 - \frac{\Delta A_{lm}}{A}}. \tag{A1}$$

The second mechanism is the break-up of floes by waves. If a wave break-up event is identified, $l_{var}$ is then updated according to the following expression:

$$l_{var} = \max\left(d_{min}, \frac{\lambda_W}{2}\right). \tag{A2}$$

Here $\lambda_W$ is a representative wavelength, in units of metres. In order to calculate the value of $\lambda_W$ and to identify where the ocean surface conditions are sufficient to drive wave break-up, the WIPoFSD model uses a wave attenuation and floe break-up scheme adapted from the waves-ice model of the Nansen Environmental and Remote Sensing Centre (NERSC) Norway,

details are given by Williams et al. (2013a, 2013b). The WIPoFSD model also uses a wave advection scheme developed by NOC in the NEMO-CICE-WIM model. Further details of how these schemes have been incorporated into the WIPoFSD model, and how break-up events are identified, are available in Bateson et al. (2020).

The WIPoFSD model treats floe growth using a simple floe restoring scheme. During periods where CICE identifies frazil ice growth i.e. when the freezing / melting potential is positive, $l_{var}$ is restored to its maximum value:

$$l_{var,final} = \min\left(d_{max}, l_{var,initial} + \frac{d_{max}\Delta t}{T_{rel}}\right). \tag{A3}$$

$T_{rel}$ is a relaxation timescale that represents how quickly floes would be expected to grow to cover the entire grid cell area. It is set to 10 days, taken as representative of the rapid increase in sea ice concentration during the early freeze-up season. In grid cells that newly transition to having a sea ice cover from an ice-free state, $l_{var}$ is initiated with the value $d_{min}$.

The fourth and final mechanism treated by the WIPoFSD model is advection. $l_{var}$ is transported using the horizontal remapping scheme with a conservative transport equation, the standard within CICE for ice area tracers (Hunke et al., 2015). Since $l_{var}$ is not defined independently for each thickness category, the change in $l_{var}$ after advection and subsequent mechanical redistribution is calculated independently for each thickness category, with the net change in $l_{var}$ taken as the average across all the thickness categories.

**Appendix B – The updated lateral melting scheme**

A derivation is presented in this section of Eq. (14), the updated lateral melting scheme within the version of the WIPoFSD model used in this study. The floe number distribution can be written explicitly according to Eq. (13) and by evaluating the constant, $C$, according to its definition as described in section 2.3:

$$N(x \,|d_{min} \leq x \leq l_{var}) = \frac{(3+\alpha)Al^2}{\alpha_{shape}} \frac{x^\alpha}{[l_{var}^{3+\alpha} - d_{min}^{3+\alpha}]}. \tag{B1}$$

Bateson et al. (2020) derives an expression of $l_{eff}$ for this distribution:

$$l_{eff} = \frac{(2-\alpha)[l_{var}^{3-\alpha} - d_{min}^{3-\alpha}]}{(3-\alpha)[l_{var}^{2-\alpha} - d_{min}^{2-\alpha}]}. \tag{B2}$$

It is also possible to derive an expression for $l_{eff}$ after lateral melting. If floes experience an amount $\Delta l$ of lateral melting on each edge, the total diameter of each floe must decrease by $2\Delta l$. This changes the size of the floes but does not impact the shape of the number distribution i.e. floes of diameter $G$ prior to lateral melting and floes of diameter $G - 2\Delta l$ after lateral melting have the same number density, $N(G)$, where $N(x)$ is the number distribution prior to lateral melting. This description is true provided $d_{min} > 2\Delta l$ i.e. no floes are completely lost from the distribution due to lateral melting. The total perimeter after the lateral melting event, $P_{lm}$, can therefore be calculated as:

$$P_{lm} = \int_{d_{min}}^{l_{var}} \pi(x - 2\Delta l)N(x)\,dx. \tag{B3}$$

$N(x)$ in Eq. (B3) is the number FSD prior to lateral melting, and this equation holds provided $d_{min} > 2\Delta l$. This can then be evaluated as:

$$P_{lm} = \frac{\pi(3+\alpha)A_{old}l^2}{\alpha_{shape}[l_{var}^{3+\alpha} - d_{min}^{3+\alpha}]}\left(\frac{[l_{var}^{2+\alpha} - d_{min}^{2+\alpha}]}{(2+\alpha)} - \frac{2\Delta l[l_{var}^{1+\alpha} - d_{min}^{1+\alpha}]}{(1+\alpha)}\right). \tag{B4}$$

The subscript for $A_{old}$ indicates that this is the sea ice area fraction before lateral melting. An expression for the total perimeter in terms of the new effective floe size, $l_{eff,new}$, can also be written using the updated sea ice area fraction after lateral melting, $A_{new}$:

$$P_{leff} = \frac{A_{new}l^2\pi}{\alpha_{shape}l_{eff,new}}. \tag{B5}$$

The two expressions for total perimeter after lateral melting can then be equated to give the updated effective floe size, $l_{eff,new}$:

$$l_{eff,new} = \frac{[l_{var}^{3+\alpha} - d_{min}^{3+\alpha}]A_{new}}{(3+\alpha)\,A_{old}} \left( \frac{[l_{var}^{2+\alpha} - d_{min}^{2+\alpha}]}{(2+\alpha)} - \frac{2\Delta l[l_{var}^{1+\alpha} - d_{min}^{1+\alpha}]}{(1+\alpha)} \right)^{-1}. \tag{B6}$$

It is possible to calculate an analytical result for $A_{new}$ as a result of lateral melting of floes across the distribution, however CICE already accounts for changes to the sea ice area fraction. For internal model consistency, it is this internal CICE $A_{new}$ that will be used.

In order to parameterise processes in terms of $l_{eff}$, a method is needed to calculate $l_{var}$ from $l_{eff}$. There is no analytical solution to this problem; instead a numerical approach must be used such as Newton-Raphson iteration:

$$x_{n+1} = x_n - \frac{f(x_n)}{f'(x_n)}. \tag{B7}$$

Here $x$ is $l_{var}$ and the function to solve is derived from the expression to calculate $l_{eff}$ for the WIPoFSD model i.e.

$$f(l_{var}) = 0 = \frac{(2+\alpha)[l_{var}^{3+\alpha} - d_{min}^{3+\alpha}]}{(3+\alpha)\,[l_{var}^{2+\alpha} - d_{min}^{2+\alpha}]} - l_{eff}. \tag{B8}$$

The iterative scheme can then be evaluated as:

$$l_{var,n+1} = l_{var,n} - \frac{\left( \frac{[l_{var.n}^{3+\alpha} - d_{min}^{3+\alpha}]}{(3+\alpha)} - \frac{l_{eff}[l_{var,n}^{2+\alpha} - d_{min}^{2+\alpha}]}{(2+\alpha)} \right)}{l_{var}^{1+\alpha}(l_{var.n} - f(l_{var.n}))}. \tag{B9}$$

Note that, for simplicity, where $\alpha$ = -1, -2 or -3, a value of 0.001 will be taken off. Whilst an exact solution is possible for these cases, this adds additional and unnecessary complexity to a scheme that is already an approximation. This scheme is evaluated until either $l_{var,n+1}$ - $l_{var,n}$ is less than 0.01% of the change in $l_{eff}$ over a timestep or until a maximum of 50 iterations are

complete. In general, the threshold for convergence is achieved within 10 iterations, however where $l_{eff}$ and $l_{var}$ are close in value i.e. where $l_{var}$ is within a few metres of $d_{min}$, convergence can take longer than 50 iterations. These circumstances are associated with conditions of very low sea ice concentration, where the net error in the lateral melt volume calculation associated with the failure to reach the threshold condition for convergence is negligible.

**Data Availability**

Model output used in this paper is publicly available via the University of Reading research data archive (http://dx.doi.org/10.17864/1947.300; Bateson, 2021b). Output for the simulation *prog-16-nobf* can be found listed under chapter 6 as *cice_cpom_prog_16cat_nobf*. Output for the simulation *prog-16* can be found listed under chapter 7 as *cice_cpom_prog_16cat*. Output for the simulations *ref*, *prog-best*, and *WIPo-best* can be found listed under chapter 8 as *cice_cpom_ref*, *cice_cpom_prog_best*, and *cice_cpom_WIPo_best* respectively. Mask files defining the regions described in

Fig. 3 can be found listed under chapter 6. Please contact the corresponding author to discuss access to model code.

**Author Contributions**

YW and BH produced the novel FSD observations from satellite imagery. AB completed the comparison of the novel FSD observations to model output, with support from YW, BH, DF, and DS. DS adapted the prognostic FSTD model of Roach et al. (2018) into the CPOM CICE stand-alone set-up. AB implemented the novel modifications described in this paper to the

original FSD model setups. AB completed simulations and analysis under the supervision of DF and DS and with further support from YW, BH, JR, and YA. DS provided additional technical support. AB composed the paper with feedback and contributions from all authors.

**Competing interests**

Daniel Feltham, David Schröder, and Yevgeny Aksenov are members of the editorial board of The Cryosphere. The peer-

review process was guided by an independent editor, and the authors have also no other competing interests to declare.

**Acknowledgements**

AB was funded through a NERC industrial CASE studentship with the UK Met Office (NE/M009637/1). DF, DS, YW, and BH were supported by NERC grant (NE/R000654/1). DS was also supported under the NERC projects ACSIS

(NE/N018044/1) and UKESM. YW and BH were also supported through NERC grant (NE/S002545/1). JR was supported by the Joint UK BEIS/Defra Met Office Hadley Centre Climate Programme (GA01 101). YA was supported by the NERC projects ACSIS (NE/N018044/1), "Towards a marginal Arctic sea ice cover" (NE/R000085/1), the NERC Project "PRE-MELT" (NE/T000260/1) and the NERC LTS-S Programme Climate-Linked Atlantic Sector Science (CLASS; NE/R015953/1). This work was also supported by NERC through National Capability funding, undertaken by a partnership between the Centre for Polar Observation Modelling and the British Antarctic Survey. We would like to express our gratitude to Lettie Roach (University of Washington, USA) for her guidance and support in the use of the prognostic FSTD model. Similarly, we would like to thank Lucia Hosekova (University of Reading, UK; Applied Physics Laboratory, University of Washington, USA) and Stefanie Rynders (National Oceanography Centre, Southampton, UK) for their support in the use of the WIPoFSD model. We would also like to thank the Isaac Newton Institute for Mathematical Sciences for support and hospitality during the "Mathematics of Sea Ice Phenomena" programme when work on this paper was undertaken. Finally, we would like to thank both the reviewers and the editor, whose detailed and constructive comments have enabled us to significantly improve the manuscript.

A brief note on WIPoFSD model development: The WIPoFSD model code has been derived and modified from a scheme used within the coupled NEMO–CICE–WIM sea ice–ocean–waves interaction model developed by the L. Hosekova and Y. Aksenov at the National Oceanography Centre (NOC) in the EC FP7 project 'Ships and Waves Reaching Polar Regions' (SWARP)' in 2014-2017 (https://cordis.europa.eu/project/id/607476, grant agreement 607476). The physics of waves–ice interactions in NOC–WIM (WIM – stands for waves in ice model) model uses the framework of the waves–in–ice model of the Nansen Environmental and Remote Sensing Center (NERSC, Norway) by Williams et al. (2013a, b). Given the differences between the NERSC Arctic regional model setup (HYCOM ocean model with an early EVP sea ice rheology realisation and Semtner's sea ice thermodynamics) and the global NOC–NEMO–CICE setup, the coding of the NOC–WIM model and all coupling with ocean and sea ice modules has been done completely from the start by NOC. During the course of the NOC–NEMO–CICE–WIM model development L. Hosekova, advised by Y. Aksenov, has incorporated extra key processes in the model, including up-wind wave spectrum advection in the ice-covered areas, floe sizes evolution due to lateral floes melting, including renormalization algorithm with the Newton-Raphson method to compute mean from maximum floe sizes, freeze-up and floe sizes advection, and an optional choice of the multiple power law exponents for the FSD (Rynders, 2017); for these we also acknowledge contributions from G. Madec (IPSL) and A.J.G Nurser (NOC). The NOC–NEMO–CICE–WIM model includes the novel development of the combined EVP-collisional rheology (EVCP) (Feltham, 2005) coded in CICE by S. Rynders and advised by Y. Aksenov and D. Feltham (Rynders, 2017, Rynders et al., 2022). The NEMOv3.6–CICEv5.1–WIM model code had been shared with the Centre for Polar Observations and Modelling (CPOM) at the University of Reading for the joint research under the UK Joint Marine Modelling Programme (the UK Joint Weather and Climate Research Programme – JWCRP).

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

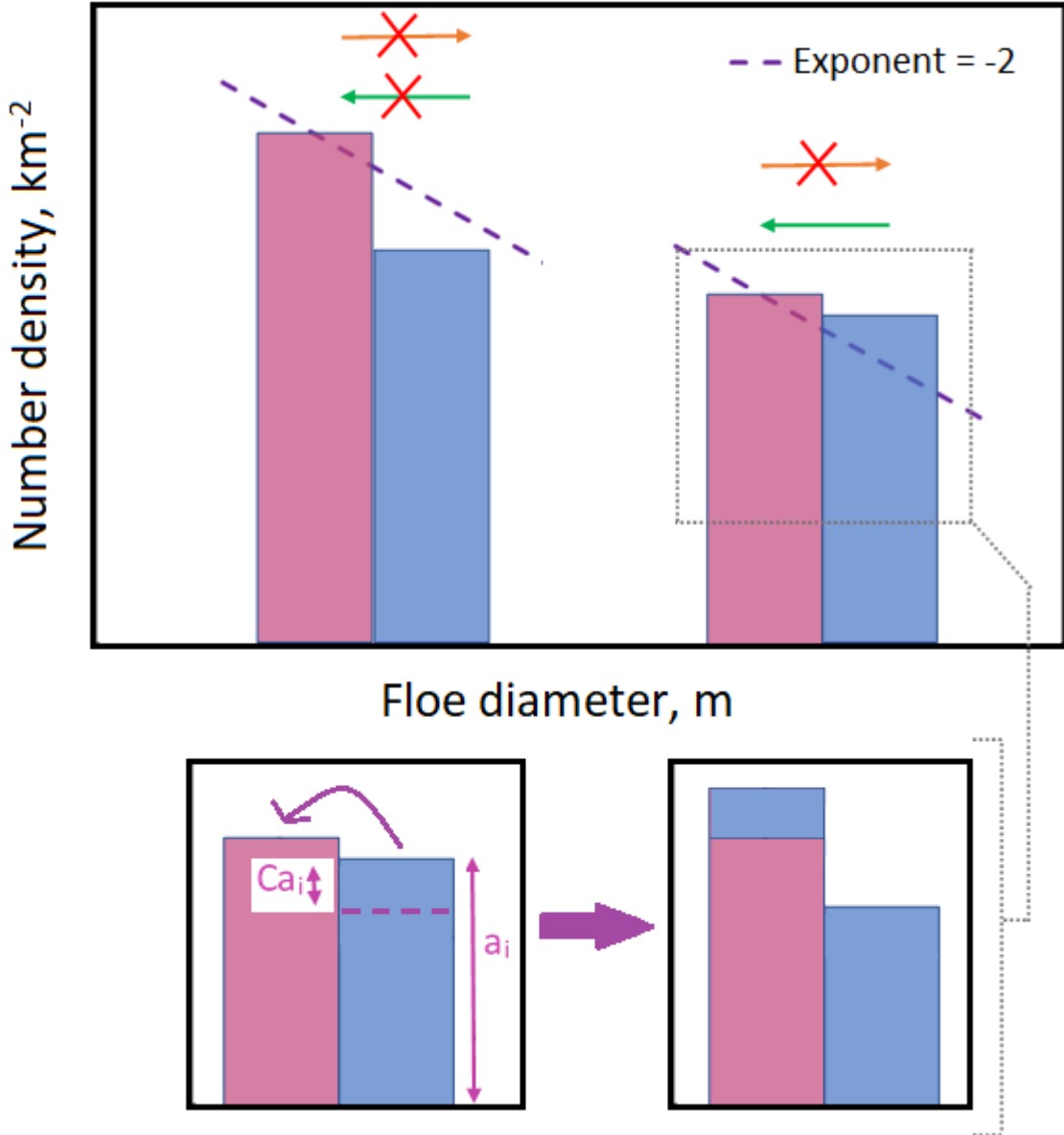

**Figure 1:** Diagram of the quasi-restoring brittle fracture scheme introduced to the prognostic FSD model. The upper section of the plot shows two adjacent pairs of floe size categories, where blue highlights the larger category and red the smaller category. The brittle fracture scheme only transfers sea ice area fraction from a larger category to the adjacent smaller category and only where the number density gradient between adjacent categories in log-log space is larger (more positive) than $-2$. A slope with this gradient is shown by the purple, dashed line. The lower section of the plot shows how the scheme can modify the FSD. The sea ice area fraction transferred from the larger to the smaller floe size category is $Ca_i$ where $a_i$ is the total sea ice area fraction in the larger category and $C$ is the restoring constant.

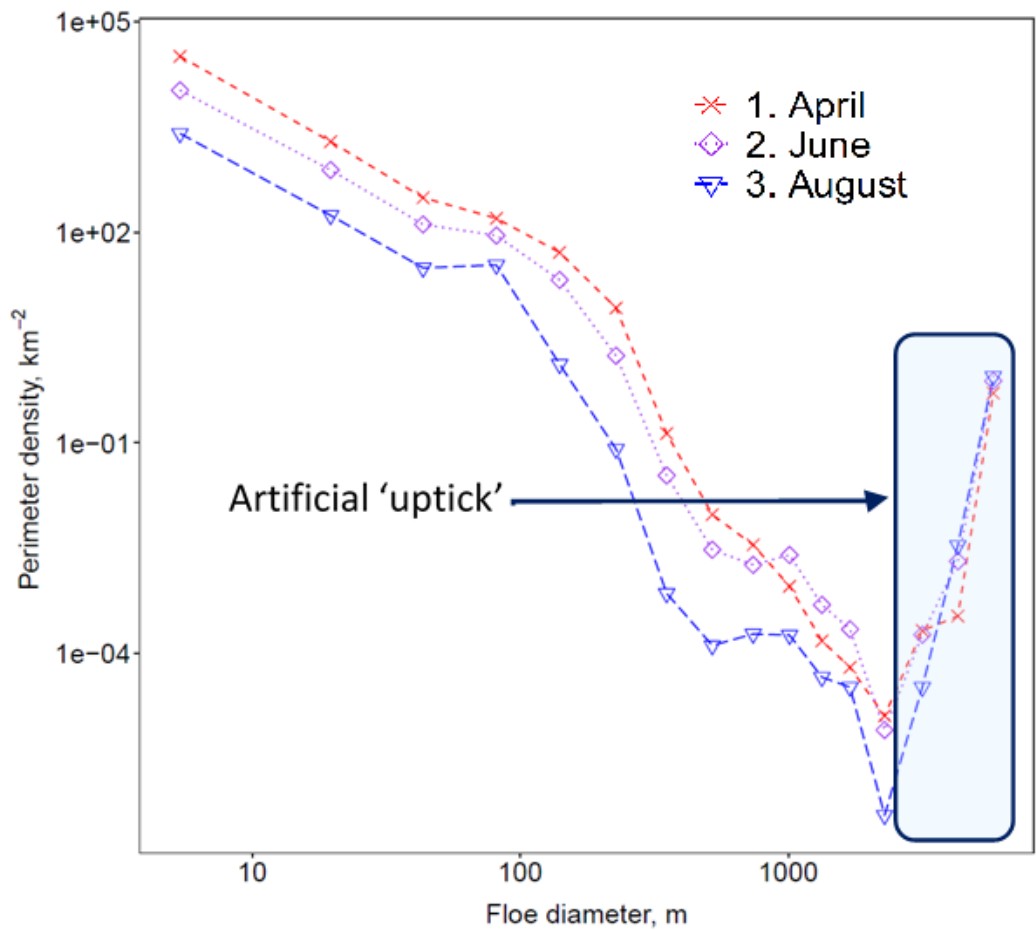

**Figure 2:** An example of prognostic model output from the *prog-16-nobf* simulation averaged over the MIZ (i.e regions with between 15 – 80% sea ice concentration). Presented in the figure is the perimeter density distribution, $km^{-2}$ for the April (red, cross, dashed), June (purple, diamond, dotted), and August (blue, triangle, long-dash) averaged over 2000 – 2016. Also highlighted in the figure by a blue transparent box is an artificial 'uptick', a feature of the model also reported by Roach et al. (2018) that results from prognostic model design and structure (e.g. missing fragmentation processes, upper limit on floe size) and does not represent a physical behaviour seen for the FSD.

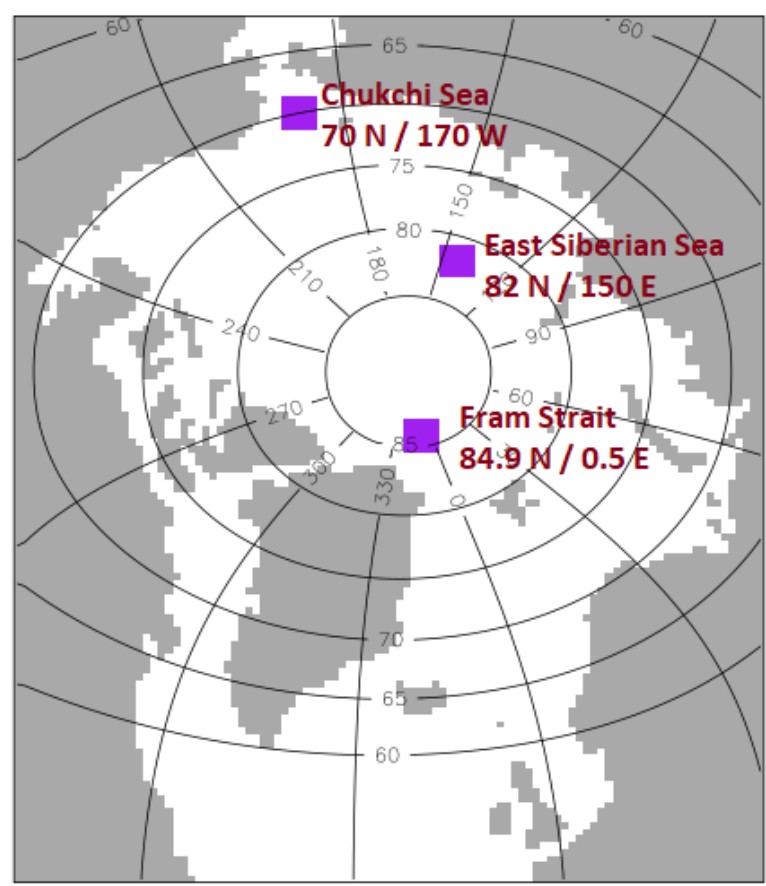

**Figure 3:** Boxes indicate the areas over which the prognostic model emergent FSD is averaged to represent the three locations included in the observational study. Each case study area spans a set of 5 x 5 grid cells that includes the site stated for collection of observations.

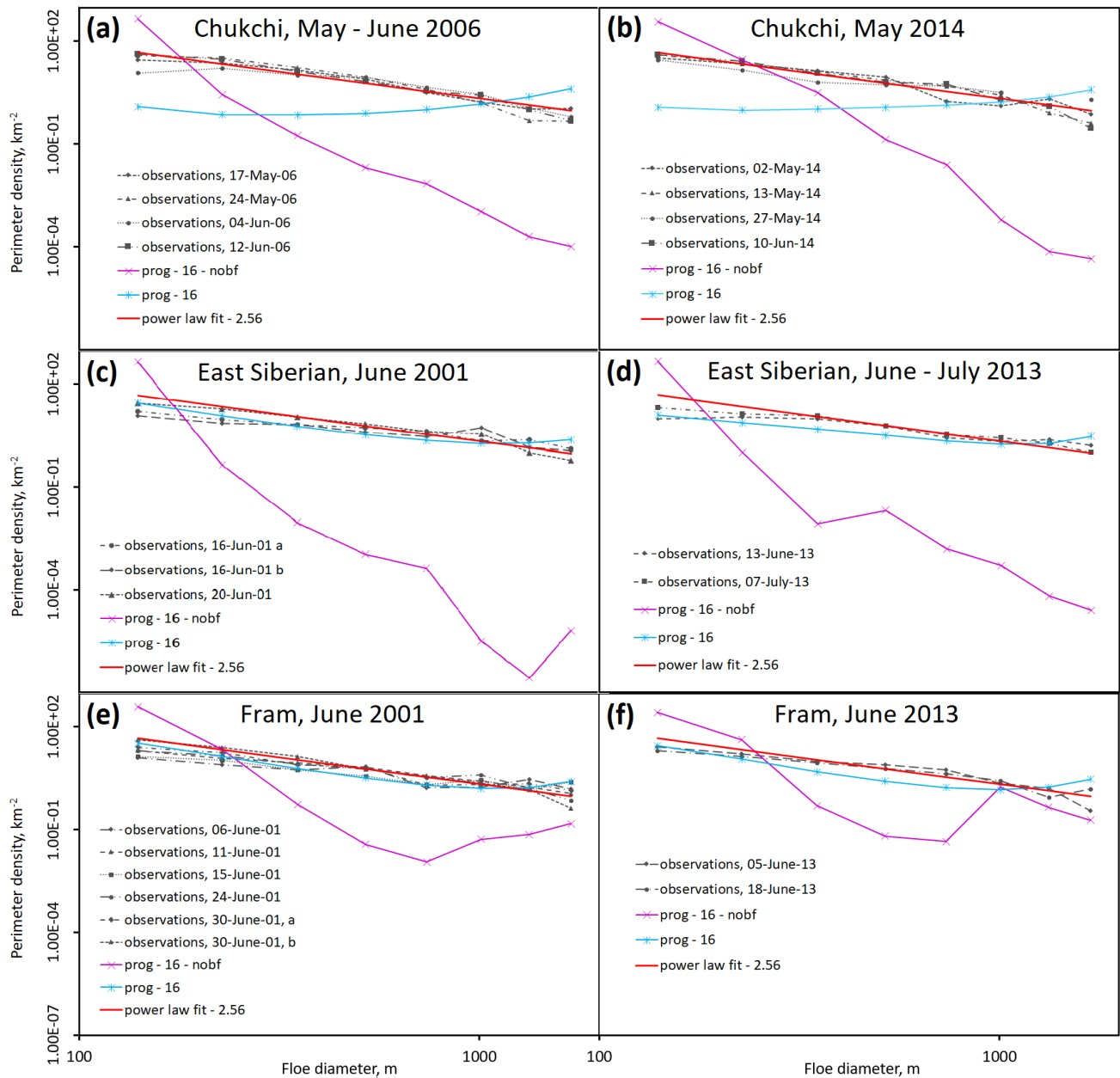

**Figure 4:** A comparison of the observations and prognostic model output for the perimeter density distributions for the Chukchi Sea in May – June 2006 (a) and May 2014 (b), the East Siberian Sea in June 2001 (c) and June – July 2013 (d), and the Fram Strait in June 2001 (e) and June 2013 (f). Observations are identified with pink or purple dashed lines. Output for *prog-16* (light blue, solid, stars) and *prog-16-nobf* (dark blue, solid, crossed) is averaged across the relevant region identified in Fig. 3 over the stated month(s). The average power-law fit across all locations is also shown (red, solid). The floe size data used within this figure was obtained from imagery using the methodology of Hwang et al. (2017) and the exponent of the power-law fit was calculated according to Virkar and Clauset (2014).

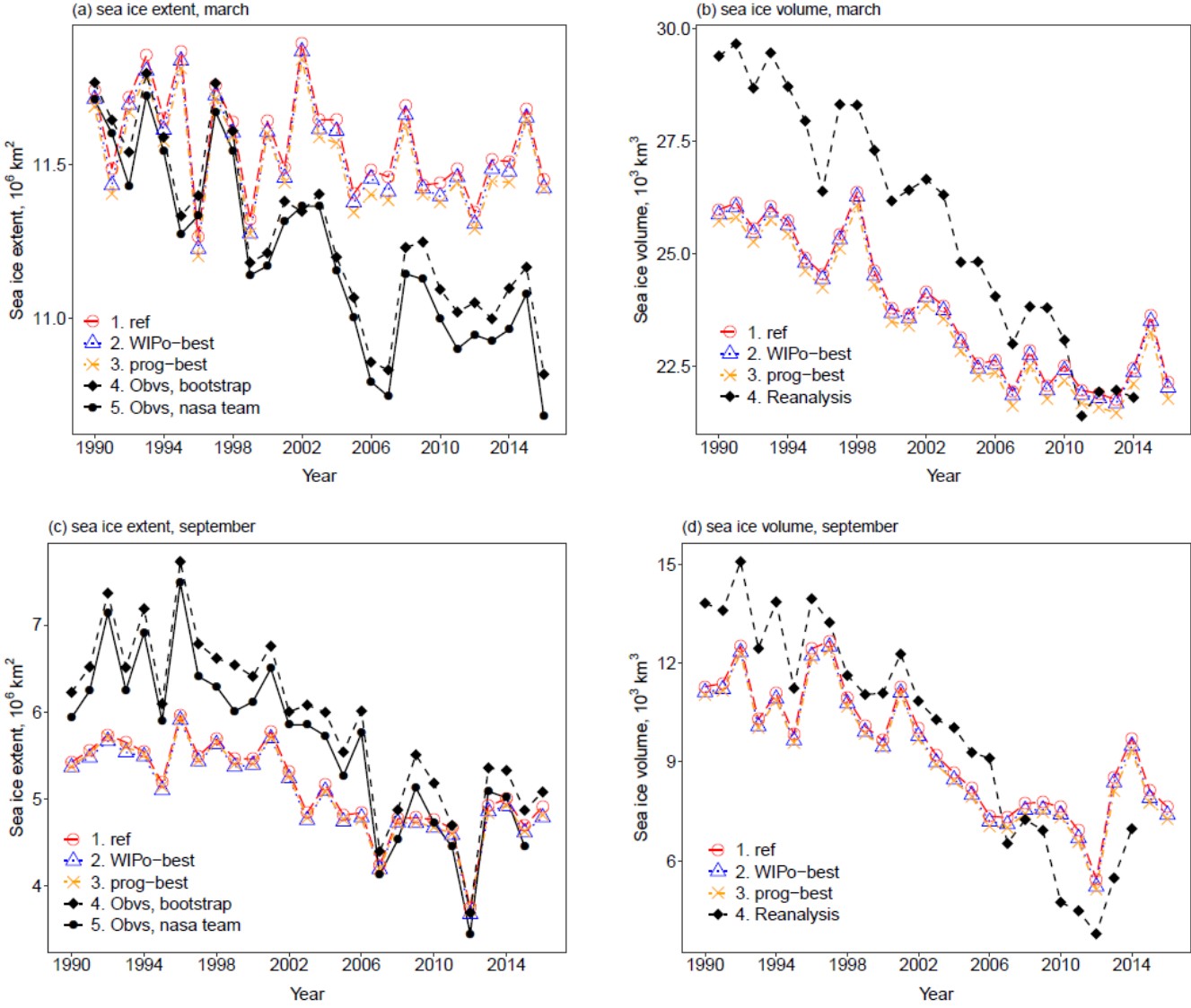

**Figure 5:** The total Arctic sea ice March extent (a, top left), March volume (b, top right), September extent (c, bottom left), and September volume (d, bottom right) within the model domain over the period 1990 – 2016 for *ref* (red, circles), *WIPo-best* (blue, triangles), *prog-best* (yellow, cross) and observations / reanalysis (black). Sea ice concentration data is obtained from satellites using the Bootstrap (filled diamond, dashed) algorithm version 3 (Comiso, 1999) and the NASA Team (filled circle, solid) algorithm version 1 (Cavalieri et al., 1996). Sea ice volume data (filled diamond, dashed) is taken from PIOMAS (Zhang and Rothrock, 2003).

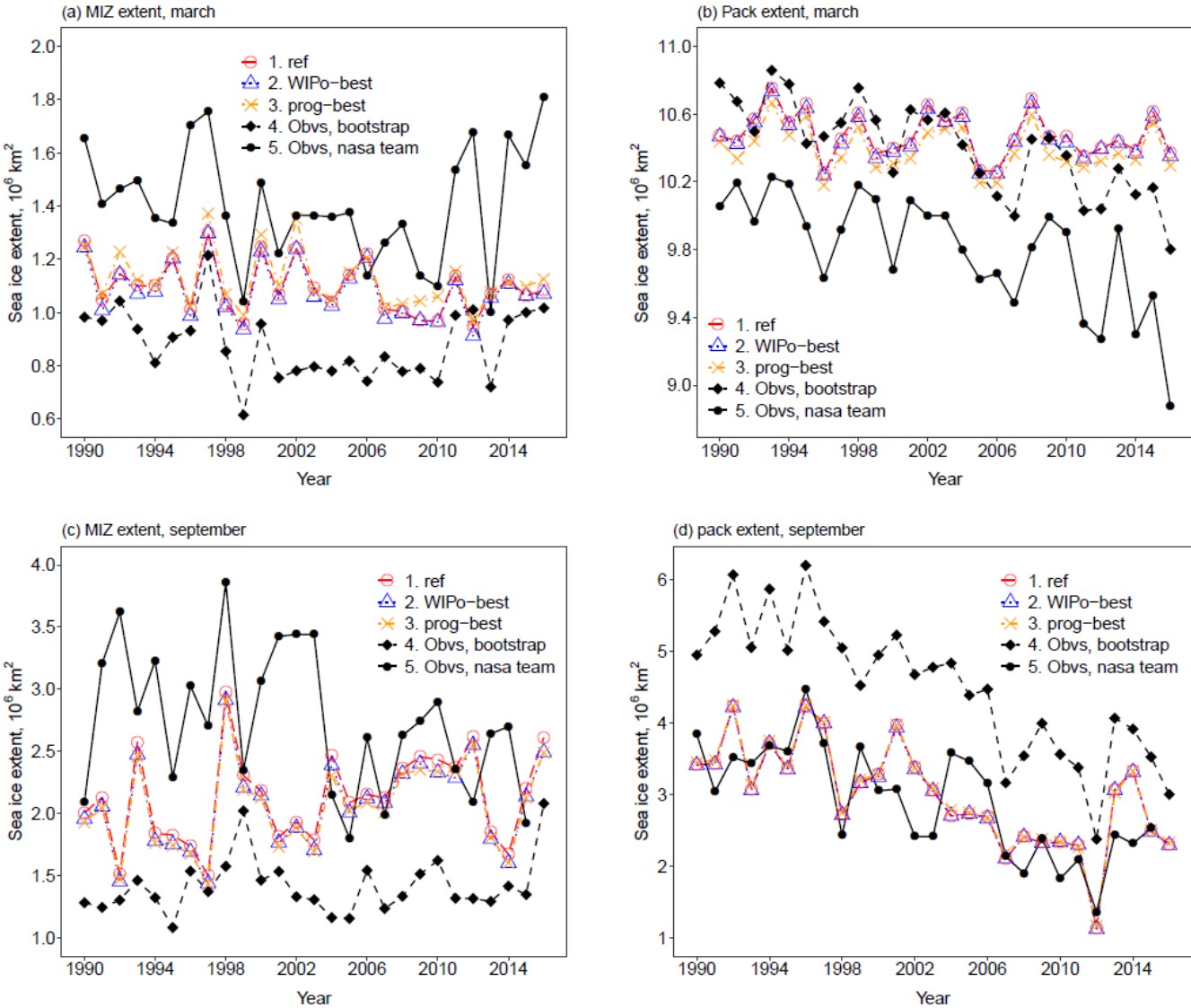

**Figure 6:** The total Arctic sea ice March MIZ extent (a, top left), March pack extent (b, top right), September MIZ extent (c, bottom left), and September pack extent (d, bottom right) over the period 1990 – 2016 for *ref* (red, circles, long dash), *WIPo-best* (blue, triangles, dotted), *prog-best* (yellow, cross, dot-dash) and observations (black). Sea ice concentration data is obtained from satellites using the Bootstrap (filled diamond, dashed) algorithm version 3 (Comiso, 1999) and the NASA Team (filled circle, solid) algorithm version 1 (Cavalieri et al., 1996). Sea ice volume data (filled diamond, dashed) is taken from PIOMAS (Zhang and Rothrock, 2003). The MIZ is here defined as the region with between 15% and 80% sea ice concentration.

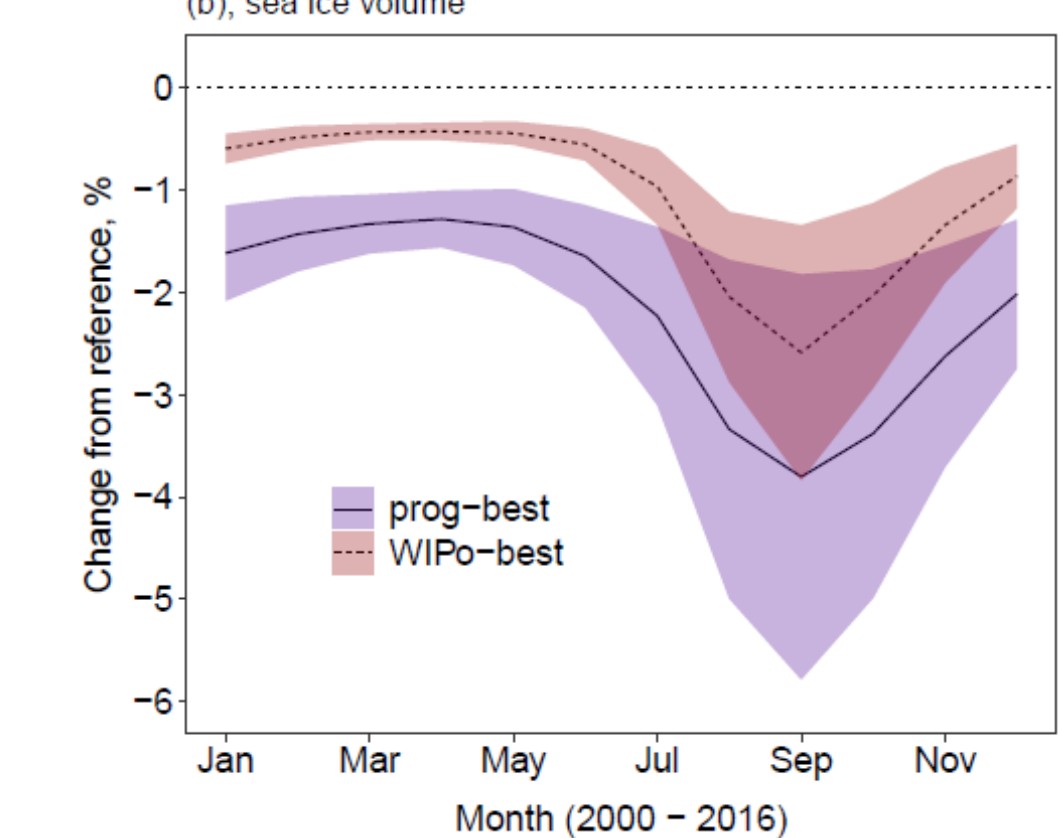

**Figure 7:** Difference in sea ice extent (top) and volume (bottom) of *prog-best* (solid, blue shading) and *WIPo-best* (dashed, red shading) relative to *ref* averaged over 2000 - 2016. The shading shows, in each case, plus or minus two times the standard deviation around the mean.

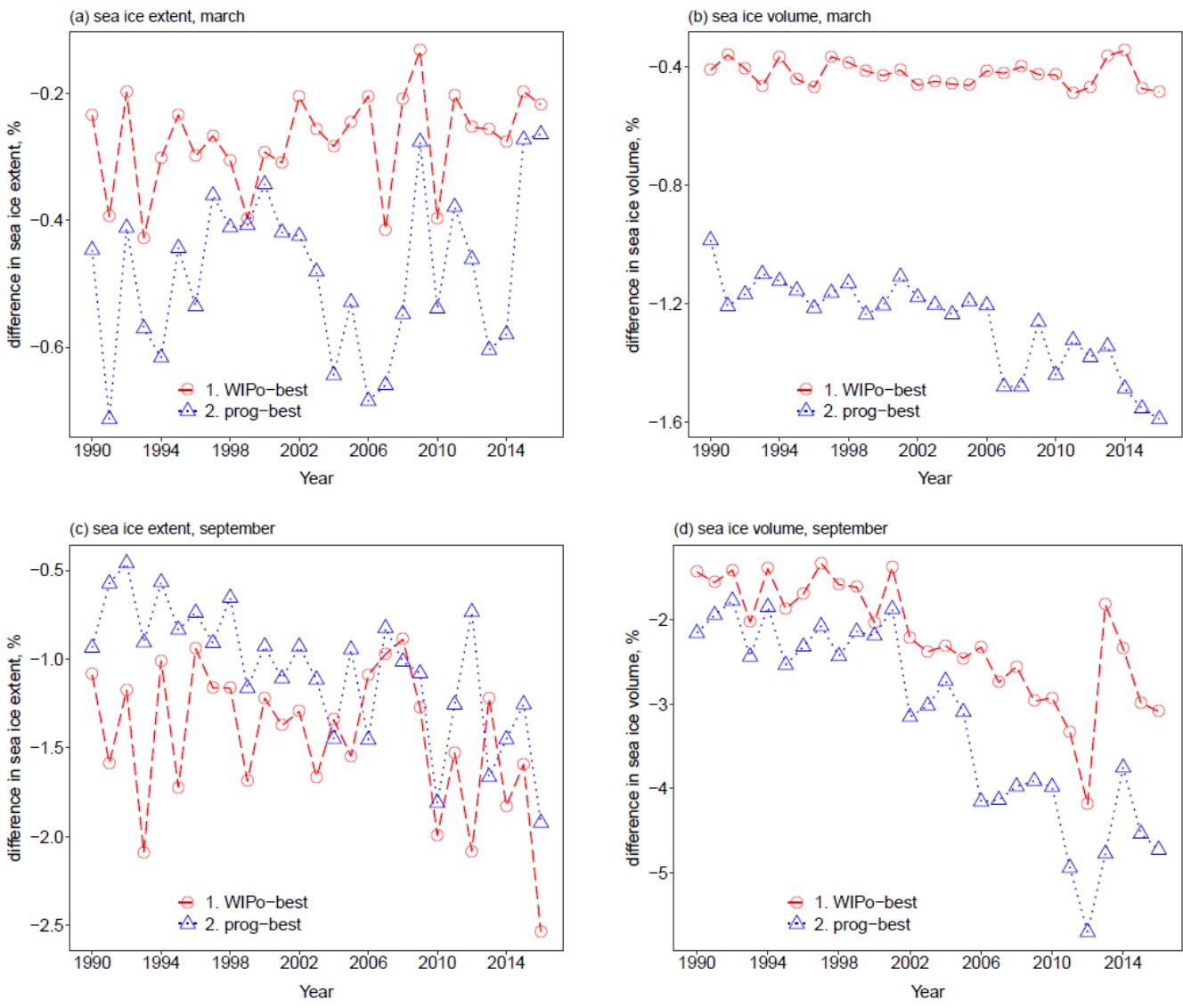

**Figure 8:** The % difference in the Arctic total sea ice March extent (a, top left), March volume (b, top right), September extent (c, bottom left), and September volume (d, bottom right) over the period 1990 – 2016 for *WIPo-best* (red, circles, dashed), and *prog-best* (blue, triangles, dotted) relative to *ref*.

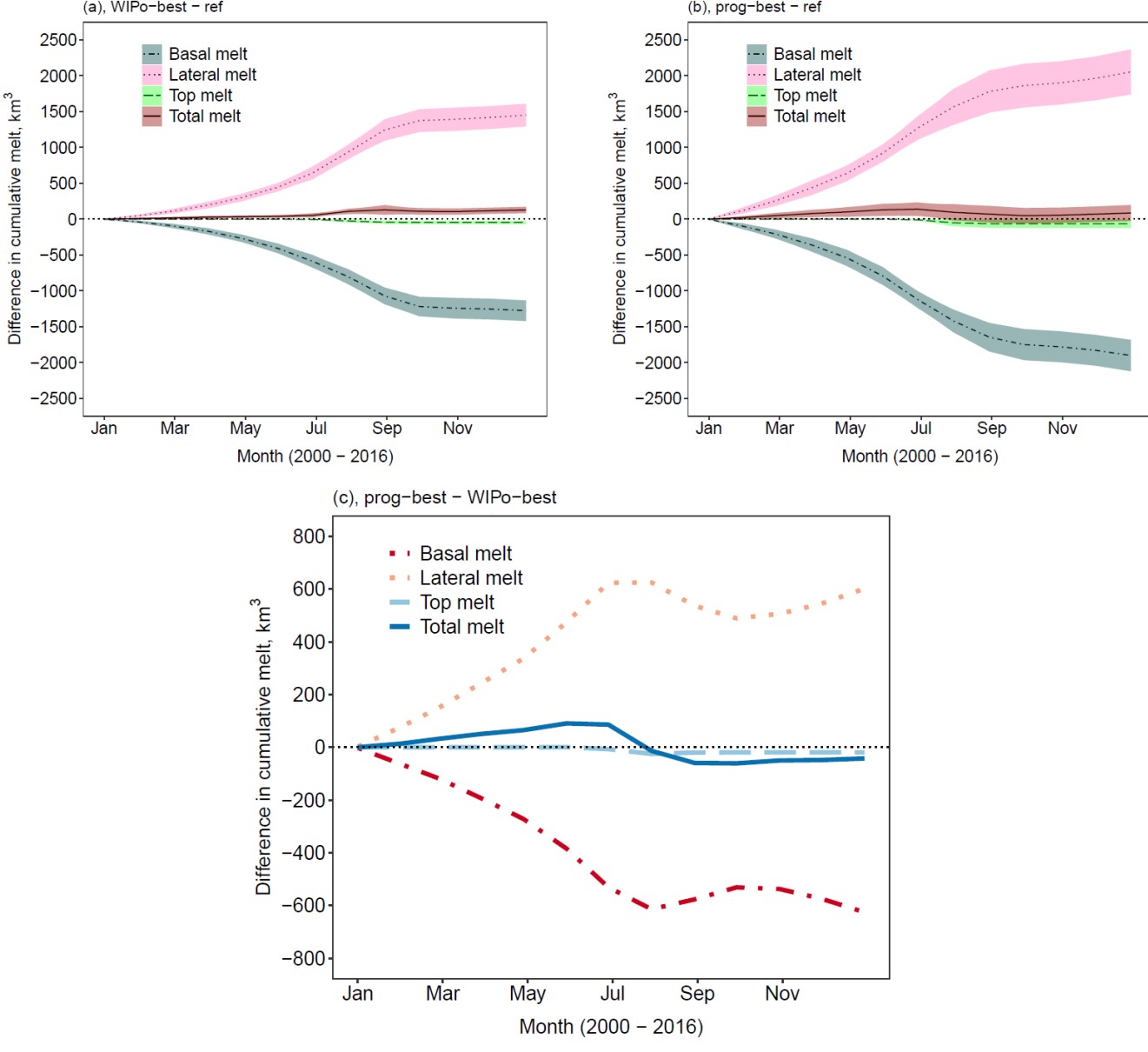

**Figure 9:** The top two plots show the difference in the cumulative lateral (pink ribbon, dotted), basal (grey ribbon, dot-dashed), top (green ribbon, dashed), and total (red ribbon, solid) melt averaged over 2000–2016 for *prog-best* (a, left) and *WIPo-best* (b, right) relative to *ref*. The ribbon shows, in each case, the region spanned by the mean value plus or minus twice the standard deviation. The bottom plot shows show the difference in the cumulative lateral (orange, dotted), basal (red, dot-dashed), top (light blue, dashed), and total (dark blue, solid) melt averaged over 2000–2016 for *prog-best* relative to *WIPo-best* (c).

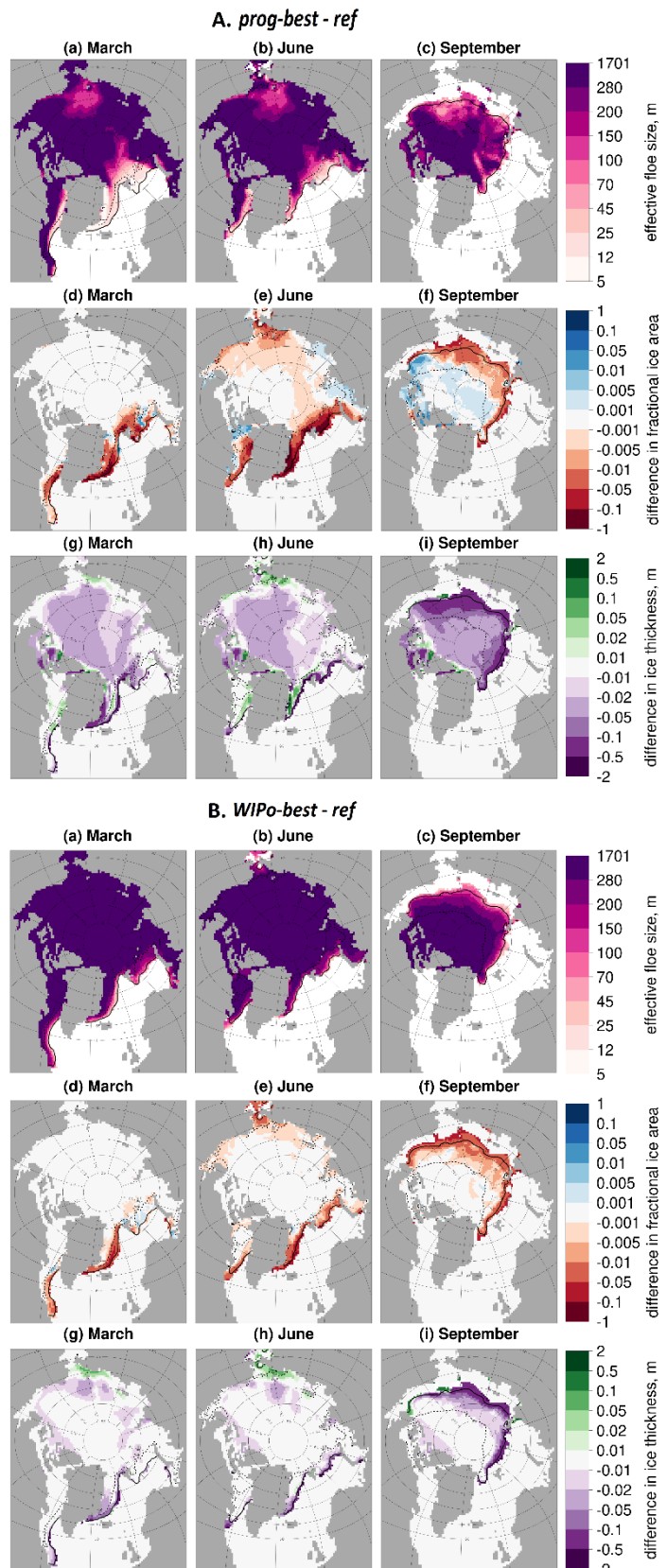

**Figure 10:** Absolute values of $l_{eff}$ (a–c) for *prog-best* (A, top) or *WIPo-best* (B, bottom) and the difference in sea ice area fraction (d–f) and difference in ice thickness (g–i) between *prog-best* (A, top) or *WIPo-best* (B, bottom) and *ref* averaged over 2000–2016. Results are presented for March (a, d, g), June (b, e, h), and September (c, f, i). Values are shown only in locations where the sea ice concentration exceeds 5 %. The inner (dashed black) and outer (solid black) extent of the MIZ averaged over the same period is also shown.

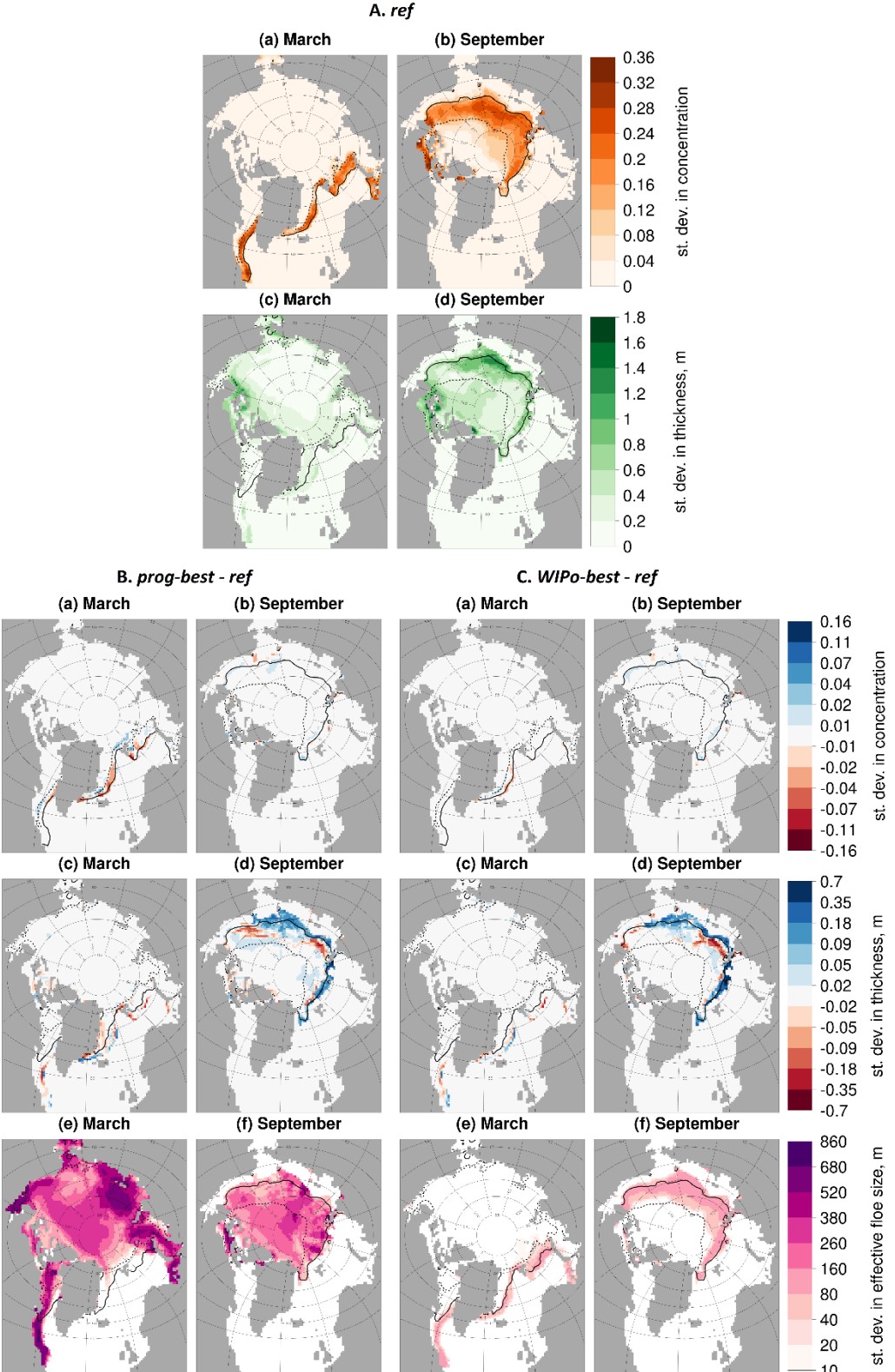

**Figure 11:** Section A (top) shows the standard deviation of the sea ice area fraction (a, b) and thickness (c, d) in March (a, c) and September (b, d) for *ref*. Section B (bottom left) and C (bottom right) show difference plots in the standard deviation of the sea ice area fraction (a, b) and thickness (c, d) in March (a, c) and September (b, d) for *prog-best* and *WIPo-best* relative to *ref* respectively. In B and C, the standard deviation in $l_{eff}$ is also plotted for both March (e) and September (f). Values are shown only in locations where the sea ice concentration exceeds 5 %. The inner (dashed black) and outer (solid black) extent of the MIZ averaged over the same period is also shown. Plots show that changes to the standard deviation in the sea ice area fraction and thickness are generally localised to the outer edge of the MIZ.

| Term | Description |
|------|-------------|
| *Floe size* | The mean diameter of a sea ice floe. |
| *Perimeter density* | Floe perimeter per unit sea ice area. Calculated as the total perimeter of an ensemble of floes divided by the total sea ice area. |
| *Effective floe size* | The diameter of the set of identical floes that has the same perimeter density as an ensemble of floes of variable size. |
| *Floe size distribution / FSD* | General term to refer to the size distribution of an ensemble of sea ice floes. The FSD can be considered in terms of the probability a floe will have a given size (probability distribution), the number of floes with a given size (number distribution), the perimeter of floes with a given size (perimeter distribution), or the area of floes with a given size (area distribution). |
| *Density distribution* | When expressing the FSD in terms of a 'real' descriptor i.e. as a number, perimeter, or area distribution, the FSD can be expressed per unit sea ice area i.e. as a density distribution. |
| *Non-cumulative distribution* | The value reported for each floe size (category) in a non-cumulative FSD (e.g. number of floes) includes floes of that size only. Some studies alternatively consider a cumulative FSD where the value reported per floe size (category) refers to all floes larger than the given size (category), in addition to those of that floe size (category). |
| *Fragment size distribution* | A generic term to describe the size distribution of any system consisting of an ensemble of individual components with a varying size metric. The FSD is a specific example of a fragment size distribution. |
| *Ice thickness distribution* | Describes the proportion of the total sea ice area within discrete floe thickness categories. This study uses the standard CICE formulation for the ice thickness distribution, as described in Hunke et al. (2015). |
| *Prognostic FS(T)D model* | A modified version of the prognostic FSTD (Floe-Size-Thickness distribution model) of Roach et al. (2018, 2019). Details of the version used here are provided in section 2.2. |
| *WIPoFSD model* | A modified version of Waves-in-Ice module and Power law Floe Size Distribution model of Bateson et al. (2020). Details of the version used here are provided in section 2.3. |
| *(Truncated) Power-law fit* | A power law of the form $x^\alpha$ (where $x$ refers to the floe diameter) between lower and upper floe size limits that has been fitted to observations of floe size. |
| *Power-law exponent* | The value of $\alpha$ for a power law of form $x^\alpha$. Can also be described as the slope of a power law when plotted using logarithmic axes. |

**Table 1:** A summary of important terms related to the sea ice floe size distribution (FSD) that are used within this study. Note that these terms are defined in the context of this study only.

| Simulation | Description of Simulation | Technical details |
|---|---|---|
| *ref* | Reference simulation with fixed floe size and no FSD model. | $l_{eff} = 300\ m$ |
| *prog-16* | Simulation uses standard prognostic FSD model setup but using 16 floe size categories. | 16 floe size categories following Gaussian spacing; includes brittle fracture scheme described in section 2.2.2. |
| *prog-16-nobf* | Simulation uses standard prognostic FSD model setup but without brittle fracture and using 16 floe size categories. | 16 floe size categories following Gaussian spacing. Brittle fracture scheme, described in section 2.2.2, excluded from model. |
| *prog-best* | Simulation uses standard prognostic FSD model setup. | 12 floe size categories following Gaussian spacing; includes brittle fracture scheme described in section 2.2.2. |
| *WIPo-best* | Simulation uses WIPoFSD model, with WIPoFSD model parameters optimised against observations. | $d_{min} = 5.375\ m, d_{max} = 30,000\ m,$ $\alpha = -2.56$ |

**Table 2:** A summary of the CPOM CICE simulations described in section 3.1. All simulations are initiated sea ice free on the 1st January 1980 and evaluated until 31st December 2016.

