# Peer review of "Sea ice floe size: its impact on pan-Arctic and local ice mass, and required model complexity"

_The Cryosphere, 2021_

## Referee Comment (RC1)

This is a review of the manuscript entitled *Sea ice floe size: its impact on pan-Arctic and local ice mass, and required model complexity*. In this manuscript, the authors investigate the differences between two ways of representing the floe size distribution (FSD) in sea ice models. The first approach uses a prognostic floe size model, in which floe size evolves freely depending on some physical processes (breakup, lateral melting, welding…). The second approach is simpler, as it constrains the floe size distribution to always obey a truncated power-law. Only the upper-limit of this power-law varies with processes affecting the floe size. After having described the two models and their implementation in a stand-alone version of the sea ice model CICE, the authors evaluate the simulated floe size distribution against available observations in the summer. They show that the original prognostic model leads to unrealistic results in the absence of a process able to break the largest floes considered in their FSD. They suggest this process corresponds to brittle fracture of the ice, and that it can be represented by relaxing the FSD in the prognostic model towards a power-law. They further investigate the impact of the two FSD models on sea ice extent and volume in Pan-Arctic simulations. They find little evidence of any significant improvement of model results related to the addition of FSDs in the sea ice model. They discuss the differences between the two FSD models, as well as their advantages and drawbacks.

The manuscript is overall easy to read, and the method followed by the authors is rigorous and well explained. I acknowledge the quality of the work that has been done, but I think there are a few problems to fix before I can support the publication of the manuscript.

General Comments:

The manuscript is in general well-written and not very long, however I find that some topics are repeated and too much time is spent describing results that are, in my opinion, not key to the study. I find it particularly detrimental to the potential impact of the study, as it makes the paper confusing in places, and the interesting findings and discussions are a bit lost among things that have already been long discussed in previous studies. I think there is potential for this manuscript to address the whole sea ice community, but in its current state I don't believe anyone not familiar with FSD modelling would get what the key findings are. I will try to highlight these problems and suggest ways of improving the manuscript in my specific comments.

My second main comment concerns the brittle fracture mechanism. This process occupies a large amount of the manuscript, however I am not fully satisfied with the way it is presented and discussed. I have the feeling (but I might be wrong, in which case I am sorry) that the authors found out during their evaluation against observations that a mechanism was missing in the prognostic model to break the largest floes into floes of "mid-range" sizes. They realized that observations were showing a power-law behaviour, and therefore improved the prognostic model by adding a relaxation towards this power-law. To explain this behavior, they suggest that brittle-facture is a good candidate, even though it has not really been demonstrated before for this spatial scale. However, the way it is presented in the paper is confusing, mixing LKFs in pack ice, fragmentation by waves and scientific intuition. The study does not do enough to justify the use of this relaxation in the winter in my opinion. I also find

the discussion about this process a bit shallow, particularly as it is a major change compared to the prognostic model used in Roach et al. (2018). My recommendations would be a) to present this new process more carefully, i.e. introduce it as a relaxation of the prognostic FSD towards a power-law, b) motivate this introduction by the fact that the prognostic model fails to reproduce observed FSDs in the summer without this relaxation, c) discuss what this relaxation represents (and I agree with the authors that brittle-failure is a good candidate), and when and where it should be applied. I will also explain these recommendations and my criticism in more detail in the specific comments.

Specific comments (major):
PXLY → Page X Line Y

P2L28 to 44. This is the first introduction to the brittle-fracture process. It is quite long, and for a reader that has not read the full paper yet, I believe the relationship with the rest of the introduction is very obscure. I also believe this level of detail would fit better in section 2.2.2. The paragraph first explains in detail the mechanism of LKF formation in pack ice, then switches to Perovich et al. (2001) that relate floe breakup to melt as thin ice is very weak, and finally refers to Kohout et al. (2016) who report flexural failure (and not in-plane failure) of ice under wave action in places where ice strength is minimal. I think I get that the authors want to say that processes responsible for LKFs at scales >1km might affect floe size at scales <1km, and that heterogeneity in the ice strength/thickness exists at these scales that would ease brittle fracture, but this does not appear explicitly in the text. I also think the mechanical behaviour of sea ice depending on the spatial scale of interest is an open question, which should appear more clearly in the text. Details about the spatial scales discussed in each reference and the one of interest for the study are missing.
I also find the conclusion (*"These observations and model studies collectively suggest that brittle fracture processes impact floe size in winter and that the resulting pattern of linear features resulting from brittle fracture may also influence floe breakup in the subsequent melt season."*) quite far-fetched and not well motivated with these references. Unless the manuscript addresses sea ice rheology at floe scale (which is not the case to me), I find this statement too strong to belong to the introduction.
In this introduction, I would recommend only mentioning the fact the prognostic model used by Roach and others has not been thoroughly evaluated against FSD observations, and that some processes might be missing. For instance, brittle fracture of ice might occur as the ice is thinning in the summer (Perovich et al., 2001).

P4L4->L10: *The lateral heat flux…*
I find this paragraph a bit hard to follow:
a) I don't understand why it is important to explain how heat fluxes are dealt with in CICE here. Please inform the reader of their use in this study. Also, it would be nice to highlight if this is a change from the "standard" in CICE, or if these are all default settings (as is done well further in the text for other code changes).
b) *$F_{frzmlt}$ is computed as…* The authors might want to write the equations instead of describing them in the text, that would likely improve the readability.

c) Why is F_frzmlt capped, and why is it important?

P5L12: Please clarify the definition of l_eff. What do you call *"the perimeter of a FSD"*?

P5L16: You might want to explain briefly why l_eff is better than the average floe size (I believe this is mentioned somewhere else in the text, but it would fit nicely here).

P5L28: *Note that the in-ice…* I don't understand this sentence. Could the authors clarify the difference between their parameterization and the one by Roach et al. (2019)?

Section 2.2.2:

For a reader that has not been through the Results section, the motivations behind the addition of this process remain very obscure. I think part of the motivations currently in the introduction (e.g Perovitch et al., 2001) would fit nicely here. Hinting at the results a bit would also help. Stating that the prognostic model was found to fit poorly with observations without this model would really help to understand the addition of this process. The authors should at least reassure the reader by saying that they are going to investigate the effect of this addition by running two experiments, one with this modified prognostic model, and one without it.

*This condition means…* It could be worth giving the physical interpretation of this sentence, as not all readers are familiar with FSDs.

*Fracture events occur regularly through autumn, winter and spring within the pack ice to form linear features like leads, which subsequently freeze up again…*

I find the current discussion quite vague, and not very well linked to other references in the literature. The way I understand it, the authors assume that at subgrid scale, leads create a network of cracks that define floes in a kind of mechanical strength sense, even though they are separated by thinner ice and not open water. Distribution of these floes is assumed to follow a power-law of exponent -2 as it results from successive fracture events. Between these floes there are therefore weak joints made of thinner ice that will melt faster in the summer than the surrounding ice, to the point where their strength will become very weak (Perovitch et al., 2001). The idea here is therefore that sea ice has a memory of fracturing events, is this correct? If so, this is not so different from the definition of damage in brittle rheologies (cf. the work of Weiss that is already cited, and the models of Dansereau or Rampal, see references at the end of the review), and it could be worth commenting on that, if not here, maybe in the discussion. If not, the text needs to be clarified to make clearer what the actual point being discussed is.

Freezing/floe welding/convergence will act as mechanical healing that will erase any memory of fracturing events, in the winter at least (e.g Rampal et al., 2016). How would it compete with brittle failure in your model? This point is quite important to me, as Roach et al. (2018) present how FSD processes are balanced in the prognostic model over the year, and here a modification is introduced that likely breaks this balance. In a sense, it also relaxes the prognostic model towards a truncated power law with upper limit the maximum floe size in

the model, does it not? This goes against what I think is the original philosophy of the freely-evolving FSD of the prognostic model but this fact is not really discussed here.

*However, the brittle fracture-derived mechanisms operate over different timescales and scale with different properties.* → This is a very vague sentence.

*The timescale for a crack or linear feature in the sea ice to fully melt through is taken to be of the order of 1 month. For simplicity, τ is here set to 30 days.*
I find it very confusing to relate the time scale of brittle fracture, which is a dynamical process occurring in very little time, to the time scale of sea ice melt. To me 30 days is not related to the fracture itself, but to the reduction of ice strength that tends towards 0 as ice melts, making it very sensitive to brittle failure. It would also be nice to a provide more quantitative details to the reader: what thickness/melt rate are you considering here to end up with tau=30days?
Also, to me, this justification does not motivate the addition of the brittle-fracture process in the freezing season. With this motivation, tau should tend towards infinity in the winter. The lack of discussion about this process in winter is particularly detrimental to the study as modelled FSDs are only evaluated in the summer (or at least melting season) in section 4.1. The seasonal impact of this relaxation process should be discussed: does it overwhelm the floe size growth process in freezing conditions, or is it negligible?

Section 4.1
*P9L39: Overall, the inclusion of the quasi-restoring brittle fracture scheme represents a significant improvement in the ability of the prognostic model to capture the shape of the FSD for mid-sized floes.*
I agree, but you have only shown it in summer.

*P9L42: [...] does not include floes smaller than 100 m in the comparison, which are particularly important for determining the impact of the FSD on the sea ice mass balance.*
These floes are important for lateral melting, but is lateral melting important for the mass balance? This is not what I retain from this manuscript, or from studies such as Bateson et al. (2020) and others, at least not for the Pan-Arctic mass balance.

*P10L5: Whilst a reduction in ice area fraction in the largest category and an increase in the smallest category can be expected, the change in ice area fraction in the remaining categories depends precisely on the balance between ice area fraction lost from that category and ice area fraction gained from the adjacent larger category.*

I find this sentence very unclear. Do you mean that, in the absence of brittle fracture, the very large floes (that are not used in the comparison with observations) occupy a significant fraction of the ice covered area in the model, and that *prog-16-nobf* demonstrates that a breakup mechanism of these large floes is missing in the original prognostic model ?

Section 4.2:

To me, this section could be quite a bit shorter, given the few changes introduced by the FSD and shown in Figures 5,6,7. For instance, I am not sure that Figure 5 is really needed and even Figure 6 could be simply summarized in the text. As it is, I felt like I was reading details about why the addition of FSDs in models is relatively useless, which does not really help to convince me of the interest of this paper. In my view, there are some topics addressed in the Discussion section that would deserve more highlights than the Pan-Arctic impact of FSDs, which is less than a small change in the ice albedo for instance.

P11L34 *Previous studies e.g. Bateson et al. (2020) and Roach et al. (2018), have shown large FSD model impacts locally even where Pan-Arctic impacts are small.*
Exactly! So it might not be worth the price of 3 figures.

P11L19 *Bateson et al. (2020) demonstrated...*
They did, but so did Roach et al. (2018) and others before (It is reported in Tsamados et al., 2015, which already involve some of the co-authors of this manuscript).

P11L28: *The similar magnitude of change in the total melt also means that the results shown in Figs 8 and 9, where the sea ice volume is lower in both September and March for prog-best compared to WIPo-best, are unlikely to be driven by an increase in the total melt.*
This is a nice teasing of the discussion, but you should either discuss it here, or refer to the section where it is discussed, otherwise it is quite upsetting for the reader.

Section 4.3.3 is interesting. It could be worthy of a bit more in-depth analysis (particularly if section 4.2 is shortened). For instance, why do you see a relatively low $l_{eff}$ in the Chukchi/Siberian area for the prognostic model in March? Could you suggest what is driving this drop? L_eff in March with the WIPoFSD model is more like one would expect, with lower floe size found around the ice edge, where wave activity is strong. If the authors wanted to define a MIZ based on $l_{eff}$, they would likely find a pretty good agreement between this MIZ and the one based on wave-activity suggested in Horvat et al. (2020). That could be worth a mention, given the authors already refer to this study.
Results for the prognostic model also differ sensibly from the one shown in Figure 4 and 5 of Roach et al. (2018) manuscript. Could the authors suggest why?

*Figure 11 shows much higher spatial variability in $L\_eff$ for prog-best compared to WIPo-best. Further analysis (not presented here) indicates the high spatial variability in $leff$ for the prognostic model cannot easily be attributed to a single process but is particularly sensitive to the floe formation mechanism, brittle fracture scheme, and welding, all processes not explicitly represented in the WIPoFSD model. Processes included in the WIPoFSD model, such as wave break up of floes and lateral melt, are not found to have a large impact on the spatial distribution of $L\_eff$ within the prognostic model.*

I think this paragraph briefly addresses what is missing in the manuscript that, in my opinion, would really increase its significance. I don't know how far the authors can go in their analysis

with the simulations they already have, but it would really improve the paper to discuss the contribution of the different processes a bit more. This is particularly true for the brittle-fracture process. The manuscript in its current state sometimes sounds like a criticism of the prognostic model as used by Roach et al. (2018) but does not really show the impact of the changes they made on the model (except for the FSD in the summer).

P12L30 Section 5.1
I feel like a lot of things in this section are repeated but not necessarily developed. I have already given several comments about the brittle-fracture mechanism. I think a lot of the problems I have with this addition could be solved with more clarity in its presentation.

Section 5.3
P14L26: I think the performance aspect, even though it is quite short, is very important for people that would like to use FSDs in the future, but that are not necessarily experts. It could be nice to highlight this a bit more, maybe by starting this section with this topic. I think it would also be fair to refer to the preprint of Horvat and Roach (2021), as it tries to address some of the shortcomings of the prognostic model.

P15L1 Section 5.4:
Lots of points are discussed, but they are a bit all mixed. Maybe cut this sub-section into paragraphs to clearly show the structure of the argument.

P15L19 Conclusion

The conclusion is a bit long. I think the significance of the paper would appear more clearly with a better hierarchy in the importance of the findings developed in this manuscript. To me, it is not clear what are the most important results according to the authors.
The level of detail in this conclusion is too high, a lot of things are repeated (motivation for the inclusion of the brittle-failure, no improvement of the models at Pan-Arctic scale, utility of l_eff…) that could be removed in my opinion.

*Future work should focus on the development of a full physical treatment of the impact of brittle fracture on the FSD.*
Again, would it not be interesting to relate this with the work carried out on emerging brittle rheology models (e.g. Dansereau et al., 2016)?  Or to the work of Rynders that is already mentioned in the introduction? As it is, the paper does not really demonstrate the interest of using FSDs, which reduces its potential impact, at least in my opinion. Giving more context would highlight how the comparison made in this manuscript can contribute to the future of sea ice modelling.

Minor comments:

General:

I believe this is the first manuscript I have read that does not use a chronological order when citing 2+ references. This is not a big deal, but you might want to change that.

The image resolution of the figures seems in general quite low to me. This is purely aesthetics, but it gives a "draft" impression of the Figures.

Readability of graphs would also be improved with more ticks (Figure 4) or maybe a grid in the background (Figure 2,5,6,7,9).

Aesthetics again, but the authors should consider using roman (normal) text to subscripts in equations/variable names when they have more than 1 letter. It improves the readability.

I find the name "WIPoFSD" a bit hard to read (too long for an acronym, and not straightforward to pronounce). The authors might consider using a shorter name, or a name that would be related to a key property of this model (fixed-shape FSD model?).

Abstract:
P1L13→17: The beginning of the abstract would gain from being a bit more synthetic/sharper (until […] "*In this study*".

Introduction:
P1L36: The number of references for the mechanical response of sea ice to stress is quite high compared to the rest, given this is not the main topic of the paper. I acknowledge these references are relevant to this topic. My main concern is that as all these references are linked to only one team working on this topic, it gives a misleading picture of the field to the reader (see for instance the studies by Shen et al., 1986 a,b; Williams et al., 2017; Boutin et al., 2021…). As mentioned earlier, the link between this manuscript and these references could fit well in the conclusion, so maybe the authors should move some of these references there.

P1L42: I was a bit confused by the word "province". Whether it is correct or not, I would recommend using the word "region" that is clearer for international English speakers/reader.

P2L2: "*here*" is a bit unexpected given the introduction has just started.

P2L9: "*Note that all…*" This sentence breaks the flow of the introduction a bit. The authors might want to move it a bit earlier or find a smoother way of stating this fact (just a suggestion obviously).

P3L4: "…*FSD in the model is actively constrained according to observations, in this case by approximating the FSD as a power law.*"
All observations do not conclude that the FSD follows a power-law. Horvat et al., (2019) does not for instance.

P3L6: "*though with some dependency on model structure such as how the FSD is discretised over floe size categories.*"
This has been addressed in previous studies I believe (Horvat and Tziperman, 2015?), it could be nice to refer to them.

P4L12: I find the beginning of section 2.1.2 a bit confusing. I suggest starting with one sentence to summarize why the MLD matters in your CICE setup. The second sentence of this paragraph would make a better start for instance. The way section 2.1.3 is introduced is much clearer for instance.

P7L1: Could you remind the reader of what l_var is (physically)? There are a lot of floe size names in this paragraph, it is quite easy to lose the reader.

P7L9: "*The broader impacts of a power-law distribution on the sea ice cover can be explored whilst also including spatial and temporal variability of the FSD within the model. For mechanical processes such as wave break-up, the use of $l_{var}$ is particularly suitable*"
I find these sentences a bit vague. The link with the rest of the paragraph could be more explicit.

P7L13: "*For thermodynamic processes it makes less intuitive sense. It is not possible to define two clear regimes; instead, floes across the distribution reduce in diameter by the same magnitude in response to a lateral melting event. Here, we have modified the lateral melting scheme to calculate the change in $l_{eff}$ rather than $l_{var}$, since it is possible to calculate exactly how $l_{eff}$ would change in response to a given perturbation of the FSD.*"
Same comment here, I am afraid that a reader that is not familiar with FSD modelling would get quite confused. A few more details about the physical reasons behind these statements could improve the readability.

P7L28: The reference for the CPOM CICE could be given here instead of further in the text.

P7L34: It is likely a very naive question, but why do the authors use a winter climatology?

P9L21: Please give the spatial and temporal resolution of these datasets.

P9L25: Another (important) reason PIOMAS is used as a reference it that it has been carefully evaluated against available sea ice thickness observations. See for instance:
Schweiger, A., R. Lindsay, J. Zhang, M. Steele, H. Stern, and R. Kwok, 2011: Uncertainty in modeled Arctic sea ice volume. J. Geophys. Res., 116, C00D06, https://doi.org/10.1029/2011JC007084.

P9L36: It would help to add the panels of interest in the references to Figure 4.

P9L35: *"In particular, the slope of the distribution is much steeper (more negative) for the model output than observations."*
It would help to give a physical interpretation of this statement.

P9L40: *[…] a significant improvement in the ability of the prognostic model to capture the shape of the FSD for mid-sized floes.*
In summer.

P10L25: I am a bit confused by this *"However,"*.

P14L5→P14L13 *Therefore…* This is interesting, it would gain from being a bit clearer.

P15L15: *"reduce the sea ice mass balance"*
This expression is confusing. The comparison with the results by Roach et al. (2019) in the next sentence is also a bit unclear to me.

Caption of Figure 4: *month(s)*
Why is the "s" between brackets?
*"prog-16 performs particularly well in the Fram Strait and East Siberian Sea but less well for the Chukchi Sea. It represents a significant improvement to prog-16-nobf in all three locations."*
I do not think this comment should be part of the caption.

Caption of Figure 7:
*All three simulations generally lie within the range spanned by the observational products except for pack ice extent in March after 2010.*
I do not think this comment should be part of the caption.

Figure 9:
It would be better to use the same vertical scale, at least for the extent (a,c) and volume (b,d). As it is, it looks like differences between models are larger in March than in September.

Figure 12:
I was a bit confused by the use of the blue and red colormaps in section A, as these colors are later used to represent a positive/negative difference in sections B and C. I would recommend using the same colormap for all quantities that are not a difference, for instance the pink/purple colormap used for panels B(e,f) and C(e,f) could be used for all panels in section A.

*References:*

Boutin, G., Williams, T., Rampal, P., Olason, E., and Lique, C.: Wave–sea-ice interactions in a brittle rheological framework, The Cryosphere, 15, 431–457, https://doi.org/10.5194/tc-15-431-2021, 2021.

Dansereau, V., Weiss, J., Saramito, P., and Lattes, P.: A Maxwell elasto-brittle rheology for sea ice modelling, The Cryosphere, 10, 1339–1359, https://doi.org/10.5194/tc-10-1339-2016, 2016.

Horvat, C. and Roach, L. A.: WIFF1.0: A hybrid machine-learning-based parameterization of Wave-Induced sea-ice Floe Fracture, Geosci. Model Dev. Discuss. [preprint], https://doi.org/10.5194/gmd-2021-281, in review, 2021.

Rampal, P., Bouillon, S., Ólason, E., and Morlighem, M.: neXtSIM: a new Lagrangian sea ice model, The Cryosphere, 10, 1055–1073, https://doi.org/10.5194/tc-10-1055-2016, 2016.

Tsamados, M., Feltham, D., Petty, A., Schroeder, D., and Flocco, D.: Processes controlling surface, bottom and lateral melt of Arctic sea ice in a state of the art sea ice model, Philos. T. R. Soc. Lond, 373, 20140167, https://doi.org/10.1098/rsta.2014.0167, 2015.

Shen, H. H., Hibler, W. D., and Leppäranta, M.: On Applying Granular Flow Theory to a Deforming Broken Ice Field, Acta Mechanica, 63, 143–160, 1986.  a, b

Williams, T. D., Rampal, P., and Bouillon, S.: Wave–ice interactions in the neXtSIM sea-ice model, The Cryosphere, 11, 2117–2135, https://doi.org/10.5194/tc-11-2117-2017, 2017.

---

## Referee Comment (RC2)

The manuscript by Bateson et al., "Sea ice floe size: its impact on pan-Arctic and local ice mass, and required model complexity" compares two of the main approaches for incorporating floe size distribution into a sea ice model (both using CICE) with observations in two ways. The first compares the floe size distribution with new FSD estimates from satellite imagery; the second is an evaluation of Arctic sea ice mean state over approximately the past 3 decades. With the newness of these models and the community focus on their implementation, this work is well justified. The manuscript is generally easy to read and complete. However, I have a number of concerns about how the comparison has been completed, and the presentation of the results.

**Major comments:**
- **Journal fit**. This fundamentally is presented as a model evaluation study. Is The Cryosphere the appropriate venue for this? Would GMD perhaps be a better fit? If published in TC, the authors should more clearly address and center what new science is presented.
- **Comparison with FSD observations**: I'm fundamentally a bit hesitant that floe size distribution models in global climate models are yet at a point where we expect them to match with observations (from specific location) as many other, more rigorously tested and developed features of models, can still not do so.
  To phrase this as a question: Why do you expect the models to represent realistic floe size distributions at a given point? Do you think there are other model factors that this representation is sensitive to, such as thickness distribution? Please discuss how other factors might impact this comparison
- **Comparison of sea ice mean state**. It is worth noting that CICE, as most models, has had parameters largely tuned to best represent current state. As a result, the comparison of model with no additional tuning to observations (of sea ice extent and thickness) seems poorly motivated. Would we expect it to improve representation of mean state without tuning of other variables? Additionally, the implementation in a forced, standalone setup is likely to see less response in sea ice state, without the possibility of atmosphere and ocean feedbacks.
  What I think is more interesting is changes to sea ice mean state between models, which can suggest something about how different physics and processes relating to floe size feedback onto other sea ice characteristics. However, this is hard to see in a forced (rather than coupled) model, where feedbacks are limited. I think it is worth focusing on differences in the seasonal cycle and maps between the model which may impact these feedbacks, in the absence of coupled runs. Do these suggest improvement in how ice evolves? In short, I suggest that in the absence of additional, fully-coupled runs, the conclusions should be re-framed.
- I was left confused how what appears to be substantial, meaningful changes in sea ice thickness in Fig. 11 (5-50 cm across much of the Arctic) corresponds to almost no change in volume in Fig. 5. Perhaps it is just an interpretation error on my part, but the presentation needs to be clarified to illuminate whether there are meaningful changes in ice state, or not.

**Minor/specific comments:**

- P1, L1: It would be helpful to provide a brief introduction to the range of floe ice sizes and key processes (and why it is useful to capture it with a distribution, as is often done for thickness)
- P1, L16: I would suggest that the sentence beginning with "Observations show…" should go before the sentence beginning with "Large-scale…" on L15.
- P1, L36: Is cluster of sea ice into larger floes really a process impacted by floe size, or is it primarily a process in determining floe size?
- P1, L37-38: May be worth considering additionally/alternatively citing Keen et al., 2021, which summarizes CMIP6 sea ice models and show that most use some derivation of CICE or LIM, which all have the same lateral melting parameterization
- P1, L39: Floe size is also not considered in dynamics.
- P2, L5: Why? Need to briefly describe the floe size distribution to justify why a power law is used (i.e., that typically more small floes). Replace "…generally fitted to a…" with "…summarized by fitting to a…"
- P2, L17: Please note to what degree power law does/doesn't fit observations summarized. Is it a manner of convenience, or do observations support its use?
- P2, L23-27: as noted above, it would be helpful to mention these processes earlier in introduction
- P2, L28: It would be helpful to include a transition sentence motivating introduction of brittle facture – that it is missing in most models, and may be important. Perhaps something like what is currently L42-43.
- P3, L3 and L4: replace "represents" with "is in"
- P3, L32: Would be helpful to also introduce the ITD, which is referenced in relation to the prognostic model
- P4, L11: Is there any possibility for ice-ocean feedbacks, such as albedo feedback, in this setup? Please specifically address in the text
- P4, L24: It would be helpful to be more clear in the description of CICE and general model that this is being used in a standalone setup.
- P4, L39: What is the L used in standard implementation? Is it 300 m, as used in lateral melt? Please define.
- P5, L19: In the introduction you present the WIPoFSD before the prognostic model. It would be helpful to be consistent about the order throughout manuscript. I suggest presenting WIPoFSD, then prognostic model, the brittle fracture scheme.
- P6, L31: change to "spatial and temporal scales" or "spatial scale and timescale" or similar
- P7, L1, 11, etc.: I'm not really sure I understand the physical implications of l_var. What is it intended to represent? What are the implications for observational comparisons?
- P7, L36: Is there anything that can be referenced to demonstrate that ERA-Interim wave product is reasonable to use for the Arctic? What is the treatment for waves in sea ice?
- P8, L9: Why 'best'? Unless you plan to show others runs involved in selection process, I suggest use of just 'WIPoFSD' and 'prog' for simplicity and clarity
- P8, L27: Perhaps simply "FSD observations" as title?

- P8, L29-30: I believe since this is included in contributions and acknowledgements, it is not necessary to include funding or names of contributors here.
- P8, L31: replace 'samples' with images
- L31: remove "three months"
- L39-40: What is the impact of this (as well as lower cutoff, L43+)? It seems like you could instead include the largest floes in the largest category, which may sometimes show a FSD with the uptick demonstrated by the prognostic model.
- P9, L11: change "is not novel" to "has been used previously"
- P9, L27: Is there an appropriate reference for this statement?
- P10, L8: Perhaps simply "Comparison of sea ice extent and volume" might be a clearer heading
- P10, L1-3: It would be helpful to show some results of when and where the brittle fracture scheme is implemented. Where is it most necessary? What does this suggest about what it corresponds to physically?
- P10, L 18-20: I think it is necessary to clarify here that these runs are done in a standalone setting, and that the possibility for feedbacks in a fully coupled climate model may give different results.
- P10, L23-24: It's not clear what negative trends in the percent difference suggest. Does this suggest some sort of feedback in model?
- P13, L14-16: Maps show more substantial changes in representation of sea ice state. Do these suggest improvements?
- P14, L34: It may be worth mentioning that this is particularly relevant in a standalone sea ice model, as run here. In a coupled context (for which climate models are often used) the sea ice model is typically a small component of the total cost, and so the additional cost from the FSD is relatively not substantial.
- P15, L6: What is meant by 'in-ice wave scheme'? Is it more accurate to say that the waves are forced with reanalysis?
- P15, L19: Future work to address the impact on Antarctic sea ice representation and comparison with observations may also be useful
- Figure 1: I find this figure really hard to interpret currently. A few reasons/suggestions… It might be better to use more realistic 'floe diameter' bounds, or to remove numbers from y axis, as it currently is hard to interpret these as actual bins. Only 2 examples are needed showing where redistribution is applied and where it isn't (for example, far left and far right). For one where redistribution is applied, show the new floe size distribution resulting more clearly. It would also be helpful to add lines showing actual density gradient for comparison to dashed purple line.
- Figure 2: I'm not convinced that this is necessarily a "non-physical feature" of the model, as it is simply capturing floes that are potentially beyond the bounds here, and is not reported as such in Roach et al., 2018. A comparison to observations without largest floes removed may be helpful to show if this is ever observed in observations. Additionally, please add a label to this figure demonstrating that it is only for areas of SIC 15-80%

- Figure 3: Are these exact bounds of model areas? If not, a different symbol may better communicate that, as the boxes suggest that this is the exact selection of grid cells.
- Figure 4: Again, not necessary to name co-authors in the text.
- Figure 5: The change of prognostic models being compared is a bit confusing. Is it worth including other prognostic models somewhere as well, to show if/that there is little difference in sea ice state? Also, as the 'prog-best' doesn't include brittle fracture (right?) it would be helpful to note that in the short name for clarity.
- Figure 5-7: This feels like a lot of plots to show for almost no change between any of the models. Can this be simplified to one or two key panels, and then state there is no observable change in others?
- Figure 8: I might suggest to swap these plots around to show both models and same subplot, with top for sea ice extent, bottom for volume. This would then allow to show some comparison in difference of observations from reference. (e.g., Are model changes moving it in the right direction?)
- Figure 9: I am unclear how this figure is different from what is shown in Fig. 6, in terms of the take-aways. How do we know if this is improving the comparison if the scale of change is not comparable to the difference from observations?
- Figure 11: I think this is the most useful and interesting plot! But, I'm quite confused how what appears to be substantial changes in sea ice thickness in A(f) agree with what is in Fig. 5 – where almost no change is observed. Some thoughts: Could the difference in fractional ice area/thickness compared to observations also be shown? It would be helpful to place the effective floe size upfront (at the top) to set it apart from differences, and also make this more clear in the figure caption (meaning, that floe sizes are NOT a difference).
- Figure 12: I'm not sure what to take from these plots, based on the units show. Would it be more helpful to show standard deviation as a percent of mean value? Also, it would be nice to be consistent with the months shown in Fig. 11.
- Table 1: "CPOM-CICE" is not needed in model description, as all are the same. It might be helpful to separate technical details into finer resolution categories, such as brittle fracture (yes/no), # floe size categories; d_min/d_max (where applicable)

**References:**

Keen, A., Blockley, E., Bailey, D. A., Boldingh Debernard, J., Bushuk, M., Delhaye, S., ... & Wyser, K. (2021). An inter-comparison of the mass budget of the Arctic sea ice in CMIP6 models. *The Cryosphere*, *15*(2), 951-982.

---

## Author Comment (AC1)

In this document we have compiled our response to both referee and editor comments for the manuscript entitled, 'Sea ice floe size: its impact on pan-Arctic and local ice mass, and required model complexity'.

This document includes the following:

- Response to referee 1, pg 2 17
- Response to referee 2, pg, 18 28
- Response to editor, pg 29 33
- Updated manuscript, pg 34 72

In this document referee / editor comments are shown in black, our response is in blue, and descriptions of changes to the manuscript are shown in red. Red is also used to indicate where the updated manuscript has been modified from the original submitted manuscript.

**Response to Anonymous Referee #1**

**Firstly, we would like to thank the reviewer for providing such thorough and thoughtful comments on our manuscript. They have been invaluable in improving the manuscript.**

This is a review of the manuscript entitled Sea ice floe size: its impact on pan-Arctic and local ice mass, and required model complexity. In this manuscript, the authors investigate the differences between two ways of representing the floe size distribution (FSD) in sea ice models. The first approach uses a prognostic floe size model, in which floe size evolves freely depending on some physical processes (breakup, lateral melting, welding...). The second approach is simpler, as it constrains the floe size distribution to always obey a truncated power-law. Only the upper-limit of this power-law varies with processes affecting the floe size. After having described the two models and their implementation in a stand-alone version of the sea ice model CICE, the authors evaluate the simulated floe size distribution against available observations in the summer. They show that the original prognostic model leads to unrealistic results in the absence of a process able to break the largest floes considered in their FSD. They suggest this process corresponds to brittle fracture of the ice, and that it can be represented by relaxing the FSD in the prognostic model towards a powerlaw. They further investigate the impact of the two FSD models on sea ice extent and volume in Pan-Arctic simulations. They find little evidence of any significant improvement of model results related to the addition of FSDs in the sea ice model. They discuss the differences between the two FSD models, as well as their advantages and drawbacks. The manuscript is overall easy to read, and the method followed by the authors is rigorous and well explained. I acknowledge the quality of the work that has been done, but I think there are a few problems to fix before I can support the publication of the manuscript.

**General Comments:**

The manuscript is in general well-written and not very long, however I find that some topics are repeated and too much time is spent describing results that are, in my opinion, not key to the study. I find it particularly detrimental to the potential impact of the study, as it makes the paper confusing in places, and the interesting findings and discussions are a bit lost among things that have already been long discussed in previous studies. I think there is potential for this manuscript to address the whole sea ice community, but in its current state I don't believe anyone not familiar with FSD modelling would get what the key findings are. I will try to highlight these problems and suggest ways of improving the manuscript in my specific comments.

My second main comment concerns the brittle fracture mechanism. This process occupies a large amount of the manuscript, however I am not fully satisfied with the way it is presented and discussed. I have the feeling (but I might be wrong, in which case I am sorry) that the authors found out during their evaluation against observations that a mechanism was missing in the prognostic model to break the largest floes into floes of "mid-range" sizes. They realized that observations were showing a power-law behaviour, and therefore improved the prognostic model by adding a relaxation towards this power-law. To explain this behavior, they suggest that brittle-facture is a good candidate, even though it has not really been demonstrated before for this spatial scale. However, the way it is presented in the paper is confusing, mixing LKFs in pack ice, fragmentation by waves and scientific intuition. The study does not do enough to justify the use of this relaxation in the winter in my opinion. I also find the discussion about this process a bit shallow, particularly as it is a major change compared to the prognostic model used in Roach et al. (2018). My recommendations would be a) to present this new process more carefully, i.e. introduce it as a relaxation of the prognostic FSD

towards a power-law, b) motivate this introduction by the fact that the prognostic model fails to reproduce observed FSDs in the summer without this relaxation, c) discuss what this relaxation represents (and I agree with the authors that brittle-failure is a good candidate), and when and where it should be applied. I will also explain these recommendations and my criticism in more detail in the specific comments.

Thank you for these overall comments regarding the manuscript. To address the concerns regarding the potential impact of the study, we have made significant edits to both the discussion and conclusions to highlight the novel aspects of this study and to clarify why the results are relevant to the broader sea ice community. We have also moved the motivation for the brittle fracture scheme out of the introduction and into section 2.2.2 as suggested in the specific comments. To address concerns regarding the brittle fracture scheme, we have made significant edits to section 2.2.2, including several new references, to more clearly explain and motivate why we introduced this scheme and the model choices made. This includes a more careful explanation of the physical interpretation of the restoring timescale used in the scheme and the limitations of this scheme. We have also reviewed how brittle fracture and the brittle fracture scheme are presented in the results, discussion, and conclusions in respect of the modifications made to section 2.2.2. Full details on the modifications to the manuscript described above can be found in the response to the specific comments below.

**Specific comments (major):**

P2L28 to 44. This is the first introduction to the brittle-fracture process. It is quite long, and for a reader that has not read the full paper yet, I believe the relationship with the rest of the introduction is very obscure. I also believe this level of detail would fit better in section 2.2.2. The paragraph first explains in detail the mechanism of LKF formation in pack ice, then switches to Perovich et al. (2001) that relate floe breakup to melt as thin ice is very weak, and finally refers to Kohout et al. (2016) who report flexural failure (and not in-plane failure) of ice under wave action in places where ice strength is minimal. I think I get that the authors want to say that processes responsible for LKFs at scales >1km might affect floe size at scales <1km, and that heterogeneity in the ice strength/thickness exists at these scales that would ease brittle fracture, but this does not appear explicitly in the text. I also think the mechanical behaviour of sea ice depending on the spatial scale of interest is an open question, which should appear more clearly in the text. Details about the spatial scales discussed in each reference and the one of interest for the study are missing.

As suggested, the full introduction to the brittle fracture process has been moved to section 2.2.2 (see the first two paragraphs in section 2.2.2 within the updated manuscript). The text has also been modified to address the comments above including providing details of the spatial scales considered in the references, a discussion of the scale variability of brittle fracture processes in sea ice, and that gaps in understanding remain within this topic.

I also find the conclusion ("These observations and model studies collectively suggest that brittle fracture processes impact floe size in winter and that the resulting pattern of linear features resulting from brittle fracture may also influence floe breakup in the subsequent melt season.") quite far-fetched and not well motivated with these references. Unless the manuscript addresses sea ice rheology at floe scale (which is not the case to me), I find this statement too strong to belong to the introduction.

In this introduction, I would recommend only mentioning the fact the prognostic model used by Roach and others has not been thoroughly evaluated against FSD observations, and that some processes might be missing. For instance, brittle fracture of ice might occur as the ice is thinning in the summer (Perovich et al., 2001).

We have moved the paragraph that discusses brittle fracture to section 2.2.2. Instead, as suggested, we add the following to the existing paragraph discussing the state of FSD modelling:

'The limited spatial and temporal coverage of floe size observations has prohibited effective evaluation of these models, though there have been recent efforts to develop satellite derived FSD product to enable such evaluations (Horvat et al., 2019). It is nevertheless anticipated that important processes are not yet represented in these models. For example, thermodynamically-driven breakup of floes along existing cracks and refrozen leads in the sea ice cover (Perovich et al., 2001).'

P4L4->L10: The lateral heat flux...

I find this paragraph a bit hard to follow:

a) I don't understand why it is important to explain how heat fluxes are dealt with in CICE here. Please inform the reader of their use in this study. Also, it would be nice to highlight if this is a change from the "standard" in CICE, or if these are all default settings (as is done well further in the text for other code changes).

One of the two ways in which CICE has been adapted to use an FSD model is by modifying how lateral melt is calculated (the other is the calculation of momentum exchange between sea ice, ocean, and atmosphere via the form drag scheme). As such, we believe it is important to provide complete details of how lateral melt is treated in standard CICE, to make it clear how this treatment is different in the modified setup.

The final sentence of the first paragraph in section 2.1.1, 'An overview of lateral melt treatment within CICE is presented here.' is replaced with, 'Below we provide an overview of features of standard CICE that are pertinent to the evaluation of the lateral melt volume. In section 2.1.4 we will explain how this standard treatment is adapted for use with an FSD model.'

Additional sentences have also been added to the introduction and the opening of section 2 to add clarification regarding the importance of the lateral melt treatment in this study.

b) F\_frzmlt is computed as...The authors might want to write the equations instead of describing them in the text, that would likely improve the readability. Manuscript updated as suggested.

c) Why is F\_frzmlt capped, and why is it important?

This is a component of the standard CICE formulation for calculating the lateral and basal met volume. We have mentioned this as it is a component to how lateral melt volume is calculated in the standard CICE model. This cap is sufficiently large to be effectively immaterial to any calculations. The modifications made to point (a) above also apply here (i.e. clarifying the importance of explaining the treatment of lateral melt in standard CICE).

P5L12: Please clarify the definition of I\_eff. What do you call "the perimeter of a FSD"? Original definition, 'I\_eff is the floe diameter that has the same perimeter per unit sea ice area as a given FSD', has been replaced with, 'I\_eff is the diameter of the set of identical floes that has the same total perimeter as a set of floes of variable size with the same total ice area'.

P5L16: You might want to explain briefly why l\_eff is better than the average floe size (I believe this is mentioned somewhere else in the text, but it would fit nicely here). The following clarification is added: 'l\_eff is applicable here because the lateral melt volume is proportional to the total floe perimeter.'

P5L28: Note that the in-ice... I don't understand this sentence. Could the authors clarify the difference between their parameterization and the one by Roach et al. (2019)? The following section:

'Note that the in-ice wave scheme used by Roach et al. (2018) has been adapted here to calculate H\_m0, the spectral height parameter, and  $\lambda_p$ , the wavelength corresponding to the peak wave

energy, within the sea ice-covered grid cells for use with the wave-dependent floe formation parameterisation.'

Has been replaced with:

'Unlike Roach et al. (2019), we do not use a separate wave model coupled to CICE to calculate the necessary wave properties within the sea ice-covered grid cells for use with the wave-dependent floe formation parameterisation. Instead, we adapt the scheme used in Roach et al. (2018), which calculates in-ice wave properties using an extrapolation from forcing external to the sea ice cover, to calculate the necessary in-ice wave properties.'

**Section 2.2.2:**

For a reader that has not been through the Results section, the motivations behind the addition of this process remain very obscure. I think part of the motivations currently in the introduction (e.g Perovitch et al., 2001) would fit nicely here. Hinting at the results a bit would also help. Stating that the prognostic model was found to fit poorly with observations without this model would really help to understand the addition of this process. The authors should at least reassure the reader by saying that they are going to investigate the effect of this addition by running two experiments, one with this modified prognostic model, and one without it.

We have moved an updated version of the motivation of the brittle fracture scheme from the introduction to section 2.2.2 (first two paragraphs). We have also included the following sentences at the start of section 2.2.2 to explain the need to introduce new model physics to the prognostic model:

'It will be shown in section 4.1 that the prognostic model struggles to capture the shape of the observed FSD for mid-sized floes. Sensitivity studies show that it not possible to modify existing parameterisations in the prognostic FSD model to substantially improve model performance against observations (Bateson, 2021). This suggests there are important processes currently not represented within the prognostic model.'

The section also now concludes with the following line:

'Results will be presented in section 4.1 to demonstrate that the inclusion of this new brittle fracture scheme significantly improves prognostic FSD model performance against observations in simulating FSD shape for mid-sized floes.'

This condition means... It could be worth giving the physical interpretation of this sentence, as not all readers are familiar with FSDs.

The following clarification has been added: 'i.e. only when the ratio of larger floes to smaller floes exceeds a given value'

Fracture events occur regularly through autumn, winter and spring within the pack ice to form linear features like leads, which subsequently freeze up again...

I find the current discussion quite vague, and not very well linked to other references in the literature. The way I understand it, the authors assume that at subgrid scale, leads create a network of cracks that define floes in a kind of mechanical strength sense, even though they are separated by thinner ice and not open water. Distribution of these floes is assumed to follow a power-law of exponent -2 as it results from successive fracture events. Between these floes there are therefore weak joints made of thinner ice that will melt faster in the summer than the surrounding ice, to the point where their strength will become very weak (Perovitch et al., 2001). The idea here is therefore that sea ice has a memory of fracturing events, is this correct? If so, this is not so different from the definition of damage in brittle rheologies (cf. the work of Weiss that is already cited, and the models of Dansereau or Rampal, see references at the end of the review), and it could be worth commenting on that, if not here, maybe in the discussion. If not, the text needs to be clarified to make clearer what

the actual point being discussed is.

The summary above is indeed consistent with our thinking of the purpose of the brittle fracture scheme. We have made updates to the relevant text to make these points clearer. The relevant paragraph in section 2.2.2 now reads as follows:

'A value for the restoring timescale,  $\tau$ , needs to be determined. Both direct and indirect mechanisms have been discussed above describing how brittle fracture can impact the sea ice cover. Fracture events occur regularly through autumn, winter and spring within the pack ice to break up floes and form features such as leads, though these generally freeze up again. The result of these fracture events is to create a network of linear features that define weaker regions of ice interspersing stronger ice. Idealised models of brittle fracture suggest that the size distribution of the stronger regions of ice follow a power law with an exponent of -2. The linear features are then vulnerable to increased thinning and melting, increasing the likelihood of break-up along these features during late spring and summer as the sea ice retreats. This effectively 'releases' the floe size distribution defined during brittle fracture events outside of the melt season. It is this second mechanism that is of more relevance when considering the impacts of the FSD on the seasonal retreat of the Arctic sea ice.'

We thank the reviewer for highlighting the similarities between the brittle fracture mechanism suggested and the concept of damage in brittle rheology. This is an interesting comparison and worth noting in the manuscript.

We have added the following discussion to the end of section 5.1:

'An interesting comparison can be made between the treatment of brittle fracture within the prognostic model presented here and recent developments introducing the concept of 'damage' to the treatment of rheology within sea ice models (Dansereau et al., 2016). One such sea ice rheology, named the Maxwell-elasto-brittle (Maxwell-EB) rheology, has been applied within the continuous and fully Lagrangian sea ice model neXtSIM (Rampal et al., 2019). This new rheology retains a 'memory' of any fracture events, effectively tracking how 'damaged' the sea ice cover is, and modifies the sea ice properties accordingly. This concept of 'damage' has clear parallels with the mechanism discussed above of how winter in-plane brittle fracture events can determine how the sea ice breaks-up in summer and may therefore present a useful basis for the development of a full parametrisation of brittle fracture processes for use in FSD models. Boutin et al. (2021) also demonstrated that the Maxwell-EB rheology can be combined with an FSD model in order to explore how wave break-up of floes can impact sea ice dynamics, highlighting an application of floe size modelling not considered in this study.'

Freezing/floe welding/convergence will act as mechanical healing that will erase any memory of fracturing events, in the winter at least (e.g Rampal et al., 2016). How would it compete with brittle failure in your model? This point is quite important to me, as Roach et al. (2018) present how FSD processes are balanced in the prognostic model over the year, and here a modification is introduced that likely breaks this balance. In a sense, it also relaxes the prognostic model towards a truncated power law with upper limit the maximum floe size in the model, does it not? This goes against what I think is the original philosophy of the freely evolving FSD of the prognostic model but this fact is not really discussed here.

We assume when you refer to the 'balance' of the model you mean that the average ratio of small floes to large floes stays roughly constant over large timescales and there is not an imbalance in processes such that the model tends excessively towards floes that are too small or too large. Roach et al. (2018) demonstrated this by including some interesting and important figures demonstrating the tendency (both scale and sign) of individual processes in changing the total sea ice area within each floe size category. Obviously the introduction of a new process will perturb the balance and can influence the magnitude of other processes that impact the FSD such as the welding of floes, but in this case we can be confident that a reasonable 'balance' still exists since we compared model output to observations and the effective floe size metric shows significant spatial and temporal variability (which would not be the case if very small or very large floes dominated the distribution). Regarding the second point, the FSD in the prognostic model including brittle fracture is still able to freely evolve since the brittle fracture scheme is applied at the scale of floe size categories rather than over the whole distribution. Whilst the brittle fracture scheme does, of course, influence the emergent FSD shape, sensitivity studies (not presented here but can be found in chapter 7 of Bateson, 2021) performed using the prognostic FSD model with brittle fracture show other processes continue to have a significant and comparable influence on the emergent FSD shape.

However, the brittle fracture-derived mechanisms operate over different timescales and scale with different properties. à This is a very vague sentence.

This point has been significantly expanded upon in section 2.2.2 to clarify the point being made and also to reflect some of the other edits made to this section (this should also address the point below about the value of tau in summer vs winter):

'The use of a fixed timescale makes it difficult to capture both the direct mechanism of brittle fracture impact on floe size, which dominates outside of the melt season, and the indirect mechanism via thermodynamic weakening, which is more important within the melt season. The latter mechanism has been prioritised in this case in determining the timescale given it is the FSD state in the melt season that is of primary importance for understanding FSD impacts on the Arctic sea ice (Bateson et al., 2020). Just considering the thermodynamic weakening mechanism, the use of a fixed timescale is still a simplification given the significant spatial and temporal variability of relevant factors to this mechanism such as melt rates, ice strength, and dynamic forcing.'

The timescale for a crack or linear feature in the sea ice to fully melt through is taken to be of the order of 1 month. For simplicity,  $\tau$  is here set to 30 days.

I find it very confusing to relate the time scale of brittle fracture, which is a dynamical process occurring in very little time, to the time scale of sea ice melt. To me 30 days is not related to the fracture itself, but to the reduction of ice strength that tends towards 0 as ice melts, making it very sensitive to brittle failure. It would also be nice to a provide more quantitative details to the reader: what thickness/melt rate are you considering here to end up with tau=30days?

The discussion of  $\tau$  in section 2.2.2 has been modified to clarify what it is supposed to represent and to provide qualitative details, as suggested:

'In this context,  $\tau$ , the restoring timescale, refers to the timescale for the sea ice to thin sufficiently that the sea ice is vulnerable to in-plane fracture events along existing weaknesses. Sea ice thickness away from the ice edge at the start of the melt season is generally in the range of 1 - 3 m. Vertical melt rates are of the order of 5 - 15 cm day-1. Therefore, significant thinning can generally occur over timescales as short as a week up to a couple of months. For simplicity,  $\tau$  is here set to 30 days.'

Also, to me, this justification does not motivate the addition of the brittle-fracture process in the freezing season. With this motivation, tau should tend towards infinity in the winter. The lack of discussion about this process in winter is particularly detrimental to the study as modelled FSDs are only evaluated in the summer (or at least melting season) in section 4.1. The seasonal impact of this relaxation process should be discussed: does it overwhelm the floe size growth process in freezing conditions, or is it negligible?

As mentioned in the manuscript, this treatment of brittle fracture is supposed to be a simple approximation that has been included since, without this process, the prognostic model struggles to capture the shape of the distribution for mid-sized floes. As such, there are significant limitations with the treatment, including the use of a fixed tau throughout the year. We chose not to include these results here since the focus of the paper is intended to be the comparison between the two FSD models, but in Chapter 7 in the thesis from Bateson (2021), a series of sensitivity studies are presented with the version of the prognostic model used here including brittle fracture,

demonstrating that the new brittle fracture scheme does not dominate the shape of the distribution and other processes continue to influence FSD shape, including winter floe growth processes. This is an important point to at least mention within the manuscript, however.

The following comment has been included in the relevant discussion section i.e. 5.1: 'However, sensitivity studies show that the brittle fracture scheme does not dominate the shape of the emergent FSD and other processes continue to be important in the evolution of the FSD, particularly winter growth processes such as floe formation and welding (Bateson, 2021).'

**Section 4.1**

P9L39: Overall, the inclusion of the quasi-restoring brittle fracture scheme represents a significant improvement in the ability of the prognostic model to capture the shape of the FSD for mid-sized floes.

I agree, but you have only shown it in summer.

'Over the period May – July' has been added to the end of this sentence.

P9L42: [...] does not include floes smaller than 100 m in the comparison, which are particularly important for determining the impact of the FSD on the sea ice mass balance.

These floes are important for lateral melting, but is lateral melting important for the mass balance? This is not what I retain from this manuscript, or from studies such as Bateson et al. (2020) and others, at least not for the Pan-Arctic mass balance.

Several studies, including Bateson et al. (2020, see case B in table 3) and Smith et al. (2022) have shown that where an FSD is dominated by smaller floes or where the fixed floe size is of the order of metres, the change in mass balance can become significant (orders of 10% change or larger in summer). However, we recognise that in the context of this study, it would be more appropriate to highlight the sensitivity of local sea ice properties rather than pan-Arctic properties, since we do find larger impacts on these scales.

'on the sea ice mass balance' is replaced with 'on sea ice concentration and thickness'.

P10L5: Whilst a reduction in ice area fraction in the largest category and an increase in the smallest category can be expected, the change in ice area fraction in the remaining categories depends precisely on the balance between ice area fraction lost from that category and ice area fraction gained from the adjacent larger category.

I find this sentence very unclear. Do you mean that, in the absence of brittle fracture, the very large floes (that are not used in the comparison with observations) occupy a significant fraction of the ice covered area in the model, and that prog-16-nobf demonstrates that a breakup mechanism of these large floes is missing in the original prognostic model ? Intuitively, it would be expected that more breakup -> less larger floes and more smaller floes -> steeper negative gradient, but the results here show the reverse. The point being made here is that this intuitive response does not apply to a distribution since, non-withstanding the smallest and largest categories, all remaining categories have both a source and sink of floe area. The following clarification has been added at the end of this section: 'In this case, the presence of the 'uptick' shown in Fig. 2 for the prognostic model without brittle fracture results in the source of floe area being larger than the sink for most floe size categories and a net reduction in gradient overall from including brittle fracture.'

**Section 4.2:**

To me, this section could be quite a bit shorter, given the few changes introduced by the FSD and shown in Figures 5,6,7. For instance, I am not sure that Figure 5 is really needed and even Figure 6 could be simply summarized in the text. As it is, I felt like I was reading details about

why the addition of FSDs in models is relatively useless, which does not really help to convince me of the interest of this paper. In my view, there are some topics addressed in the Discussion section that would deserve more highlights than the Pan-Arctic impact of FSDs, which is less than a small change in the ice albedo for instance.

P11L34 Previous studies e.g. Bateson et al. (2020) and Roach et al. (2018), have shown large FSD model impacts locally even where Pan-Arctic impacts are small.

Exactly! So it might not be worth the price of 3 figures.

The purpose of this manuscript is to provide a full comparison and assessment of the impact of the two approaches to modelling the FSD on the sea ice cover, and we believe that these figures provide important context for the comparison, even if what they show is a 'null result' in terms of the impact of either FSD model being significant for the considered metric. However, given the present length and large number of figures within the paper, we recognise the need to remove at least one figure. As such, we have decided to removed Fig. 6 from the manuscript, since we believe that this figure offers the least additional insight of Figs 5-7.

Figure 6 and any references to this figure have been removed from the manuscript. Figure numbering has been updated accordingly.

P11L19 Bateson et al. (2020) demonstrated...

They did, but so did and others before (It is reported in Tsamados et al., 2015, which already involve some of the co-authors of this manuscript).

Both Roach et al. (2018) and Tsamados et al. (2015) identified that the increase in lateral melt was compensated by a reduction in basal melt, but only Bateson et al. (2020) went on to show that this effect could be primarily attributed to the physical reduction of sea ice area in locations of high basal melt.

The relevant two sentences have been updated with additional references:

'Several previous studies, including Tsamados et al. (2015) and Roach et al. (2018), found that increases in the lateral melt volume resulting from higher floe perimeter were compensated by a reduction in the basal melt. Bateson et al. (2020) demonstrated that this compensation effect was shown to primarily be a result of the physical reduction of sea ice area in locations of high basal melt.'

P11L28: The similar magnitude of change in the total melt also means that the results shown in Figs 8 and 9, where the sea ice volume is lower in both September and March for prog-best compared to WIPo-best, are unlikely to be driven by an increase in the total melt. This is a nice teasing of the discussion, but you should either discuss it here, or refer to the section where it is discussed, otherwise it is quite upsetting for the reader. The following additional comment has been added: 'This point will be discussed further in section 5.2.'

Section 4.3.3 is interesting. It could be worthy of a bit more in-depth analysis (particularly if section 4.2 is shortened). For instance, why do you see a relatively low I\_eff in the Chukchi/Siberian area for the prognostic model in March? Could you suggest what is driving this drop? L\_eff in March with the WIPoFSD model is more like one would expect, with lower floe size found around the ice edge, where wave activity is strong. If the authors wanted to define a MIZ based on I\_eff, they would likely find a pretty good agreement between this MIZ and the one based on wave-activity suggested in Horvat et al. (2020). That could be worth a mention, given the authors already refer to this study.

Section 4.3.3 has been expended to include the following comments, as suggested above: 'A good case study is the relatively low I\_eff seen in the Chukchi Sea during March and June. The floe formation mechanism is important to FSD evolution in this region of the Arctic since it experiences ice-free conditions for at least part of the year. Higher wave activity is also expected in this region due to an increased fetch via the Bering Strait, and this will increase the proportion of floes that form in smaller floe size categories. Other regions that experience ice-free conditions are generally more sheltered from wave activity due to adjacency to continental land mass. The only comparable regions in terms of wave exposure are the Greenland Sea and Barents Sea, where lower values of I\_eff can also be seen.'

**And:**

'One point of interest here is the regions of reduced I\_eff shown for the WIPoFSD model appear to correspond well with the MIZ defined using wave activity presented in Horvat et al. (2020), which suggests that a possible application of FSD models would be an alternative way of defining the MIZ compared to the sea ice concentration-derived definition.'

**Results for the prognostic model also differ sensibly from the one shown in Figure 4 and 5 of Roach et al. (2018) manuscript. Could the authors suggest why?**

Regarding the differences with Fig. 4 and 5 in Roach et al. (2018), the leading order difference appears to be that our setup shows a reduction in sea ice thickness across the sea ice cover rather than some regions of decrease and some regions of increase as shown in Roach et al. (2018). This makes sense given we have introduced the brittle fracture scheme, which will have a net effect of reducing the effective floe size and therefore increasing the lateral melt rate. It is difficult to accurately compare the magnitude of the changes due to the very different scales used by the two sets of figures, but broadly they are of comparable order.

Figure 11 shows much higher spatial variability in  $l\_eff$  for prog-best compared to WIPo-best. Further analysis (not presented here) indicates the high spatial variability in leff for the prognostic model cannot easily be attributed to a single process but is particularly sensitive to the floe formation mechanism, brittle fracture scheme, and welding, all processes not explicitly represented in the WIPoFSD model. Processes included in the WIPoFSD model, such as wave break up of floes and lateral melt, are not found to have a large impact on the spatial distribution of  $l\_eff$  within the prognostic model.

I think this paragraph briefly addresses what is missing in the manuscript that, in my opinion, would really increase its significance. I don't know how far the authors can go in their analysis with the simulations they already have, but it would really improve the paper to discuss the contribution of the different processes a bit more. This is particularly true for the brittle fracture process. The manuscript in its current state sometimes sounds like a criticism of the prognostic model as used by Roach et al. (2018) but does not really show the impact of the changes they made on the model (except for the FSD in the summer).

The referenced paragraph has been edited in response to a previous comment to explain the region of reduced effective floe size in the Chukchi Sea in March and June. Whilst this is a single case study it does at least provide further details of how individual processes (in this case the floe formation mechanism) can influence the spatial distribution of the effective floe size. We appreciate the point that it would be interesting to explicitly evaluate the contribution of individual processes, but this has been a focus of previous papers (Roach et al., 2018, in the case of the prognostic model, and Bateson et al., 2020, in the case of the WIPoFSD model). Our focus in this study is to compare and discuss two alternative methods of modelling the FSD and given the significant differences in model structure and how individual processes are represented within the model we wanted to keep the focus on overall model performance and impacts. It is worth noting that chapter 7 in Bateson (2021) presents a series of sensitivity studies using the prognostic model (including the brittle fracture scheme) that explores the role of individual processes extensively but, as the length of this chapter should indicate, a complete presentation and discussion of these studies would be a paper in its own right. To be clear, the intention of this paper was never to be critical of the prognostic model as presented in Roach et al. (2018). We have added a comment to section 4.1 that we hope should make clear the purpose of evaluating FSD model performance against observations of floe size.

The following comment has been added to section 4.1:

'The purpose of this comparison against floe size observations is to ensure that the WIPoFSD and prognostic model setups used in this study perform comparably well to the same dataset. There are limitations to this evaluation of model performance, however. In particular, floes smaller than 100 m or larger than 2 km are not considered for the reasons outlined in section 3.2, and the former are especially significant for determining the impact of a given FSD on sea ice concentration and thickness.'

**P12L30 Section 5.1**

I feel like a lot of things in this section are repeated but not necessarily developed. I have already given several comments about the brittle-fracture mechanism. I think a lot of the problems I have with this addition could be solved with more clarity in its presentation. Significant modifications have been made to this section to reflect the modifications made to section 2.2.2 in response to earlier comments. The points made in this section have also been developed to better explain their relevance.

**Section 5.3**

P14L26: I think the performance aspect, even though it is quite short, is very important for people that would like to use FSDs in the future, but that are not necessarily experts. It could be nice to highlight this a bit more, maybe by starting this section with this topic. I think it would also be fair to refer to the preprint of Horvat and Roach (2021), as it tries to address some of the shortcomings of the prognostic model.

The relevant paragraph has been moved to the start of the section as suggested. This section also now references the suggested preprint:

'It is worth noting that future advancements in modelling techniques may reduce or mitigate the computational expense or complexity of either model e.g. Horvat and Roach (2022) presented a machine-learning-based parameterisation to simulate wave break-up of sea ice floes that can replace the existing treatment of wave break-up in the prognostic model. The study found that CICE simulations including the prognostic model with this new parameterisation have an approximately 40% longer run time than CICE simulations without the prognostic model i.e. a comparable cost to the WIPoFSD model.'

**P15L1 Section 5.4:**

Lots of points are discussed, but they are a bit all mixed. Maybe cut this sub-section into paragraphs to clearly show the structure of the argument.

As suggested, this section has been separated into paragraphs and slightly edited to make the individual points being made clearer.

**P15L19 Conclusion**

The conclusion is a bit long. I think the significance of the paper would appear more clearly with a better hierarchy in the importance of the findings developed in this manuscript. To me, it is not clear what are the most important results according to the authors.

The level of detail in this conclusion is too high, a lot of things are repeated (motivation for the inclusion of the brittle-failure, no improvement of the models at Pan-Arctic scale, utility of I\_eff...) that could be removed in my opinion.

Significant edits have been made to the conclusion to both shorten it, remove unnecessary repetition, and increase the prominence of the key findings in the study.

Future work should focus on the development of a full physical treatment of the impact of brittle fracture on the FSD.

Again, would it not be interesting to relate this with the work carried out on emerging brittle

rheology models (e.g. Dansereau et al., 2016)? Or to the work of Rynders that is already mentioned in the introduction? As it is, the paper does not really demonstrate the interest of using FSDs, which reduces its potential impact, at least in my opinion. Giving more context would highlight how the comparison made in this manuscript can contribute to the future of sea ice modelling.

As suggested, we have related this work to the work of Dansereau et al., (2016):

'Whilst the quasi-restoring scheme presented here is a useful tool to improve prognostic model performance and based on idealised models of brittle fracture, its current formulation relies on significant approximations. The concept of defining the 'damage' of a given area of sea ice such as used within the Maxwell-EB rheology (Dansereau et al., 2016) presents a promising basis for future developments of the brittle fracture scheme.'

Minor comments:

General:

I believe this is the first manuscript I have read that does not use a chronological order when citing 2+ references. This is not a big deal, but you might want to change that.

When citing multiple reference, we applied the same principles as used to determine the order of the reference list. Having reviewed the Cryosphere style guide, no clear guidance is given on this, and in fact the example given for multiple references is chronological rather than alphabetical.

As suggested, citations have been updated to be chronological rather than alphabetical.

The image resolution of the figures seems in general quite low to me. This is purely aesthetics, but it gives a "draft" impression of the Figures.

We believe this to be an artifact of importing the figures into a word document and therefore should not be an issue for the publication of any final paper.

Readability of graphs would also be improved with more ticks (Figure 4) or maybe a grid in the background (Figure 2,5,6,7,9).

Our personal preference is to not include grids within graphs since this can make figures appear cluttered and distort how they are interpreted.

Ticks have been added to Fig. 4 as suggested.

Aesthetics again, but the authors should consider using roman (normal) text to subscripts in equations/variable names when they have more than 1 letter. It improves the readability.

The Cryosphere journal style guide suggests that Microsoft Equation Editor should be used for equations and variables when compiling a manuscript with Microsoft Word. The font used for equations is the standard when using the Microsoft Equation Editor and cannot easily be changed. The font of equations / variables will also be updated during typesetting prior to final publication (if this paper is accepted).

I find the name "WIPoFSD" a bit hard to read (too long for an acronym, and not straightforward to pronounce). The authors might consider using a shorter name, or a name that would be related to a key property of this model (fixed-shape FSD model?).

The name WIPoFSD was defined in a previous paper (Bateson et al., 2020) so renaming it here would create an inconsistency in the literature.

**Abstract:**

P1L13à17: The beginning of the abstract would gain from being a bit more synthetic/sharper (until [...] "In this study".

The first few sentences in the abstract have been modified to improve flow and clarity:

'Sea ice is composed of discrete units called floes. Observations show that these floes can adopt a range of sizes spanning orders of magnitude, from metres to tens of kilometres. Floe size impacts the nature and magnitude of interactions between the sea ice, ocean, and atmosphere including lateral melt rate and momentum and heat exchange. However, large-scale geophysical sea ice models employ a continuum approach and traditionally either assume floes adopt a constant size or do not include an explicit treatment of floe size.'

**Introduction:**

P1L36: The number of references for the mechanical response of sea ice to stress is quite high compared to the rest, given this is not the main topic of the paper. I acknowledge these references are relevant to this topic. My main concern is that as all these references are linked to only one team working on this topic, it gives a misleading picture of the field to the reader (see for instance the studies by Shen et al., 1986 a,b; Williams et al., 2017; Boutin et al., 2021...). As mentioned earlier, the link between this manuscript and these references could fit well in the conclusion, so maybe the authors should move some of these references there.

**Original reference list:**

Feltham, 2005; Rynders, 2017; Rynders et al., 2020; Wilchinsky and Feltham, 2006

**Has been updated with:**

e.g. Shen et al., 1986; Wilchinsky and Feltham, 2006; Rynders et al., 2020

P1L42: I was a bit confused by the word "province". Whether it is correct or not, I would recommend using the word "region" that is clearer for international English speakers/reader.

'Province' changed to 'region' as suggested.

P2L2: "here" is a bit unexpected given the introduction has just started.

The 'here' is included since the MIZ does not have a singular, unique definition.

'Here' has been changed to 'in this study' to avoid confusion,

P2L9: "Note that all..." This sentence breaks the flow of the introduction a bit. The authors might want to move it a bit earlier or find a smoother way of stating this fact (just a suggestion obviously).

This line has been removed and FSD in the previous sentence replaced with 'non-cumulative floe number density' to improve flow without loss of information.

P3L4: "...FSD in the model is actively constrained according to observations, in this case by approximating the FSD as a power law." All observations do not conclude that the FSD follows a power-law. Horvat et al., (2019) does not for instance.

'According to observations' has been removed to avoid incorrect inferences.

P3L6: "though with some dependency on model structure such as how the FSD is discretised over floe size categories." This has been addressed in previous studies I believe (Horvat and Tziperman, 2015?), it could be nice to refer to them.

Horvat and Tziperman (2015) do not appear to test sensitivity to model structure. Zhang et al. (2015) test model sensitivity to spacing of floe size categories for a different prognostic FSD modelling approach and do not find a significant effect (though this result may not hold for the prognostic model being used here). Given the significant uncertainty regarding this statement, I have removed it from the text.

P4L12: I find the beginning of section 2.1.2 a bit confusing. I suggest starting with one sentence to summarize why the MLD matters in your CICE setup. The second sentence of this paragraph would make a better start for instance. The way section 2.1.3 is introduced is much clearer for instance.

The opening three sentences to section 2.1.2 have been modified to the following:

'Ocean mixed-layer properties are important in determining lateral and basal melt rates, which are both relevant for evaluating the impact of floe size on the sea ice cover (e.g. Bateson et al., 2020). Here, a modified version of the prognostic bulk mixed-layer model of Petty et al. (2014) is used rather than a constant prescribed mixed-layer depth, to better represent sea ice-mixed layer interactions and feedbacks without the complexity and computational expense of a full ocean model.'

P7L1: Could you remind the reader of what I\_var is (physically)? There are a lot of floe size names in this paragraph, it is quite easy to lose the reader.

The following line has been added to the relevant section: 'Bateson et al. (2020) suggested that I\_var can be taken as representing the history of a given area of sea ice in terms of physical processes that affect the FSD.'

P7L9: "The broader impacts of a power-law distribution on the sea ice cover can be explored whilst also including spatial and temporal variability of the FSD within the model. For mechanical processes such as wave break-up, the use of *lvar* is particularly suitable" I find these sentences a bit vague. The link with the rest of the paragraph could be more explicit.

The referred to sentences (in addition to both the prior and subsequent sentences) have been updated as follows:

'The appeal of this approach is that it is both simple and enables an exploration of the broader impacts of a power-law distribution on the sea ice cover whilst retaining spatial and temporal variability in I\_eff. For mechanical processes such as wave break-up, the use of I\_var is particularly suitable; it marks a transition from a regime where floes are being broken up to a regime where the number of floes is increasing due to the break-up of larger floes.'

P7L13: "For thermodynamic processes it makes less intuitive sense. It is not possible to define two clear regimes; instead, floes across the distribution reduce in diameter by the same magnitude in response to a lateral melting event. Here, we have modified the lateral melting scheme to calculate the change in *leff* rather than *lvar*, since it is possible to calculate exactly how *leff* would change in response to a given perturbation of the FSD." Same comment here, I am afraid that a reader that is not familiar with FSD modelling would get quite confused. A few more details about the physical reasons behind these statements could improve the readability.

This section has been modified to add clarity:

'However, it is not possible to define two clear regimes of how floe size would change in response to lateral melting; instead, floes across the distribution reduce in diameter by the same magnitude in response to a lateral melting event. Here, we have modified the lateral melting scheme to calculate the change in I\_eff rather than I\_var, since it is possible to calculate exactly how much the total floe perimeter, and therefore I\_eff, would increase or decrease in response to any change in the FSD.'

P7L28: The reference for the CPOM CICE could be given here instead of further in the text.

Schröder et al. (2019) tested a series of different parametrisations and parameter choices within CPOM CICE, not just those adopted here.

P7L34: It is likely a very naive question, but why do the authors use a winter climatology?

Note the winter climatology is applied for deep ocean properties only. This is an approximation, but it is a reasonable approximation for two reasons. Firstly, changes in deep ocean properties occur over much longer timescales compared to the surface ocean. Secondly, mixed-layer deepening tends to happen during periods of freeze-up.

P9L21: Please give the spatial and temporal resolution of these datasets.

The following line has been added: 'Both datasets have a spatial resolution of 25 km x 25 km and a temporal resolution of 1 day.'

P9L25: Another (important) reason PIOMAS is used as a reference it that it has been carefully evaluated against available sea ice thickness observations. See for instance:

Schweiger, A., R. Lindsay, J. Zhang, M. Steele, H. Stern, and R. Kwok, 2011: Uncertainty in modeled Arctic sea ice volume. J. Geophys. Res., 116, C00D06, https://doi.org/10.1029/2011JC007084.

Thank you for pointing this out. The following clause has been added to the relevant sentence: ', it has been evaluated using available observations of sea ice thickness (e.g. Schweiger et al., 2011).'

P9L36: It would help to add the panels of interest in the references to Figure 4.

Where relevant to specific panels and not the full figure, the text in the manuscript has been updated to clarify the specific panels being referred to in Fig. 4.

P9L35: "In particular, the slope of the distribution is much steeper (more negative) for the model output than observations." It would help to give a physical interpretation of this statement.

The following clarification has been added to the relevant sentence: 'i.e. the model predicts smaller floes within the range 104.8 m - 1892 m take up a much larger proportion of the total sea ice area than is suggested by observations.'

P9L40: [...] a significant improvement in the ability of the prognostic model to capture the shape of the FSD for mid-sized floes. In summer.

Following clarification added: 'over the period May – July.'

P10L25: I am a bit confused by this "However,".

The relevant comment has been removed since it was part of the discussion of Fig. 6, which has been removed from the manuscript.

P14L5àP14L13 Therefore... This is interesting, it would gain from being a bit clearer.

Section has been edited to improve clarity:

'This means that increasing the lateral melt contribution to the total melt increases the loss of thick ice in a given melt season. Vertical sea ice growth rates are inversely proportional to the sea ice thickness. Therefore, whilst the moderate reductions in thickness across large areas of sea ice from basal melt can be recovered within a single freeze-up season, the recovery of thick ice that has completely melted out from lateral melt will take several seasons of freeze-up to recover despite being over a smaller area. The reason for larger reductions in sea ice volume for prog-best compared to WIPo-best may therefore be a result of a changes to the ice thickness distribution that emerge due to the higher lateral to basal melt ratio for prog-best compared to WIPo-best.'

P15L15: "reduce the sea ice mass balance" This expression is confusing. The comparison with the results by Roach et al. (2019) in the next sentence is also a bit unclear to me.

'reduce the sea ice mass balance' has been replaced with 'result in a significant increase in total melt and a large corresponding reduction in sea ice volume.'

The following additional sentence has been added prior to the final sentence to clarify the comparison to the results in Roach et al. (2019):

'Conclusions regarding the role of the FSD in Arctic sea ice evolution also do not necessarily extend to the Antarctic.'

Caption of Figure 4: month(s)

Why is the "s" between brackets?

Some panels include data from a single month and others include data from several months.

"prog-16 performs particularly well in the Fram Strait and East Siberian Sea but less well for the Chukchi Sea. It represents a significant improvement to prog-16-nobf in all three locations."

I do not think this comment should be part of the caption.

**Comment removed as suggested.**

Caption of Figure 7:

All three simulations generally lie within the range spanned by the observational products except for pack ice extent in March after 2010.

I do not think this comment should be part of the caption.

Comment removed as suggested.

Figure 9:

It would be better to use the same vertical scale, at least for the extent (a,c) and volume (b,d).

As it is, it looks like differences between models are larger in March than in September.

Whilst we appreciate the point being made here, our intention with these plots is to present a comparison between the two simulations rather than the differences at different times of year and we have used the vertical scale for each subplot that we believe best serves this purpose. The range of the plots in September are about 4-5 times that in March. Therefore, using the same scales for

each would result in the data points in March covering about a 20 % fraction of the figure, reducing the ability to clearly identify the differences between the two simulations.

**Figure 12:**

I was a bit confused by the use of the blue and red colormaps in section A, as these colors are later used to represent a positive/negative difference in sections B and C. I would recommend using the same colormap for all quantities that are not a difference, for instance the pink/purple colormap used for panels B(e,f) and C(e,f) could be used for all panels in section A.

We agree that the use of blue and red may lead to confusion with the difference plots. We have decided to use different colours to the pink/purple scheme used in B/C(e,f) to make it clear that different sea ice metrics are being presented in these different plots.

The red and blue colour schemes in Fig. 12A have been replaced with orange and green colour schemes respectively.


**Response to Anonymous Referee #2**

Firstly, we would like to thank the reviewer for providing such thorough and thoughtful comments on our manuscript. They have been invaluable in improving the manuscript.

The manuscript by Bateson et al., "Sea ice floe size: its impact on pan-Arctic and local ice mass, and required model complexity" compares two of the main approaches for incorporating floe size distribution into a sea ice model (both using CICE) with observations in two ways. The first compares the floe size distribution with new FSD estimates from satellite imagery; the second is an evaluation of Arctic sea ice mean state over approximately the past 3 decades. With the newness of these models and the community focus on their implementation, this work is well justified. The manuscript is generally easy to read and complete. However, I have a number of concerns about how the comparison has been completed, and the presentation of the results.

**Major comments:**

Journal fit. This fundamentally is presented as a model evaluation study. Is The Cryosphere the appropriate venue for this? Would GMD perhaps be a better fit? If published in TC, the authors should more clearly address and center what new science is presented.

The Cryosphere has previously published numerous sea ice modelling-focused studies e.g. Bennetts et al., 2017; Bateson et al., 2020; Smith et al., 2022. We therefore strongly believe that this study focused on the numerical modelling of sea ice is also within the scope of The Cryosphere. In support of this we would also like to highlight that: (a) this is first study to motivate and incorporate an explicit treatment of brittle fracture into FSD models; (b) this is the first study that has compared different approaches to modelling the FSD in simulating the observed FSD shape; (c) this is the first study to present a direct comparison of the impacts on the sea ice cover of two alternative paradigms to modelling the FSD. Points (a) and (c) in particular highlight how The Cryosphere is a more appropriate fit for this research over GMD. In our revised version we highlight this novel science more clearly.

Comparison with FSD observations: I'm fundamentally a bit hesitant that floe size distribution models in global climate models are yet at a point where we expect them to match with observations (from specific location) as many other, more rigorously tested and developed features of models, can still not do so. To phrase this as a question: Why do you expect the models to represent realistic floe size distributions at a given point? Do you think there are other model factors that this representation is sensitive to, such as thickness distribution? Please discuss how other factors might impact this comparison

The points you raise are important and something that we have considered. You are entirely correct that there are several good reasons why FSD models may perform poorly in capturing the observed distribution at a given location that could result from entirely different aspects of the model (e.g. a poor representation of the ice thickness distribution, or in-ice wave fields). However, the reason we felt able to perform this comparison is because we found a remarkable consistency in the FSD plotted across the different locations and time periods considered. Figure A below shows all the observations included in our analysis included within the same figure. Whilst there is clearly variability between these different sites, the variability is still much smaller than the differences between the prognostic model without brittle fracture and the observations across all the case studies considered. We cannot expect any FSD model to precisely replicate an observed FSD, but we can expect that a simulated FSD should be within the variability in FSD shown by observations if an FSD model is accurately capturing the relevant processes. Figure 4 shows this not to be the case for the prognostic model without brittle fracture across all the case studies considered.

**Figure A:** A summary of the perimeter density distribution reported from 37 different LIDP satellite images in three different locations: Chukchi sea (plum, medium dash), Fram straight (pink, short dash), and East Siberian sea (blue, long dash).

**The following comment has been added to section 4.1:**

'When comparing observations across the sites considered in Fig. 4 there is clear variability between the different case studies, but this variability is substantially smaller than the differences between the prognostic model without brittle fracture and the observations across all the case studies considered. It cannot be expected that an FSD model can precisely replicate an observed FSD given other differences will exist between the model and the observed sea ice state such as ice thickness and concentration, but it can be expected that a simulated FSD should be within the variability in FSD shown by observations if an FSD model is accurately capturing the relevant processes.'

Comparison of sea ice mean state. It is worth noting that CICE, as most models, has had parameters largely tuned to best represent current state. As a result, the comparison of model with no additional tuning to observations (of sea ice extent and thickness) seems poorly motivated. Would we expect it to improve representation of mean state without tuning of other variables? The general setup of CICE used in this study (when using a fixed floe size of 300 m) has been adopted from Bateson et al. (2020), where the reference setup was shown to perform well against observations of sea ice extent. Similarly, Figs 5-6 in this study show the reference state performs sufficiently against observations for our requirements. If the changes resulting from the inclusion of FSD processes are not overly large, we will remain within a realistic sea ice state and would expect to see a similar impact on sea ice state from the inclusion of FSD processes relative to the reference case after retuning the model. Where significant changes to the sea ice state can be seen, we can then refer to known biases in simulating Arctic sea ice (e.g. Ivanova et al., 2016; Notz and the SIMIP Community, 2020) and consider whether the changes to the mean sea ice state would counter these biases.

**Notz, D., & SIMIP Community (2020). Arctic sea ice in CMIP6. Geophysical Research Letters, 47, e2019GL086749. https://doi.org/10.1029/2019GL086749.**

Additionally, the implementation in a forced, standalone setup is likely to see less response in sea ice state, without the possibility of atmosphere and ocean feedbacks. What I think is more interesting is

changes to sea ice mean state between models, which can suggest something about how different physics and processes relating to floe size feedback onto other sea ice characteristics. However, this is hard to see in a forced (rather than coupled) model, where feedbacks are limited. I think it is worth focusing on differences in the seasonal cycle and maps between the model which may impact these feedbacks, in the absence of coupled runs. Do these suggest improvement in how ice evolves? In short, I suggest that in the absence of additional, fully-coupled runs, the conclusions should be reframed.

What we achieve in this paper is an understanding of direct FSD impacts on the sea ice state (i.e. via increases to lateral melt rate, form drag). A significant advantage of running standalone sea ice models for such studies is we can characterise these direct impacts on sea ice state and more easily identity the mechanisms responsible for these changes. Internal variability in a climate model e.g. of atmospheric heat content, would make distinguishing the impacts of FSD processes extremely challenging. Of course, the inclusion of feedbacks with the atmosphere and ocean may increase the impact of FSD models on sea ice state (or indeed reduce it) and we do acknowledge the absence of such feedbacks as a limitation of this study within the manuscript. The challenge we would envision with refocusing the paper onto how changes in the sea ice state might influence any feedbacks with the ocean and atmosphere is we would not be able to conclude on the magnitude of any changes to feedbacks, so the conclusions reached would be very speculative.

I was left confused how what appears to be substantial, meaningful changes in sea ice thickness in Fig. 11 (5-50 cm across much of the Arctic) corresponds to almost no change in volume in Fig. 5. Perhaps it is just an interpretation error on my part, but the presentation needs to be clarified to illuminate whether there are meaningful changes in ice state, or not.

It is useful to also consider Fig. 8 here e.g. Fig. 8 shows an average reduction in volume of about 4 % for *prog-best* compared to *ref* over 2000 – 2016. Fig. 5 shows the average September volume over this period was about  $7.5 \times 10^3 \text{ km}^2$ , so 4 % of this would be about  $0.35 \times 10^3 \text{ km}^2$ . Given the scale for the relevant panel spans about  $3 - 15 \times 10^3 \text{ km}^2$ , the spacing between the three simulations can be expected to be very small. Figure 5 is not a useful figure for interpreting the differences between the simulations; it exists to compare all three simulations to observations. A 4 % reduction in volume would correspond to a 4 cm average decrease in sea ice thickness for a sea ice cover of a fixed 1 m thickness, and whilst these numbers represent very rough averages, they do indicate that the scale of change shown in Fig. 11 is consistent with the results in Fig. 8.

Minor/specific comments:

P1, L1: It would be helpful to provide a brief introduction to the range of floe ice sizes and key processes (and why it is useful to capture it with a distribution, as is often done for thickness) As noted below, these points are addressed later on in the introduction.

P1, L16: I would suggest that the sentence beginning with "Observations show..." should go before the sentence beginning with "Large-scale..." on L15. Corrected as suggested.

P1, L36: Is cluster of sea ice into larger floes really a process impacted by floe size, or is it primarily a process in determining floe size?

A quote from the discussion / conclusions of Herman (2012): 'As shown here, clusters of ice floes have a power-law size distribution with an exponent  $\alpha$  dependent on (and generally different from) the exponent  $\alpha_r$  of the FSD. Assuming the clustering-freezing scenario as one of the floe-generation mechanisms, this should lead to changes in the FSD exponent.' i.e. clustering of sea ice is both impacted by and impacts floe size. P1, L37-38: May be worth considering additionally/alternatively citing Keen et al., 2021, which summarizes CMIP6 sea ice models and show that most use some derivation of CICE or LIM, which all have the same lateral melting parameterization Additional reference added as suggested.

P1, L39: Floe size is also not considered in dynamics. Following is added to sentence: 'or dynamics (Tsamados et al., 2014)'.

P2, L5: Why? Need to briefly describe the floe size distribution to justify why a power law is used (i.e., that typically more small floes). Replace "...generally fitted to a..." with "...summarized by fitting to a..."

P2, L17: Please note to what degree power law does/doesn't fit observations summarized. Is it a manner of convenience, or do observations support its use?

The opening of the relevant paragraph has been modified to address the above points. It is worth noting that the points raised above are not trivial questions, and indeed whole papers have been written that explore these questions (e.g. Stern et al., 2018b; Horvat et al., 2019). The opening of the relevant paragraph now reads:

'Observations of the floe size distribution (FSD) show a large ratio of smaller floes to larger floes; this distribution of floe sizes is often summarized using a truncated power law (Rothrock and Thorndike, 1984; Toyota et al., 2006; Perovich and Jones, 2014; Stern et al., 2018b). Studies generally show that a power law produces a reasonable fit to the observations presented, though the validity of using a power law to fit floe size data remains an open question (Stern et al., 2018b), with several studies disputing the extent to which a power law is a good description of the FSD (Herman, 2010; Horvat et al., 2019; Herman et al., 2021).'

P2, L23-27: as noted above, it would be helpful to mention these processes earlier in introduction The primary purpose of the first paragraph of the introduction is to highlight the important of floe size to the wider Arctic system, and we are concerned that describing processes that influence floe size within the paragraph would detract from this important point.

P2, L28: It would be helpful to include a transition sentence motivating introduction of brittle facture – that it is missing in most models, and may be important. Perhaps something like what is currently L42-43.

In response to comments from the first reviewer, we have moved most of the discussion of brittle fracture previously presented in the introduction to section 2.2.2. The updated reference to brittle fracture in the introduction addresses the above point.

Brittle fracture is now briefly introduced in the introduction in the following way:

'The limited spatial and temporal coverage of floe size observations has prohibited effective evaluation of these models, though there have been recent efforts to develop satellite derived FSD produces to enable such evaluations (Horvat et al., 2019). It is nevertheless anticipated that important processes are not yet represented in these models. For example, thermodynamically-driven break-up of floes along existing cracks and refrozen leads in the sea ice cover (Perovich et al., 2001).'

P3, L3 and L4: replace "represents" with "is in" 'Represents' has been replaced with 'is within'.

P3, L32: Would be helpful to also introduce the ITD, which is referenced in relation to the prognostic model

Following description of ITD has been added to section 2.1.1:

'The standard sea ice thickness distribution in CICE distributes ice area between five thickness categories, with the spacing increasing for thicker categories. The ice area in a given category evolves in response to dynamic and thermodynamic processes according to a linear remapping scheme (Lipscomb, 2001).'

P4, L11: Is there any possibility for ice-ocean feedbacks, such as albedo feedback, in this setup? Please specifically address in the text

In this study we use a modified version of the prognostic bulk mixed-layer model of Petty et al. (2014) rather than a constant prescribed mixed-layer depth. In this model, mixed-layer temperature, salinity, and depth all evolve in response to sea ice-ocean interactions. A version of this model (with some additional physics related to snow on sea ice) has been used in a previous study to evaluate sea ice-ocean feedbacks in Antarctic shelf seas, including the albedo feedback (Frew et al., 2019). The introduction to the mixed-layer model in section 2.1.2 has been modified accordingly: 'Here, a modified version of the prognostic bulk mixed-layer model of Petty et al. (2014) is used rather than a constant prescribed mixed-layer depth, to better represent sea ice-mixed layer interactions and feedbacks (e.g. the ice-ocean albedo feedback) without the complexity and computational expense of a full ocean model (e.g. Frew et al., 2019).'

Frew, R. C., Feltham, D. L., Holland, P. R., and Petty, A. A.: Sea ice – Ocean Feedbacks in the Antarctic Shelf Seas, J. Phys. Oceanogr., 49, 2423–2446, https://doi.org/10.1175/JPO-D-18-0229.1, 2019.

P4, L24: It would be helpful to be more clear in the description of CICE and general model that this is being used in a standalone setup.

Section 2.1.1 now opens as follows:

'In this study, we model the Arctic sea ice cover using a local version of the CICE sea ice model in a standalone setup.'

P4, L39: What is the L used in standard implementation? Is it 300 m, as used in lateral melt? Please define.

In the Tsamados et al. (2014) implementation, L is defined as a function of local sea ice concentration. We do not want to explicitly define this parameterisation here since this parameterisation is not used in this study. Instead, the following clarification will be added. We have added the following clarification to the text: 'which in Tsamados et al. (2014) is calculated as a function of sea ice concentration as per the parameterisation outlined in Lüpkes et al. (2012).'

P5, L19: In the introduction you present the WIPoFSD before the prognostic model. It would be helpful to be consistent about the order throughout manuscript. I suggest presenting WIPoFSD, then prognostic model, the brittle fracture scheme.

We have reviewed the manuscript to ensure the two FSD models are presented in a consistent order.

P6, L31: change to "spatial and temporal scales" or "spatial scale and timescale" or similar Changed as suggested.

P7, L1, 11, etc.: I'm not really sure I understand the physical implications of I\_var. What is it intended to represent? What are the implications for observational comparisons? Modifications have been made to section 2.3 to better address these points. Bateson et al. (2020) suggested that I\_var can be taken as representing the history of a given area of sea ice in terms of physical processes that affect the FSD. It is effectively a model tool to enable spatial and temporal variability of the FSD driven by relevant processes within the constraints of a fixed power law shape. As such there is no exact corresponding observable parameter to the model concept I\_var, but it

roughly corresponds to the largest possible floe size that is likely to exist within an area of sea ice after that sea ice area has experienced a series of processes that change floe size.

The following clarification for I\_var has been added:

'Bateson et al. (2020) suggested that I\_var can be taken as representing the history of a given area of sea ice in terms of physical processes that affect the FSD.'

Further modifications have been made to section 2.3 to provide a clearer explanation of the physical implications of l\_var.

P7, L36: Is there anything that can be referenced to demonstrate that ERA-Interim wave product is reasonable to use for the Arctic?

The following details have been added to section 3.1:

'The ERA-Interim reanalysis has been selected for the ocean surface wave field forcing as this dataset has generally been found to perform well in comparison to other reanalyses against observations of wind speed and wind speed profile in the Arctic during summer months (e.g. Jacobson et al., 2012; deBoer et al., 2014).'

What is the treatment for waves in sea ice?

The following clarification has been added to section 2.2.1 to describe the wave-in-ice treatment for the prognostic model:

'Unlike Roach et al. (2019), we do not use a separate wave model coupled to CICE to calculate the necessary wave properties within the sea ice-covered grid cells for use with the wave-dependent floe formation parameterisation. Instead, we adapt the scheme used in Roach et al. (2018), which calculates in-ice wave properties using an extrapolation approach from forcing external to the sea ice cover to calculate the necessary in-ice wave properties.'

For the WIPoFSD model, a summary is provided in Appendix A and full details are available in Bateson et al. (2020). The following line in section 2.3 has been modified to clarify this:

'A full description of how these processes are represented within the WIPoFSD model, including a description of the advection scheme for waves in sea ice, is available in Bateson et al. (2020); a summary has also been provided here in Appendix A.'

P8, L9: Why 'best'? Unless you plan to show others runs involved in selection process, I suggest use of just 'WIPoFSD' and 'prog' for simplicity and clarity

There are a couple of reasons we use 'best'. Firstly, to ensure these simulations are clearly distinguished from the simulations using 16 floe size categories. Secondly, this naming has been retained from Bateson (2021) to ensure consistency.

P8, L27: Perhaps simply "FSD observations" as title? Corrected as suggested.

P8, L29-30: I believe since this is included in contributions and acknowledgements, it is not necessary to include funding or names of contributors here. Corrected as suggested.

P8, L31: replace 'samples' with images Corrected as suggested.

L31: remove "three months" Corrected as suggested.

L39-40: What is the impact of this (as well as lower cutoff, L43+)? It seems like you could instead include the largest floes in the largest category, which may sometimes show a FSD with the uptick demonstrated by the prognostic model.

In the text the following explanation is provided: 'This step is necessary because the presence of a single large floe, comparable to the image size, can cause a large perturbation across the distribution reported for that location. Instead, only floe size categories that are small enough to consistently be populated by multiple floes across all sampled images are retained.' i.e. we do not include larger floes in the comparison due to the finite size of the images. Another way to consider this is the following. The prognostic model does not simulate the evolution of individual floes but instead floe area; what emerges from each grid cell is a statistical description of the FSD. In comparison, observations effectively produce a sample from that statistical population. If we compare these directly, we would not be comparing the same thing and the comparison would not be valid or useful. The FSD sample and statistical description will however converge for floe size categories in the sample that are sufficiently populated. Due to the power law shape of the FSD this is true for smaller floe size categories but not larger floe size categories, hence by applying an upper cut-off a useful and valid comparison can then be made. In terms of the lower cut-off, this is necessary since floes smaller than 100 m are not well-resolved in the observations and therefore there is no valid comparison that can be made between model output and observations for floes smaller than 100 m. The text in this section has been updated to better address this point:

'The first step of processing the raw floe size data, consisting of a list of individual floe sizes, is to sort them into the Gaussian-distributed floe size categories used within the prognostic model for ease of comparison. Any floes that exceed the upper diameter cut-off of the largest category, 1892 m, will be discarded from the analysis. This step is necessary because the two models simulate the full FSD, and not individual floes. Floes large compared to the image size are inadequately sampled in observations to construct the full FSD. For example, the presence of a single large floe, comparable to the image size, can cause a large perturbation across the distribution reported for that location. Instead, only floe size categories that are small enough to consistently be populated by multiple floes across all sampled images are retained. A lower floe diameter cut-off of 104.8 m is also applied to this analysis, taken to be the smallest floe size that can be reliably resolved from the observations for the methodology and resolution used. The limiting factor on the smallest resolved floe size is the ability to resolve gaps between floes.'

P9, L11: change "is not novel" to "has been used previously" Corrected as suggested.

P9, L27: Is there an appropriate reference for this statement?

Appropriate references have been added for this statement (and this section more generally): 'Whilst the PIOMAS volume product is a reanalysis and does not incorporate direct observations of the sea ice thickness, it has been evaluated using available observations of sea ice thickness (e.g. Schweiger et al., 2011). This product is often used to test model performance in simulating the total Arctic sea ice volume (Schröder et al., 2019) due to the challenges in estimating sea ice thickness from radar altimetry and limited availability of in-situ thickness measurements (e.g. Massonnet et al., 2012; Stroeve et al., 2012; Ridley et al., 2018).'

P10, L8: Perhaps simply "Comparison of sea ice extent and volume" might be a clearer heading Given the later section 4.3.1 compares model output in terms of extent and volume, it is important to be clear that this section is explicitly a comparison of model output to the observed extent and volume.

P10, L1-3: It would be helpful to show some results of when and where the brittle fracture scheme is implemented. Where is it most necessary? What does this suggest about what it corresponds to physically?

The results in Fig. 4 provide case studies of the impacts of this scheme over different locations. We anticipate that the brittle fracture scheme is most necessary in regions that are part of the pack ice

in winter (i.e. subject to in-plane brittle fracture) but then transition to being within the MIZ during the melt season and therefore likely to be subject to significant thermodynamic-weakening of the sea ice cover.

Existing discussions in section 5.1 have been edited an extended to better address the above points. The 2nd paragraph in this section provides a more complete discussion of the different impacts of brittle fracture scheme for the case studies presented in Fig. 4. Paragraph 3 has been updated to give a clearer physical interpretation of the results presented in Fig. 4.

P10, L 18-20: I think it is necessary to clarify here that these runs are done in a standalone setting, and that the possibility for feedbacks in a fully coupled climate model may give different results. The following clarification has been added to the end of the 2nd paragraph in section 4.2: 'though this conclusion does not necessarily extend to climate simulations where FSD impacts on sea ice feedbacks with the ocean or atmosphere could produce larger changes to the sea ice state.'

P10, L23-24: It's not clear what negative trends in the percent difference suggest. Does this suggest some sort of feedback in model?

The line being referred to is just highlighting that the Arctic sea ice extent and volume has decreased over the period 1990 – 2014, as expected. This point is perhaps unnecessary to make and the section has been modified appropriately.

The line, 'Clear negative trends in the March volume and September volume and extent can be seen.' has been removed.

P13, L14-16: Maps show more substantial changes in representation of sea ice state. Do these suggest improvements?

The following section later in the same paragraph was intended to address this point: 'Nevertheless, significant biases have been identified in coupled climate models in simulating the sea ice concentration (Ivanova et al., 2016) and CICE, in particular, has been shown to overpredict the sea ice concentration at the sea ice edge and underpredict the concentration within the pack ice (Schröder et al., 2019). In Bateson et al. (2020), the WIPoFSD model was found to provide a limited correction to this model bias. Similarly, Fig. 10 shows that the prognostic model produces a stronger correction to this model bias, driving reductions in sea ice area fraction in the MIZ and small increases in area fraction in the pack ice.'

Limitations in the accuracy of sea ice concentration and thickness data obtained from satellites preclude a more detailed comparison.

The following sentences have been added to the relevant paragraph to better address this point: 'The accuracy of sea ice concentration measured using passive microwave data can be as low as  $\pm$ 20% in summer or the MIZ (Meier and Notz, 2010). Measurements in sea ice thickness from radar altimetry can also have high uncertainty, with snow depth and density being the primary source of error (Tilling et al., 2018).'

P14, L34: It may be worth mentioning that this is particularly relevant in a standalone sea ice model, as run here. In a coupled context (for which climate models are often used) the sea ice model is typically a small component of the total cost, and so the additional cost from the FSD is relatively not substantial.

We are inclined to disagree with this comment. Whilst the inclusion of an FSD model would clearly represent a proportionally smaller increase in run time for a climate model compared to a sea ice model, the pressure on model efficiency is also much higher for climate models than standalone sea ice models due to the existing bulk of the model.

P15, L6: What is meant by 'in-ice wave scheme'? Is it more accurate to say that the waves are forced with reanalysis?

The forcing defines the wave properties external to the sea ice cover, but both models have an internal in-ice wave scheme to determine the wave properties within the sea ice cover. The text 'in-ice wave scheme' has been replaced with 'external forcing combined with in-ice wave scheme'.

P15, L19: Future work to address the impact on Antarctic sea ice representation and comparison with observations may also be useful

Following comment has been added to the final paragraph of the conclusions: 'In addition, it would also be beneficial to evaluate whether the conclusions reached in this study extend to the Antarctic.'

Figure 1: I find this figure really hard to interpret currently. A few reasons/suggestions... It might be better to use more realistic 'floe diameter' bounds, or to remove numbers from y axis, as it currently is hard to interpret these as actual bins. Only 2 examples are needed showing where redistribution is applied and where it isn't (for example, far left and far right). For one where redistribution is applied, show the new floe size distribution resulting more clearly. It would also be helpful to add lines showing actual density gradient for comparison to dashed purple line.

Figure 1 has been modified based on the above comments. Numbers have been removed both axes since these can produce an incorrect interpretation of the figure. One of the three examples has been removed. The redistribution is now shown in a separate panel showing the FSD before and after a brittle fracture event. We have decided not to use lines to show the actual density gradient to avoid adding too many details to the diagram.

Figure 2: I'm not convinced that this is necessarily a "non-physical feature" of the model, as it is simply capturing floes that are potentially beyond the bounds here, and is not reported as such in Roach et al., 2018. A comparison to observations without largest floes removed may be helpful to show if this is ever observed in observations. Additionally, please add a label to this figure demonstrating that it is only for areas of SIC 15-80%

By 'non-physical feature', what we mean here is that a sudden, discontinuous change in gradient from negative to positive in the perimeter density does not represent a physical behaviour of the FSD, and instead is a feature that emerges due to how the model is designed (either due to missing fragmentation processes or from imposing a fixed maximum floe size). In terms of comparing this to observations, as mentioned above, we exclude larger floes from the comparison due to finite-image size effects.

We have removed the 'non-physical feature' and replaced this with a more complete explanation as to why the uptick is described as 'artificial':

'Also highlighted in the figure by a blue transparent box is an artificial 'uptick', a non-physical feature of the model also reported by Roach et al. (2018a).'

Is replaced with:

'Also highlighted in the figure by a blue transparent box is an artificial 'uptick', a feature of the model also reported by Roach et al. (2018a) that results from prognostic model design and structure (e.g. missing fragmentation processes, upper limit on floe size) and does not represent a physical behaviour seen for the FSD.'

It is has been clarified in the caption that model output is averaged over areas with between 15 – 80% sea ice concentration.

Figure 3: Are these exact bounds of model areas? If not, a different symbol may better communicate that, as the boxes suggest that this is the exact selection of grid cells. Yes, these are the exact bounds of the model areas.

Figure 4: Again, not necessary to name co-authors in the text. Corrected as suggested.

Figure 5: The change of prognostic models being compared is a bit confusing. Is it worth including other prognostic models somewhere as well, to show if/that there is little difference in sea ice state? Also, as the 'prog-best' doesn't include brittle fracture (right?) it would be helpful to note that in the short name for clarity.

*prog-best* does include brittle fracture (it is identical to prog-16 apart from the number of floe size categories, as discussed in section 3.1). The purpose of using 12 floe size categories for the model comparison rather than 16 is due to the significant computational cost associated with simulating the additional categories (and this cost increases non-linearly with the number of floe size categories). The impacts of the FSD on sea ice cover via lateral melting and form drag are determined by the proportion of sea ice taken up by smaller floes (e.g. Steele, 1992; Tsamados et al., 2015), and therefore improving the resolution of the FSD shape for larger floes will not have a significant impact on sea ice state.

Table 1 has been updated to clarify that both *prog-16* and *prog-best* include brittle fracture. The following explanation has been added to section 3.1 to explain why 12 floe size categories are used in *prog-best* rather than 16:

'Whilst 16 floe size categories could also be retained for the prog-best simulation, the increase in model run time increases non-linearly with increasing number of categories. In addition, the improved resolution of the shape of the distribution for floes of a size of 1 km or larger is not significant when considering the impact of an FSD on sea ice via the floe edge contribution to form drag and lateral melt rate, which both scale to the inverse of floe size. Therefore, 12 floe size categories represents a more practical choice for the prognostic FSD model.'

Figure 5-7: This feels like a lot of plots to show for almost no change between any of the models. Can this be simplified to one or two key panels, and then state there is no observable change in others? We appreciate the point being made here, but a key aim of this study is to establish the contexts where the impacts of the two different FSD models are either significant or can be distinguished. In order to do so, it is also important to discuss and give due prominence to contexts where the impacts are not significant i.e. a 'negative result'. However, given the present length and large number of figures within the paper, we recognise the need to remove at least one figure. As such, we have decided to removed Fig. 6 from the manuscript, since we believe that this figure offers the least additional insight of Figs 5-7.

Figure 6 and any references to this figure have been removed from the manuscript. Figure numbering has been updated accordingly.

Figure 8: I might suggest to swap these plots around to show both models and same subplot, with top for sea ice extent, bottom for volume. This would then allow to show some comparison in difference of observations from reference. (e.g., Are model changes moving it in the right direction?) Figure corrected as suggested.

Figure 9: I am unclear how this figure is different from what is shown in Fig. 6, in terms of the takeaways. How do we know if this is improving the comparison if the scale of change is not comparable to the difference from observations?

As mentioned above, Fig. 6 has now been removed from the manuscript. Note that the purpose of Fig. 9 is to make a direct comparison between the two FSD models in terms of impacts on the sea ice cover.

Figure 11: I think this is the most useful and interesting plot! But, I'm quite confused how what appears to be substantial changes in sea ice thickness in A(f) agree with what is in Fig. 5 – where almost no change is observed.

This issue has been addressed in an earlier comment.

Some thoughts: Could the difference in fractional ice area/thickness compared to observations also be shown?

As mentioned above, the high uncertainty in observed sea ice concentration and thickness at grid cell scale would make reaching useful conclusions from such a plot challenging.

It would be helpful to place the effective floe size upfront (at the top) to set it apart from differences, and also make this more clear in the figure caption (meaning, that floe sizes are NOT a difference).

Suggested modifications have been made to figure and captions.

Figure 12: I'm not sure what to take from these plots, based on the units show. Would it be more helpful to show standard deviation as a percent of mean value?

The difficulty in presenting the standard deviation as a percentage of the mean is the dominant resulting signal will then just be where the ice is thin; it does not enable us to identify regions of high sea ice variability away from the sea ice edge. By presenting absolute changes in standard deviation alongside the standard deviation in the reference case, it is possible to identify both where notable changes in variability do occur away from the sea ice and whether this change represents a fundamental change in behaviour of sea ice in that region of the Arctic.

The following additional explanation has been added to explain why Fig. 12 is important to consider: 'Furthermore, a small change in mean sea ice state may disguise a much larger change in sea ice variability.'

Also, it would be nice to be consistent with the months shown in Fig. 11.

Whilst we agree it would be nice to include June in addition to March and September in Fig. 12, this would result in a total of 24 panels (rather than the current 16, and also much larger than the 18 panels included in Fig. 11). We believe that the size of panels in such a figure would be too small for effective interpretation, and as such we reluctantly made the decision to exclude results for June.

Table 1: "CPOM-CICE" is not needed in model description, as all are the same. It might be helpful to separate technical details into finer resolution categories, such as brittle fracture (yes/no), # floe size categories; d\_min/d\_max (where applicable)

We do not think it makes sense to add additional categories here given d\_min / d\_max and fixed I\_eff are only applicable for one out of the five simulations described. Whilst 3/5 use the prognostic model, the differences are straightforward to describe (i.e. 12 or 16 floe size categories and whether or not the setup includes brittle fracture) and do not necessitate additional columns in the table. CPOM-CICE has been removed as suggested. It has been clarified for all prognostic simulations whether or not they include the brittle fracture scheme.

In this regard, I want to congratulate the authors on having presented their methodology and results in a very concise way. I could very well follow the general idea and nuances in the methods and the experimental setup. However, there are a couple of points that I have identified which might be picked up during the review.

Terminology

— I am somewhat confused by how you address the two model types. While you are rather consistent in the usage of the term 'prognostic' model, you are less stringent in your reference to the WIPoFSD model. You also refer to it as the power-law variant, etc. It took me a while to get my head around this. Yet the brittle-fracture restoring in the prognostic model is also a power law. In my view, the key difference is the either the 'prognostic' or the 'diagnostic' character to determine the FSD. Couldn't the latter label 'diagnostic' be used instead of WIPoFSD or power-law variant.

Thank you for highlighting the inconsistent labelling of the WIPoFSD model. We have reviewed the manuscript to ensure this model is referred to in a consistent manner. We have also included a new table to provide definitions for key terms used in this manuscript (e.g. WIPoFSD model, power-law fit). Regarding the term 'diagnostic', we are not sure that this would be an accurate way to describe the WIPoFSD model, since diagnostic implies a parametrisation where the FSD state is determined from co-temporal sea ice / atmosphere / ocean conditions, which is not the case for the WIPoFSD model. We think it is best to retain the WIPoFSD name for consistency with previous publications.

**All references to the WIPoFSD model have been reviewed and modified where appropriate to ensure consistent naming. We have also introduced a new table (table 1), to provide clarity on different terms used within this manuscript.**

You use a lot of quantities for the FSD like number density, perimeter density, perimeter density distribution, thickness probability distribution, fragment size distribution, etc. As they are introduced for the respective modules where they are applied, I got quite confused. To facilitate the accessibility, you could optionally present all of them and their relations in a dedicated paragraph.
We would like to the thank the editor for this suggestion. Since there are several terms related to the FSD that appear throughout the manuscript, we have decided to summarise these with a table that can easily be referred to.

**A new table (table 1) has been introduced to provide clarity on key terms used within this manuscript.**

**Floe Size Variability**

In figure 12, you present the floe size variability. As I understand it it is an inter-annual variability. If so, I have two question:

— In P12L18-19: You mention that the measurement of floe size variability might help you to differentiate between a 'prognostic' and a 'diagnostic' FSD model. Your observations in Fig.4 each cover 2 distinct years which might allow you to infer some effective floe size differences. You might need to come up with a strategy how to compute this metric from the observations (e.g., Horvat et al., 2019). Then you could give a first answer on which model is better suited. Otherwise explain why this might not be possible.

We believe this should be fairly straightforward to do given Horvat et al., 2019 calculated their observable floe size metric from linear floe size statistics, which corresponds well with the concept of effective floe size.

We have added the following clarification to the conclusion:

'especially since the methodology of Horvat et al. (2019) involved collecting linear statistics of floe size, and effective floe size is a linearly averaged representation of the FSD.'

— You further mention in the introduction that the floe size distribution shows a seasonal dependence which is partially visible from Fig. 4. Would the 'diagnostic' power-law approach benefit from a seasonally changing exponent linked to climatic conditions. Could this help to increase the variability in the effective floe sizes?

This is an interesting idea, and in fact a seasonally changing exponent has previously been explored within the WIPoFSD model by Bateson et al. (2020). In this case, we wanted to determine WIPoFSD model parameters according to the observations presented in Fig. 4. Whilst these observations may hint at a seasonal evolution in the exponent, the variability between co-temporal datasets is sufficiently high compared to any trend in the exponent that we cannot be confident in this result. It is definitely a theme to explore in future however, as more observations are made available.

**Brittle fracturing**

In the prognostic model, you employ the brick fracture scheme (BF) which redistributes floes sizes towards smaller categories. Yet in Fig. 4, you show that after activation of this BF scheme, the perimeter density stays roughly the same for low-end floe sizes (around 100 m) but it significantly increases for the larger floe sizes. Why is that? Explain.

This point is addressed by the 2nd paragraph of section 4.1 within the manuscript (note this section has been edited in response to reviewer comments to provide further clarity to the points made here):

'It is worth commenting briefly on how the brittle fracture scheme can improve model performance compared to observations, given it is a counterintuitive result that increasing floe break-up would produce a shallower slope in perimeter density. As discussed in section 3.2, the largest floe size categories in the prognostic model are excluded from the comparison to observations to exclude the non-physical 'uptick' that forms (Fig. 2). Whilst a reduction in ice area fraction in the largest category and an increase in the smallest category can be expected, the change in ice area fraction in the remaining categories depends precisely on the balance between ice area fraction lost from that category (sink) and ice area fraction gained from the adjacent larger category (source). In this case, the presence of the 'uptick' shown in Fig. 2 for the prognostic model without brittle fracture results in the source of floe area being larger than the sink for most floe size categories and a net reduction in gradient overall from including brittle fracture.'

Effectively, larger increases in perimeter density are seen for the larger floe size categories in Fig. 4 due to their adjacency to the categories spanned by the 'uptick' in the model.

**Uptick**

You mention this spurious increase in perimeter density for the prognostic model. You try to circumvent this problem for the validation agains observation by increasing the number of size categories. Fair enough. I further understand that the brittle fracture scheme might alleviate this problem to some degree. Moreover, you mention that the an increase in the radius' r\_max' (Eq. 9) could help, which is set to 1'700m. When comparing to the 'diagnostic' power-law model, a maximum diameter ('d\_max') of 30'000 m is specified. The difference is a factor 10. Why not simply adjust 'r\_max'. Explain.

In the prognostic model, the largest floe size category effectively acts as a repository for large floes. In order to resolve the shape of the distribution for these larger floes, more floe size categories are needed. This could be achieved in two ways, both of which have a significant associated cost. The first is to reduce the resolution of smaller floe size categories to improve resolution of larger floes, however higher resolution of smaller floes is more important here since these are most important in terms of FSD impacts on the sea ice cover. The alternative strategy is to increase the number of floe size categories, but the computational cost of the prognostic model scales non-linearly with the number of categories and this cost very quickly become prohibitive.

**3. SIGNIFICANCE (Impact): 3**

With my external perspective, I wonder about the key conclusions.

Can we ignore floe sizes if the interest was not in the seasonality but rather in general min/max seaice extents?

Based on the results presented, this is a reasonable conclusion, though there are limitations to this conclusion (see section 5.4 of the manuscript). As mentioned above, this in itself is an important result for climate modellers who must balance reducing structural uncertainty in climate models with maintaining computational efficiency.

We have updated the manuscript to ensure the point made above is clearly addressed in the discussion and conclusions. In particular, at the start of section 5.2, we make the following comment (regarding the impacts of FSD models on a pan-Arctic scale):

'This is an important result for climate modellers since it assuages concerns that the FSD represents a source of structural uncertainty in climate models.'

I wonder if the gain in the seasonal evolution is really significant by either FSD model.

Whether or not the impacts of the two FSD models on seasonal sea ice retreat are significant strongly depends on the research question. For example, if you are a climate modeller, the size of the impact of either FSD model on seasonal retreat of the sea ice is likely insufficient to justify the increased computational cost of the FSD model. Although, as noted above, this in itself is a useful result. Alternatively, if you are interested in regional sea ice modelling, especially the seasonal evolution of sea ice in the Greenland and Barents Sea, the results presented here do in fact indicate that FSD models have a significant role to play in the evolution of sea ice in these regions. The impact of both FSD models is particularly significant on the melt evolution (specifically the ratio of lateral to basal melt) and this can impact open water formation during the melt season (Smith et al., 2022). It is also important to note that we use a standalone sea ice model in this study and, as we note in section 5.4, larger FSD impacts may be found via feedbacks between the sea ice and ocean / atmosphere.

We have updated the manuscript to more clearly highlight the points discussed above in the discussion and conclusions. In particular, we make the following point at the end of the 2nd paragraph in the conclusion:

'These results are important for climate modellers as they suggest that the FSD is not a significant source of structural uncertainty in climate models. FSD processes will, however, be of importance for several key applications and research questions such as regional sea ice modelling and the formation of open water during the melt season (e.g. Smith et al., 2022).'

What are the main reason why the regional sea-ice area/volume are note captured by the continuum mode, if not the floe sizes?

This is a question that preoccupies many authors in this field and one which is very much beyond the scope of this paper (see for example Notz and the SIMIP Community, 2020). It is worth noting, however, that it is not necessarily internal model physics that limits sea ice model performance against observations but the representation of the atmosphere or ocean (in this case the model forcing).

Notz, D., & SIMIP Community (2020). Arctic sea ice in CMIP6. Geophysical Research Letters, 47, e2019GL086749. https://doi.org/10.1029/2019GL086749

How transferable are your methods. Are they very specific to the CICE implementation?

The methods are not specific to the CICE implementation and will be straightforward to incorporate into other continuum sea ice models (in fact the intention will be to include at least one of the two FSD models into the sea ice model SI3 in the near future).

The first sentence in the 2nd paragraph in the conclusions now reads as follows:

'Simulations were completed using the two FSD models within a standalone setup of the sea ice model CICE, though it should be noted that both FSD models can easily be implemented into any continuum sea ice model.'

Admittedly, I am not an expert on this topic but these questions moderate my evaluation of the significance of this study.

**4. PRESENTATION QUALITY: 2**

The paper is well written and structured. Findings are well supported by figures of mostly good quality. Yet I want to already suggest that you re-structure sections 2 and 3. In my view, section 2 is the methodology while section 3 is the experimental design and the cal/val data. The data section might be presented separately. Eventually, Before the experimental design. The figure amount is rather high and I sense that some could be transferred to the appendix.

Regarding the paper structure, we have used a similar approach to other FSD modelling studies (Bateson et al., 2021; Roach et al., 2018). In both referenced studies, section 2 is used specifically to describe the model. The simulations actually performed are then described in section 3 i.e. separately to the model description. As such, our preference is to retain the same paper structure here. We also would prefer to retain the description of model simulations and how the observations are processed to compare against model output in the same section since choices made for the former influence the latter and vice versa; presenting them together highlights this relationship.

In terms of the number of figures, both reviewers queried whether Figs 5-7 were all needed within the manuscript or whether one or two could be removed. As such, we have decided to remove Fig. 6 from the manuscript, since we believe that this figure offers the least additional insight of the three. We have decided to retain Figs 5 and 7 since we believe that they are important components in presenting the full picture of impacts of FSDs on the sea ice cover.

Figure 6 and any references to this figure have been removed from the manuscript. Figure numbering has been updated accordingly.

**Sea ice floe size: its impact on pan-Arctic and local ice mass, and required model complexity**

Adam W. Bateson1, Daniel L. Feltham1, David Schröder1,2, Yanan Wang3, Byongjun Hwang3, Jeff K. Ridley4, Yevgeny Aksenov5

[revised manuscript text omitted]

---

## Referee Report (RR1)

This is the second review of the manuscript entitled *Sea ice floe size: its impact on pan-Arctic and local ice mass, and required model complexity.*

I am overall satisfied with the answers to my previous review and the modifications to the manuscript. The work made by the authors has significantly improved the quality of the study in my opinion (congratulations to them), and I would be happy to recommend the paper for publication after some minor adjustments.

A general comment is that the paper is much clearer than it was, but also quite longer and with some repetitions between the results, the discussion and the conclusion. This is not a major problem, and the manuscript could be left as it is, but the authors should not hesitate to remove some comments that are repeated throughout the paper before publication (for instance the reasons behind the uptick).

P1L22: "We demonstrate that a parameterization of in-plane brittle fracture processes enables the prognostic model to achieve a reasonable match against the novel observations".
I slightly disagree with this sentence. The study demonstrates that adding a term to the prognostic is needed to achieve a reasonable match against the novel observations, and that there are good reasons to believe that this term represents the effects of in-plane brittle fracture. Also, the manuscript spends some time discussing the effect of this parameterization and why it improves the results, so it might be worth summarizing these results in the abstract.

I would therefore suggest writing something like:
"We show that adding a term that relaxes the FSD towards a power-law enables the prognostic model to achieve a reasonable match against the novel observations in the summer. We suggest this term represents the effects of in-plane brittle fracture that break the larger floes (>2km) into mid-sized floes (100m-2km)."

P6L24: "Clearly, brittle fracture events…size."
I agree that this is very likely the case for the larger floes (>100m), but more uncertain for smaller floe size. Maybe add a comment on that (if you agree)?

P7L18: "stronger regions of sea ice"
I am not sure of what you mean by this expression. Regions where the ice is thicker/stronger?

P8L17: "it marks…larger floes": I find the sentence a bit confusing.

P15L35: "This is an important…models." Yes, but I suspect the quantities you are looking at remain quite constrained by the ocean and atmosphere forcing. I am therefore not certain the conclusion you draw from a stand-alone simulation can be extended to fully coupled climate models. I am aware this is discussed a bit further in the text, but I think it should be mentioned here too.

P18L30: Same comment. Maybe replace "is" with "may".

---

## Author Response (AR2)

**Response to referee comments**

**(Referee comments are shown in black, our response is in blue and changes to the manuscript are shown in red. The revised manuscript is also included in this document.)**

**Page references are given to the updated manuscript as PXLY indicating that the manuscript has been updated on page X line Y.**

Thank you again to both the reviewers and the editor for their helpful feedback. We appreciate the recognition of the significant improvement in quality of the manuscript since the original submission and are grateful to both reviewers and the editor for their role in this process.

**Reviewer 1**

The revisions that the authors have made on the manuscript have substantially improved its quality. I appreciate the responses to my major comments and the additional text that has been added. I have a number of remaining comments on this version of the manuscript. After these comments have been addressed, I believe the manuscript should be published and will make a meaningful contribution to the literature on modeling of floe size distribution in the Arctic.

P1L29-31: suggestion to reverse sentence so that the abstract ends on a positive note. "We note that although the WIPoFSD model is unable to represent potentially important features of annual FSD evolution seen with the prognostic model, it is less computationally expensive…possibly making this a stronger candidate for inclusion in climate models."

Manuscript modified as suggested.

P2L23: Are FSD models where shape is fixed always to a power law? If so, note

As far as we are aware, yes. The following clause has been added to the end of the relevant sentence: 'generally to a power law'.

P3L18-19: Note/make clear that the prognostic mixed-layer model is an ocean model

Corrected as suggested. 'prognostic mixed-layer model' has been updated to 'prognostic mixed-layer ocean model'.

P3L34-36: Suggest moving sentence beginning with "Full details…" to earlier in the paragraph, remove the sentence beginning with "Below we provide…"

Above changes made as suggested.

and simplifying "In section…" to "The adaption of lateral melt for FSD models will be introduced in section 2.1.4".

'In section 2.1.4 we will explain how this standard treatment is adapted for use with an FSD model.'

Now reads as:

'The adaptation of the standard CICE lateral melt treatment for use with FSD models is described in section 2.1.4.'

P4L39: subscript a missing from $C^{f,floe}$?

Corrected as suggested.

P6L5: appears that $l_{eff,n}$ has . rather than ,

Corrected as suggested.

Section 2.2.2: It seems that I disagree with Reviewer 1, as I found it disorienting to read this material in the methods, which seems to me to belong in the introduction. In my read, I would suggest P6L8-38 should be in the introduction. One possible compromise would be to add sub-sections to the introduction (uncommon, but a feature I appreciate as a reader) such that the background on brittle fracture doesn't overwhelm the rest of the background presented. Ultimately, I suppose it should be up to the authors how to handle this organizational issue.

Given the disagreement between the reviewers on this point, our preference would be to leave the material referenced in section 2.2.2.

P7L23: missing "such" after sufficiently

Corrected as suggested.

P9~L18-25: Note explicitly that prog-best includes brittle fracture.

The following sentence:

'The prognostic FSD setup, prog-best, uses the standard 12 floe size categories outlined in Roach et al. (2018) and the 5 standard CICE thickness categories (Hunke et al., 2015).'

Now concludes with the following:

'and includes the brittle fracture scheme described in section 2.2.3.'

P9L25: Do you or could you briefly explore the sensitivity to these parameter choices? Or is there some references that could be included here? It would be good to clarify how sensitive the results are/are not to this fitting.

The following sentence has been added to highlight references that present results exploring the sensitivity to the parameter choices:

'Sensitivity studies to these parameter choices have previously been performed for the WIPoFSD model and the version of the prognostic FSD model considered here (i.e. including the brittle fracture scheme) in Bateson et al. (2020) and Bateson (2021a) respectively.'

Section 3.2: Apologies if this is explained elsewhere, but why is only the prognostic model compared to observations, and not WIPo-FSD? I suppose this is because the observational comparison is focused on showing that the brittle fracture improves comparisons, but it seems to me that comparing both would be useful, still. Explain reasoning here (or consider including in figures)

P11L23-24: Just to reiterate the comment above, I think this statement necessitates including the WIPo model results in Fig 4

The core assumption of the WIPoFSD model that is presented in this study is that the FSD can be approximated by a truncated power-law with a singular time-invariant exponent. The power-law distribution shown in all sub-plots uses the same exponent of -2.56 i.e. the same exponent used in *WIPo-best*. In Fig. 4 we only consider floes with a diameter of about 1700 m and smaller. The emergent FSD from the WIPoFSD model will only deviate significantly from the power-law distribution shown in Fig. 4 if $l_{var}$ drops below 1700 m for the snapshots considered; model output shows this is not the case and generally $l_{var}$ is around an order of magnitude larger for the case studies considered. As such, we do effectively compare WIPoFSD model output to observations.

In the first sentence of section 4.1, 'power-law fit' is replaced with, 'power-law fit using the same exponent across all case studies'.

The following clarification has also been added to the first paragraph of section 4.1:

'The power-law fit is used here to represent *WIPo-best*, since the WIPoFSD model is built on the assumption that the FSD can be approximated by a truncated power-law with a singular time-invariant exponent. In practice, the emergent FSD from *WIPo-best* will be identical to the power-law fit shown in Fig. 4 provided the floe size range included is consistently below $l_{var}$, which is the case for all the case studies considered.'

P12L5: I think this sentence perhaps overstates the impact still. Delete "significant", and perhaps add that there is less change the the level of observational uncertainty.

Corrected as suggested. Following clause has been added to relevant sentence: 'with the size of any changes well within observational uncertainty'.

P12L12: delete "generally". Similar to comment above, make more clear that this is well within the observational uncertainty.

Corrected as suggested. Following clause has been added to relevant sentence: 'with any differences between the simulations significantly smaller than the observational uncertainty'.

P12L18: Consider including the percent relative change of MIZ extent for the month with greatest difference?

The statement referred to here, 'Overall, inclusion of FSD processes within CICE results in changes to extent metrics of order 1 x $10^5\ km^2$', refers to the four different time series described in Fig. 6. The maximum percentage change will be different for the different timeseries. To address this, we have included a statement describing the general range that the percentage change spans across the four timeseries.

The following sentence has been added after the relevant statement:

'This corresponds to a percentage change in extent varying between around 1 % to 10 % across the different months and regions considered.'

P12L19: Note that tense is inconsistent across sub-headings

Relevant sub-heading has been updated to ensure consistency in heading style.

P12L30: edit to specifically note that the ribbon shows 2 standard deviation, not just the range

The following statement in brackets has now been removed: 'indicated by the width of the ribbon'.

Figure 7 is now introduced as follows, with the second sentence a new addition:

'Figure 7 shows the percentage difference in the sea ice extent and volume for both *prog-best* and *WIPo-best* relative to *ref* averaged over 2000 to 2016, indicating the impact of each FSD scheme compared to assuming a constant floe size. The shaded region in Fig. 7 covers twice the standard deviation from the mean in each direction.'

P14L21: discriminates should be discriminate

Corrected as suggested.

P18L30: ice-thickness distribution has a hyphen here, but not elsewhere (ice thickness distribution). Suggest the author check throughout paper for consistency (i.e., remove all such hyphens)

Corrected as suggested.

Fig 1: Edits are generally helpful. However, the y-values in the lower subpanel are now inconsistent with the examples shown above. Please edit the bar sizes to agree with the furthest right example, and perhaps make it clear that this is where it's coming from (such as with corresponding letter or box)

Edited as suggested.

Fig 2: delete repeated sentence from caption

Repeated sentence has been removed.

Fig 4: As noted above, should add WIPo-best here, but also prog-best. Otherwise, the link between this figure and result (using prog-16) and subsequent results (with prog-best, 12 categories) is not explicit.

[Figure]

**Figure A:** The perimeter density distribution, $m\ km^{-2}$, of sea ice area as a function of floe size for April MIZ (top left), April pack ice (top right), August MIZ (bottom left), and August pack ice (bottom right). Distributions are shown for *prog-16* (red, cross, dashed) and *prog-best* (purple, diamond, dotted), all averaged over 2000 – 2016.

As discussed in section 3.1 in the manuscript, *prog-best* uses 12 floe size categories as this enables the range of floe sizes relevant to FSD impacts on the sea ice cover via lateral melting and form drag to be sufficiently resolved without excessive computational cost (an increase from 12 to 16 floe size

categories increases CICE run time by about 60%). Given the aim here is to compare two alternative practical ways to model the FSD in sea ice and coupled climate models, it is important that neither method has a prohibitive computational cost. The reason we use output from *prog-16* rather than *prog*-best in the comparison to observations is due to the 'uptick' present in the emergent FSD from the prognostic model. By using 16 floe size categories rather than 12, the 'uptick' falls outside the range of floe sizes included in the comparison. Since the model output is renormalised according to the total sea ice area within the floe size range considered, the presence of the 'uptick' in *prog-best* FSD output within the range of floe sizes considered would preclude a useful comparison between *prog-16* and *prog-best*. Perimeter density from the smallest floe size categories is the most important metric in terms of FSD impact on the sea ice cover (due to the inverse relationship between lateral melt rate / floe edge contribution to form drag and floe size). Therefore, a useful way to address the link between *prog-16* and *prog-best* in the context of this study is to compare model output from both *prog-best* and *prog-16* from the smallest 6 floe size categories i.e. outside of the range of significant uptick influence for both setups. In Fig. A we consider model output averaged over 2000-2016 from both May and June i.e. the main months considered in Fig. 4. Here we show that the differences between the two models are negligible in the smallest few categories i.e. those that most strongly determine FSD impact on the FSD cover. We have added a comment to address this point in the manuscript.

The following comment has been added to section 4.1:

'Whilst in Fig. 4 we consider prog-16 with 16 floe size categories, for comparisons against *WIPo-best* on sea ice behaviour within CICE we consider *prog-best* with 12 floe-size categories since this represents a more practical setup of the prognostic FSD model for use in sea ice and climate simulations, as discussed in section 3.1. In a comparison of model output from *prog-16* and *prog-best* (not presented here) larger differences can be seen in the shape of the distributions in the larger floe size categories due to the presence of the 'uptick' but these differences tend towards negligible in the smallest categories i.e. those most relevant in determining FSD impact on the sea ice cover (e.g. Tsamados et al., 2015; Bateson et al., 2020).'

In addition, we have made minor adjustments to section 3.1 to ensure the reason why *prog-best* and *prog-16* use a different number of floe size categories is clearly explained.

Figures 7,8. Make colors consistent throughout, i.e., WIPo-best always blue, prog-best always red

As suggested, colours have been updated in Fig. 7 to ensure consistency with Fig. 8.

Figure 7 caption: replace "ribbon" with "shading"; remove "the region spanned by"

Corrected as suggested. Relevant sentence now reads as follows:

'The shading shows, in each case, plus or minus two times the standard deviation around the mean.'

Figure 10: I believe I commented on this in the previous round, but I still find it hard to believe that the thickness change is really on the order of meters. Please confirm that this should not be cm?

It is worth noting that a non-linear scale is used in this plot. Changes larger than 1 cm are shown on the plot, and thereafter categories are: 1 cm – 2 cm, 2 cm – 5 cm, 5 cm – 10 cm, 10 cm – 20 cm, 20 cm – 50 cm, 50 cm – 2 m. This quasi-logarithmic scale is used to identify locations where the largest changes are seen whilst also being able to identify broader areas of smaller changes in thickness. Fig. 10 shows that only a few grid cells have changes that exceed 50 cm (and of these even a smaller set will have changes that exceed 1 m). Whilst there is a more substantial area where changes

exceed 10 cm, particularly for *prog-best* in September, changes in thickness mostly do not exceed 10 cm i.e. they are on the order of cm. Note these differences are comparable in order to previous studies using versions of these models e.g. Bateson et al. (2020), Roach et al. (2018).

**Reviewer 2**

This is the second review of the manuscript entitled Sea ice floe size: its impact on pan-Arctic and local ice mass, and required model complexity. I am overall satisfied with the answers to my previous review and the modifications to the manuscript. The work made by the authors has significantly improved the quality of the study in my opinion (congratulations to them), and I would be happy to recommend the paper for publication after some minor adjustments.

A general comment is that the paper is much clearer than it was, but also quite longer and with some repetitions between the results, the discussion and the conclusion. This is not a major problem, and the manuscript could be left as it is, but the authors should not hesitate to remove some comments that are repeated throughout the paper before publication (for instance the reasons behind the uptick).

P1L22: "We demonstrate that a parameterization of in-plane brittle fracture processes enables the prognostic model to achieve a reasonable match against the novel observations". I slightly disagree with this sentence. The study demonstrates that adding a term to the prognostic is needed to achieve a reasonable match against the novel observations, and that there are good reasons to believe that this term represents the effects of in-plane brittle fracture. Also, the manuscript spends some time discussing the effect of this parameterization and why it improves the results, so it might be worth summarizing these results in the abstract. I would therefore suggest writing something like: "We show that adding a term that relaxes the FSD towards a power-law enables the prognostic model to achieve a reasonable match against the novel observations in the summer. We suggest this term represents the effects of in-plane brittle fracture that break the larger floes (>2km) into mid-sized floes (100m-2km)."

We agree that it is important to be precise here on the methodology used, however, since we use a scheme that extends beyond a simple relaxation based on consideration of the relevant brittle fracture derived mechanisms, we are hesitant to describe it as a relaxation scheme in the abstract. We have updated the abstract to provide further details on what has been done and be more precise in describing the impact of the new parameterisation.

The following sentence in the abstract:

'We demonstrate that a parameterisation of in-plane brittle fracture processes enables the prognostic model to achieve a reasonable match against the novel observations.'

Has been updated to:

'We introduce a parameterisation motivated by idealised models of in-plane brittle fracture to the prognostic model and demonstrate that the inclusion of this scheme enables the prognostic model to achieve a reasonable match against the novel observations for mid-sized floes (100 m – 2 km).'

P6L24: "Clearly, brittle fracture events…size." I agree that this is very likely the case for the larger floes (>100m), but more uncertain for smaller floe size. Maybe add a comment on that (if you agree)?

Following sentence:

'Clearly, brittle fracture events can have a direct impact on floe size.'

Has been updated to:

'Clearly, brittle fracture events can have a direct impact on the size of larger floes and potentially also smaller floes.'

P7L18: "stronger regions of sea ice" I am not sure of what you mean by this expression. Regions where the ice is thicker/stronger?

Text has been modified to clarify that stronger in this context effectively refers to thicker ice.

P8L17: "it marks…larger floes": I find the sentence a bit confusing.

Original clause has been clarified. Original clause:

'it marks a transition from a regime where floes are being broken up to a regime where the number of floes is increasing due to the break-up of larger floes.'

Now reads as follows:

'Wave break-up acts to reduce the number of larger floes and increase the number of smaller floes; $l_{var}$ effectively marks the boundary between these two contrasting effects.'

P15L35: "This is an important…models." Yes, but I suspect the quantities you are looking at remain quite constrained by the ocean and atmosphere forcing. I am therefore not certain the conclusion you draw from a stand-alone simulation can be extended to fully coupled climate models. I am aware this is discussed a bit further in the text, but I think it should be mentioned here too.

The following clarification has been added to the end of the relevant sentence: 'though this conclusion needs to be confirmed using fully coupled climate simulations.'

P18L30: Same comment. Maybe replace "is" with "may".

Corrected as suggested.